# Human connectome topology directs cortical traveling waves and shapes frequency gradients

Dominik P. Koller [1,2] ✉, Michael Schirner [1,2,3,4,5] & Petra Ritter [1,2,3,4,5] ✉

Traveling waves and neural oscillation frequency gradients are pervasive in the human cortex. While the direction of traveling waves has been linked to brain function and dysfunction, the factors that determine this direction remain elusive. We hypothesized that structural connectivity instrength gradients − defined as the gradually varying sum of incoming connection strengths across the cortex − could shape both traveling wave direction and frequency gradients. We confirm the presence of instrength gradients in the human connectome across diverse cohorts and parcellations. Using a cortical network model, we demonstrate how these instrength gradients direct traveling waves and shape frequency gradients. Our model fits resting-state MEG functional connectivity best in a regime where instrength-directed traveling waves and frequency gradients emerge. We further show how structural subnetworks of the human connectome generate opposing wave directions and frequency gradients observed in the alpha and beta bands. Our findings suggest that structural connectivity instrength gradients affect both traveling wave direction and frequency gradients.

Cortical traveling waves are signals of neuronal origin, measured e.g., with M/EEG, VSD, LFP, fMRI, that propagate systematically across space and time (e.g., plane waves, expanding waves, spiral waves, or impulse waves). They have been found across brain sites, frequency bands, spatial scales, and behavioral states[1]. Their properties have been linked to memory processes[2–4], visual perception[5–7], motor planning and execution[8], among many other functions. Cortical waves often follow preferred directions: for instance, waves formed by the alpha rhythm travel from parietal to anterior and posterior sites during rest and memory-tasks measured with ECoG[3,9]; waves formed by sleep spindles rotate from temporal to parietal to frontal regions during sleep measured with ECoG[2]; and infra-slow waves propagate from uni- to transmodal functional regions during rest measured by fMRI[10]. Clinical studies have found that schizophrenia[11], ADHD[12], and memory

deficits[13] are related to altered cortical wave directions. Understanding mechanisms of wave direction could yield insights on healthy and pathological cognition. In this work, we propose a mechanism that directs traveling waves − operationally defined as oscillations that show repeated (periodic) spatial propagation of their phase from sources to sinks.

Early theoretical work has shown that distance-dependent connectivity or time delays (Fig. 1a) give rise to traveling waves in weakly-coupled oscillator networks, a frequently used system to study synchronization phenomena[14–16]. Further simulation studies demonstrated that traveling waves can be directed by intrinsic frequency (IF) gradients, where IF is the frequency of an oscillatory unit disconnected from a network (Fig. 1b)[15]. IF gradients are gradual changes of IF across space, e.g., increasing IF along a chain of weakly-coupled oscillators.

[1]Berlin Institute of Health (BIH) at Charité − Universitätsmedizin Berlin, Charitéplatz 1, 10117 Berlin, Germany. [2]Department of Neurology with Experimental Neurology, Charité, Universitätsmedizin Berlin, Corporate member of Freie Universität Berlin and Humboldt Universität zu Berlin, Charitéplatz 1, 10117 Berlin, Germany. [3]Bernstein Focus State Dependencies of Learning and Bernstein Center for Computational Neuroscience, 10115 Berlin, Germany. [4]Einstein Center for Neuroscience Berlin, Charitéplatz 1, 10117 Berlin, Germany. [5]Einstein Center Digital Future, Wilhelmstraße 67, 10117 Berlin, Germany. ✉e-mail: dominik.koller@bih-charite.de; petra.ritter@bih-charite.de

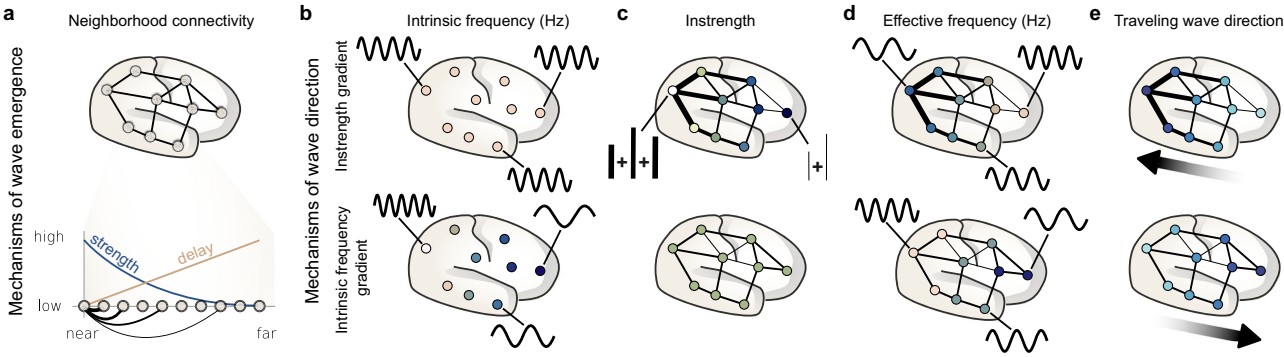

**Fig. 1 | Mechanisms of wave emergence and direction in weakly-coupled oscillator networks. a** Mechanisms of wave emergence are based on neighborhood connectivity. Illustrated is a cortical network where circles and lines represent oscillators and their connections (top). The zoomed in graph (bottom) shows a chain of oscillators and how their connection strengths decrease with distance (blue line; this could also be connection probability), while their conduction delay increases (beige line). These mechanisms create neighborhood connectivity by emphasizing local synchronization between oscillators. **b** Mechanisms of wave direction include instrength gradients (top) and intrinsic frequency gradients (bottom). Intrinsic frequency is the frequency at which the nodes oscillate isolated from the network (connections are removed). Here, the intrinsic frequency is equal across oscillators in the instrength gradient mechanism indicated by the same oscillator color and example activity (black sine curves). In contrast, the intrinsic frequency gradient mechanism is exemplified by a gradual increase of intrinsic frequency along the anterior-posterior axis illustrated by a gradual change of oscillator color. **c** Instrength is the sum of incoming connection strengths. This is illustrated for two oscillators by the addition of connections with different strength

(thick and thin black lines are high and low strength connections, respectively). The instrength increases along the anterior-posterior axis for the instrength gradient mechanism, while it is equal for the intrinsic frequency gradient mechanism. **d** Effective frequency is the frequency assumed by oscillators connected within a network (black lines present). Here, effective frequency is again illustrated by the oscillator color and activity examples. The instrength gradient mechanism generates a smooth effective frequency gradient decreasing along the anterior-posterior axis (opposite of instrength gradient), while the intrinsic frequency gradient mechanism shows clusters with gradually increasing effective frequency along the anterior-posterior axis (same as intrinsic frequency gradient). **e** Traveling waves emerge in both networks from the mechanisms described in (**a**) and are directed by the instrength or intrinsic frequency gradient mechanisms. Traveling waves propagate from low to high instrength oscillators and from fast to slow intrinsic frequency oscillators as illustrated by the thick black arrows and the color gradients. Both mechanisms of wave direction can interact as shown in Fig. 3. Figure 1 re-uses parts of Fig. 5 in ref. 98.

Once coupled in a network, traveling waves propagate from high to low IF oscillators (Fig. 1e). The IF gradient mechanism has been proposed to explain the propagation direction of cortical traveling waves in experimental recordings[3,17] but we lack evidence for IF gradients across the human cortex due to methodological challenges. In non-human animals, IF gradients have been measured invasively by slicing neural tissue into disconnected self-oscillatory units[18,19].

While we speculate that cortical IF gradients exist in humans, we propose an additional mechanism that could affect the direction of traveling waves and is accessible through non-invasive tractography, namely structural connectivity (SC) instrength gradients − the sum of incoming connection strengths (with number of streamlines as proxy for connection strength; also known as weighted in-degree; Fig. 1c). Here, SC instrength gradually changes across cortical space similar to other cortical gradients such as functional connectivity[20], gene expression[21], receptor distributions[22], myelin content[23], cortical thickness[24], or synaptic spine density[25,26]. We postulate that this instrength gradient directs traveling waves from low to high instrength cortical regions. While previous computational studies investigated the emergence of traveling waves (Fig. 1a)[27,28], we focus on their propagation direction (Fig. 1b).

Zhang and colleagues have found that cortical traveling wave direction correlates with effective frequency (EF) gradients[3]. We define EF as the oscillation frequency that emerges when a unit (e.g., cortical region, neuronal population, or weakly-coupled oscillator) is connected to a network (Fig. 1d); this contrasts the self-generated IF of a disconnected unit introduced earlier. EF is the oscillation frequency that we typically estimate in MEG, EEG, or ECoG from the connected cortical network in humans. Zhang and colleagues found that alpha and theta traveling waves measured by ECoG propagated from high to low EF regions but whether this association is causal or correlative remains unknown[3]. Other studies have found large-scale EF gradients across the human cortex but did not investigate traveling waves[29–33]. Previous theoretical studies have shown that increasing instrength

decreases an oscillator's EF in a weakly-coupled oscillator network[34,35]. Thus, we hypothesized that instrength gradients could systematically suppress EFs thereby explaining experimentally observed large-scale EF gradients.

In sum, we investigated if instrength gradients determine both traveling wave direction and EF gradients. We tested this hypothesis with a combination of cortical network models, graph-analysis of human SCs, and analyses of resting-state MEG signals. Our findings suggest that human connectome instrength gradients direct traveling waves and shape EF gradients, thereby unifying both phenomena.

## Results

### Instrength gradients direct traveling waves and shape effective frequency patterns in a 2D network model

We studied if traveling waves followed instrength gradients in a 2D weakly-coupled oscillator network model using Kuramoto oscillators (see Methods)[36]. We chose distance-dependent connectivity and conduction delays because they allow the emergence of traveling waves (Fig. 1a)[16,37,38]. We constructed random networks with connection strength and probability decreasing exponentially with euclidean distance from each oscillator. First, we normalized the connection strengths to create a uniform instrength distribution. Then, we created an instrength gradient by weighting the oscillators' incoming connection strengths with two gaussians placed on the top-right and bottom-left of the network, respectively (Fig. 2a). We calculated the time delays by dividing the euclidean distances between oscillators with a conduction speed of 3 m/s corresponding to estimates in white matter fibers[39–41]. We hypothesized that emerging traveling waves follow the gradient from low to high instrength oscillators (Fig. 1b−e).

We chose an intrinsic oscillation frequency of 10 Hz and simulated 100 random networks (~10% connection probability) for 10 s with random initial phases. We set the global coupling scaling heuristically such that clear traveling waves emerged in our system (see Fig. 2b and Supplementary Movie 1). We assumed that waves travel from sources

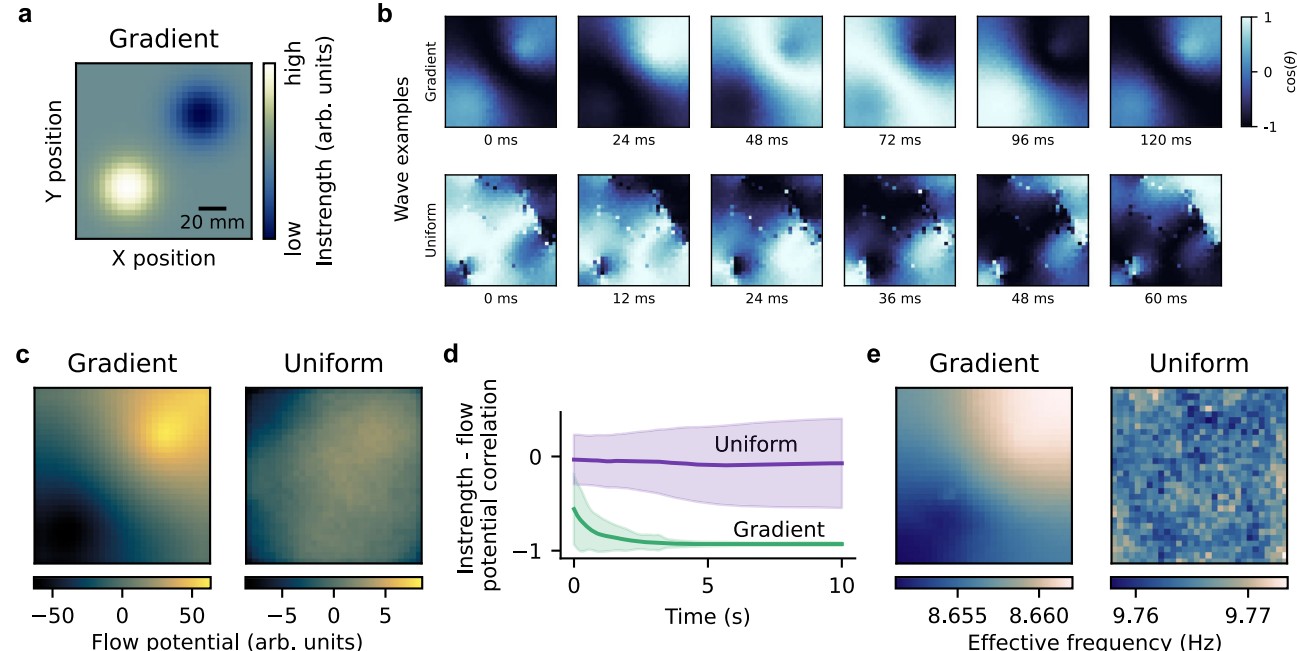

**Fig. 2 | A 2D weakly-coupled oscillator network model produces traveling waves that follow instrength gradients. a** The instrength distribution of the gradient network model increases gradually from the top right corner to the bottom left corner, thereby creating a wave source and sink. **b** The top row shows an example time series of a traveling wave emerging in the network model with an instrength gradient (see **a**; network activity was defined as the cosine of the instantaneous phase). The wave sources out from the low instrength region and propagates to the high instrength region of the network (see Supplementary Movie 1). The time series at the bottom shows traveling and rotating waves emerging in the control network model with a uniform instrength distribution (see Supplementary Movie 2). These complex waves are highly variable across simulations and do not show a systematic direction. **c** The average flow potentials describe how emerging waves propagate across the 2D network models

from higher to lower potentials (100 simulations of 10 s duration initialized with random phases and 1 s initial transients removed). The left figure shows that waves emerging in the gradient network flow along the instrength gradient shown in (**a**). On the right, the average flow potential of the control model with uniform instrength appears unsystematic. **d** The gradient model's flow potential is negatively correlated with the instrength across time and simulations (green; thick lines and shaded areas represent average and standard deviation across simulations). The uniform model's average flow potential hovers around zero across time and simulations show larger variance compared to the gradient model (purple). **e** The average effective frequency pattern generated by the gradient model resembles the instrength gradient (see **a**), while the average pattern in the uniform model appears unsystematic. Source data are provided as a Source Data file.

to sinks and developed a method to detect these singularities (see Methods). Briefly, we computed spatial phase gradients from the oscillators' instantaneous phases and quantified their alignment with an idealized diverging phase gradient around each oscillator using their angular similarity. The angular similarity is 1 if the phase gradients are fully aligned, 0 if they are unorganized or orthogonal, and −1 if they are opposing each other. Therefore, sources and sinks have positive and negative angular similarities, respectively. We detected traveling waves by identifying at least one significant sink or source per timepoint using permutation tests (see Methods).

Using this wave detection method, we found that traveling waves emerged in the network for 100% of time across all simulations (median; see Supplementary Fig. 1 for detailed proportions and distributions). An example traveling wave propagating from top-right to bottom-left shown in Fig. 2b (top) suggested that emerging waves followed the instrength gradient as predicted. We further characterized wave propagation using flow potentials along which waves travel from local maxima to minima (see Methods). The average flow potential across all waves negatively correlated with the synthesized instrength gradient ($r = -0.94$, $p < 0.01$; Fig. 2a, c), indicating that emerging waves traveled from low to high instrength oscillators. Correlating the flow potential of each timepoint with the instrength gradient showed that 94% of all waves across all simulations were instrength-directed (Fig. 2d).

Next, we built a control model with uniform instrength distribution testing if the instrength gradient caused the wave direction. The control model produced waves 39% of the time across all simulations

(Supplementary Fig. 1). Complex spatiotemporal patterns including traveling and rotating waves emerged in the control model (Fig. 2b bottom; Supplementary Movie 2). We did not quantify rotating waves in our analyses but visual inspection suggested they could make up a large proportion of time (for example Supplementary Movie 2). The averaged flow potential across control simulations showed no systematic pattern because wave source and sink locations varied without instrength gradient. As expected, the average flow potential did not significantly correlate with the top-right to bottom-left instrength gradient of the original model ($r = -0.51$, $p = 0.213$; Fig. 2c) and the time-resolved analysis reflected this (Fig. 2d and S1). The average flow potentials and thus traveling wave directions of the gradient and uniform models were not significantly correlated ($r = 0.55$, $p = 0.316$).

Previous studies have found that higher compared to lower instrength suppresses the EF in weakly-coupled oscillator models with time delays (see Supplementary Fig. 6a)[34,35]. We defined EF as the frequency that oscillators assume if connected in a network; this contrasts their IF at which they oscillate in isolation. Thus, we wondered if instrength gradients could generate EF gradients in our 2D network model. To address this question, we calculated the EF patterns across time and simulations for the gradient and uniform models (see Methods). Indeed, we found that the gradient model generated an EF gradient that negatively correlated with the instrength gradient ($r = -0.92$, $p < 0.01$; Fig. 2e left). The generated EF range was small but we will see later that the human connectome generates EFs with a wider range similar to empirical findings (see section The human connectome produces cortical frequency gradients). The uniform

model's EF pattern was unsystematic and uncorrelated with the instrength gradient ($r = 0.06$, $p = 0.558$; Fig. 2e right). EF patterns generated by the gradient and uniform models were not associated ($r = -0.09$, $p = 0.508$). We showed that instrength gradients determine traveling wave direction and generate EF gradients in a 2D network model.

## Instrength and intrinsic frequency gradients interact in a 2D network model

Previous studies proposed that IF gradients direct traveling waves in the human cortex[3,18]. We hypothesize that instrength gradients contribute to traveling wave direction and interact with IF gradients. To investigate this, we generated superimposed instrength and IF gradients with the exact same shape from a gradient template (Fig. 3a). Notably, the two mechanisms generate opposing wave directions from low to high instrength and high to low IF[15], respectively. We scaled the IF gradient from zero (IF is 10 Hz for all oscillators) to range from 8.5 to 11.5 Hz (gradient scaling = 1.5) with a superimposed fixed instrength gradient. The instrength gradient fully directed traveling waves without the IF gradient as assessed by gradient template and flow potential correlation (Fig. 3b top row). This is also reflected in the average flow potential and illustrated for an example wave timeseries (Fig. 3c, e top). The instrength gradient's influence weakened with increasing IF gradient scaling until both balanced each other out (gradient scaling $\cong$ 0.75;

IF gradient ranging from 9.25 to 10.75 Hz; Fig. 3b). At this point, simulated activity varied: we observed spiral waves, plane waves, source-sink waves, and full synchrony (Fig. 3e middle). The average flow potential suggested that waves source from the periphery and sink into the network center (Fig. 3c) but individual simulations' flow potentials varied reflecting the diversity of waves observed. A further increasing IF gradient switched the wave direction from instrength- to IF-directed (Fig. 3b) as shown by the average flow potential (Fig. 3c) and an example timeseries (Fig. 3e).

Earlier, we showed that instrength gradients shape EF patterns (Fig. 2e). We next investigated if instrength and IF gradients could cooperatively shape EF patterns. We found that the correlation between gradient template and EF switched from strongly negative to positive as we increased the IF gradient scaling (Fig. 3b, d). This suggests that instrength and IF gradients together shaped EF gradients in the 2D network model.

Do instrength gradients directly guide traveling waves or do they shape EF gradients which in turn direct traveling waves? We reasoned that EF cannot be a mediating mechanism of wave direction if parameters exist where EF patterns do not match traveling wave direction. We found a set of parameters that produced reliable instrength-directed traveling waves but varying EF patterns (Supplementary Fig. 2) including some that were orthogonal to or opposing the flow potential (Supplementary Fig. 2d, e). We found that all but one flow

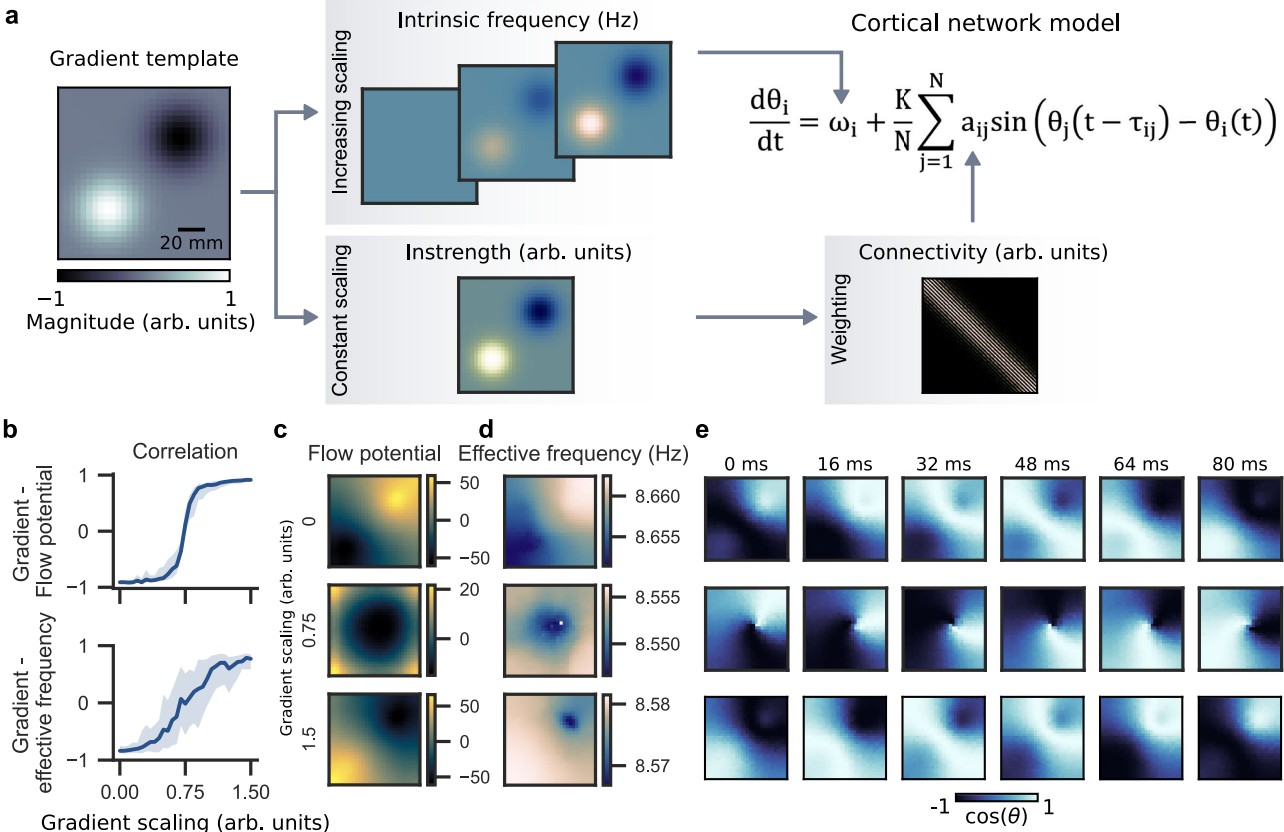

**Fig. 3 | Instrength and intrinsic frequency gradients interact to determine wave direction and shape effective frequency. a** Gradient template used to create scaled intrinsic frequency gradients (0 to 1.5) and the instrength gradient. The intrinsic frequency gradients entered the 2D network model through the intrinsic frequency term $\omega_i$, while the instrength gradient was used to scale the connectivity matrix before it entered through the local coupling term $a_{ij}$. **b** The top figure shows how the gradient template and flow potential correlated depending on the intrinsic frequency gradient scaling. The bottom figure shows how the gradient template correlated with the effective frequency pattern. Both graphs are based on 100 randomly initialized simulations of 10 s duration (1 s initial transient removed) per gradient scaling. The

thick blue lines show the mean correlation and the light blue shaded areas show the standard deviation across simulations. **c** Average flow potentials for three distinct gradient scalings describe the average propagation of traveling waves from higher to lower potentials. **d** Effective frequency maps show how the gradient scaling affects effective frequency patterns. **e** Example timeseries of activity (cosine of instantaneous phases) emerging at three distinct gradient scalings are shown. The top row shows a traveling wave following the instrength gradient. The middle row shows a spiral wave that emerged when instrength and intrinsic frequency gradients were balanced. The bottom row shows a traveling wave that follows the intrinsic frequency gradient. Source data are provided as a Source Data file.

potential across 100 randomly initialized simulations correlated significantly with the instrength gradient, while only one simulation had a significant flow potential – EF correlation (Supplementary Fig. 2a). Thus, instrength gradients guide traveling waves directly and not through mediating EF gradients. However, further systematic investigations of the precise relation between instrength gradients and EF are needed.

Our simulation experiments showed that instrength and IF gradients cooperatively direct traveling waves and shape EF patterns in a 2D weakly-coupled oscillator network. We expect that both mechanisms concert waves and EF in intact neural systems.

### The human connectome hosts instrength gradients
Previously, we showed that traveling waves followed a synthesized instrength gradient in a 2D network model. Do SC instrength gradients exist in the human connectome? To address this question, we studied the SC estimated from diffusion-weighted MRI of 776 participants of the Human Connectome Project (HCP)[42]. First, we estimated fiber tracts with probabilistic tractography between brain regions. Then, we mapped each subject's tractogram to the Schaefer parcellation with 1000 regions[43] and created a group averaged SC with consistency-thresholding (see Methods and Fig. 4a)[44]. We used the number of streamlines as a proxy for connection strength between cortical areas. High-resolution parcellations with approximately equally sized

regions, such as the one used here, are well suited to investigate cortical wave propagation.

Additionally, we estimated the average fiber lengths connecting brain regions. We replicated the common finding that connection strengths decrease with fiber lengths (Fig. 4b)[45]. Fiber lengths increased with increasing euclidean distance while connection strengths decreased (Fig. 4c and Supplementary Fig. 5b). Such distance-dependent connectivity profiles enable traveling wave emergence in weakly-coupled oscillator models (Fig. 1a)[27,37,38,46].

Next, we calculated each brain region's instrength by summing the incoming SC connection strengths and found that instrength is significantly right-skewed indicating that few regions have very large instrength (skewness = 1.33, z = 13.48, p < 0.01) while most have weaker connections (Fig. 4d).

Visually, instrength resembled a spatial gradient increasing from temporal and parietal to frontal and occipital areas (Fig. 4e). To quantify cortical instrength patterns, we used spectrospatial mode analysis – extending classical Fourier analysis to surface meshes (see Methods)[47]. Fig. 4f shows six modes ordered from higher to lower spatial wavelengths or lower to higher spatial frequency. Projecting the instrength onto each mode quantifies their contribution to spatial instrength patterns. We found that low-frequency modes significantly contributed to the group-averaged instrength pattern (Fig. 4g, f). Mode 5 dominated the instrength pattern with a spatial

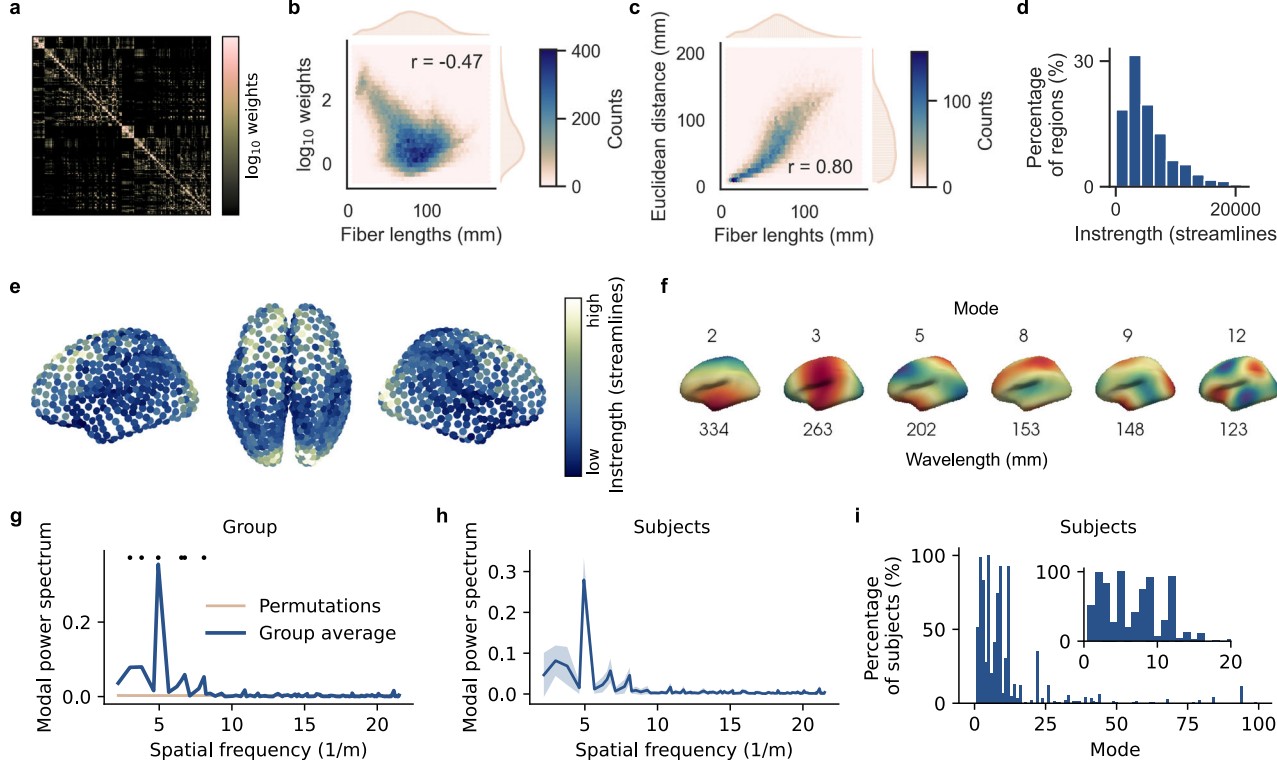

**Fig. 4 | The human connectome hosts instrength gradients. a** Average structural connectivity based on 1000 cortical regions of the Schaefer parcellation ($\log_{10}$ weights threshold at 90th percentile). The connection weights were estimated from 776 subjects of the Human Connectome Project. **b** The log-transformed connection weights are negatively correlated with the fiber lengths. **c** The euclidean distances between connected regions are positively correlated with fiber lengths. **d** The instrength distribution across all regions of the parcellation is right-skewed. **e** Instrength derived from the average structural connectivity projected onto the cortical regions of the Schaefer atlas. An instrength gradient increasing from temporal and parietal areas to frontal and occipital areas is visible. See also Supplementary Fig. 3 for instrength gradients found in other cohorts and parcellations. **f** Modes that significantly explained the group-averaged instrength pattern in (**e**, **g**).

These modes were computed by decomposing the Laplace-Beltrami operator of the parcellated mesh and used for modal spectrospatial analysis (see Methods). Note that the spatial wavelength increases with the modes. **g** Modal power spectrum of the group-averaged instrength pattern (blue). Statistical significance was assessed with one-sided permutation tests (n = 10,000; Bonferroni-corrected; beige line represents mean modal power of all permutations and shaded area the respective standard deviation; see Methods). **h** Average modal power of subject-level power spectra (thick blue line is the mean and the shaded area represents the standard deviation). **i** Percentage of subjects for which significant modes were identified. The inset shows the first 20 modes in more detail. Source data are provided as a Source Data file.

wavelength of 202 mm matching the visually observed gradients (Fig. 4e). We confirmed that low-frequency modes also dominated individual subjects' instrength patterns (Fig. 4h, i). Remarkably, mode 5 contributed significantly to all subjects' instrength patterns (Fig. 4i). We validated instrength gradients in distinct cohorts and parcellations. The three validation sets showed instrength gradients (Supplementary Fig. 3a, e, i) and low-frequency modes significantly dominated their modal spectra on the group- and subject-level (Supplementary Fig. 3b, c, d, f, g, h, j).

In summary, we found that the human connectome hosts low-frequency instrength gradients across different cohorts and parcellations.

## The human connectome directs traveling waves in a cortical network model

Connection strength and probability decrease with distance from a brain region (Fig. 4b, c)[45]. This connectivity pattern along with conduction delays allows the emergence of cortical traveling waves (Fig. 1a). Instrength gradients identified in the previous section suggest that the human connectome could not only produce but also direct traveling waves. We explored this possibility in a cortical Kuramoto model with 1000 brain regions connected through the empirically estimated SC (see Methods and Fig. 4a). For our initial experiments, we chose a 10 Hz IF and conduction delays calculated by dividing the tractography-derived average fiber lengths with a biologically realistic conduction speed of 3 m/s (see parameter exploration for a broader range of conduction speeds). Resulting conduction delays depended on distance because fiber length depends on euclidean-distance (Fig. 4c). We simulated this model 100 times for 10 s with random initial phases.

We saw that traveling waves emerged in our cortical network model (Fig. 5a and Supplementary Movie 3) and detected waves using the method described earlier (see Methods). Traveling waves emerged 87.2% of time across all simulations (median; Supplementary Fig. 3a shows detailed proportions and distributions for left and right hemispheres for all cortical models). The average flow potential across detected waves correlated significantly with the instrength gradient found in the human connectome ($r = -0.74$, $p < 0.01$; Figs. 4e and 5b) suggesting that waves travel from temporal and parietal to frontal and occipital areas (Fig. 5b - Model). More specifically, we found that all emerging traveling waves followed the instrength gradient (see Supplementary Fig. 4a). Over time the average instrength – flow potential correlation across all simulations stabilized around −0.7 for both hemispheres (Fig. 5b). In conclusion, instrength gradients directed traveling waves in our cortical network model.

We built several control models to rule out alternative explanations for directed traveling waves. In our first control model, we randomly shuffled the connection strengths between existing connections to destroy the instrength gradient (Fig. 5b - Shuffled connections; Supplementary Movie 4). Shuffling also eliminates the connection strength's dependence on euclidean distance. Consequently, almost no traveling waves emerged across simulations (Supplementary Fig. 4a) and the average flow potential did not correlate with the original instrength gradient ($r = 0.02$, $p = 0.808$). Their moment-to-moment correlation remained close to zero throughout simulations (Fig. 5 - Shuffled connections).

Next, we built a control model that preserved the connection strength's dependence on euclidean distance while destroying the original instrength gradient (Fig. 5b - Distance dependent connections; Supplementary Movie 5). We fit an exponential model to the empirical euclidean distance – connection strength relationship and synthesized a surrogate network (see Methods and Supplementary Fig. 5a, b). While this model produced traveling waves across all time and simulations they did not follow the original instrength gradient (average flow potential – instrength correlation: $r = -0.04$, $p = 0.862$; Fig. 5b and

Supplementary Fig. 4a). Instead, the average flow potential suggested frontal to parietal wave propagation determined by the instrength gradient that emerged from the network's generating process and the cortical regions' spatial embedding.

To understand the role of conduction delays, we removed time delays while preserving the original SC and instrength gradients (Fig. 5b - Zero delay; Supplementary Movie 6). We found that waves emerged 24.1% of time. Surprisingly, none of these waves followed the original instrength gradient (Fig. 5b - Zero delay and Supplementary Fig. 4a) reflected by low average flow potential – instrength correlation ($r = -0.02$, $p = 0.853$). Thus, conduction delays are crucial for the instrength gradient mechanism of wave direction.

We wondered if distance-dependent conduction delays are important for instrength-directed traveling waves or if a constant conduction delay is sufficient. We replaced distance-dependent conduction delays with their average delay across all connections (23 ms; Fig. 5b - Constant delay; Supplementary Movie 7). This control model produced traveling waves for 96.3% of time and 95.1% of those waves followed the instrength gradient (Supplementary Fig. 4a). Consequently, the average flow potential was significantly correlated with instrength ($r = -0.64$, $p < 0.01$). These findings show that distance-dependent or constant time delays along with an instrength gradient direct traveling waves.

We also investigated a control model with instrength-normalized SC to outrule node-degree influences on wave direction (see Methods). We found that this model produced traveling waves 94.7% of times but none of them followed the original instrength gradient (Supplementary Fig. 4a). Notably, a systematic average wave potential indicated that emerging waves propagated from visual cortex and medial frontal sites to the temporal lobe and lateral frontal sites (Fig. 5b – instrength-normalized; Supplementary Movie 8) suggesting a distinct mechanism directing those waves. We tested if the systematic wave potential could be explained by node degree ($r = 0.11$, $p = 0.049$), betweeness centrality ($r = 0.1$, $p = 0.011$) or eigenvector centrality ($r = 0.14$, $p = 0.292$) but we could not identify a significant relationship. Hub-structure did neither determine instrength-directed waves nor systematic wave propagation emerging in this model. However, the emerging EF correlated significantly with the wave potential ($r = 0.39$, $p < 0.01$). Further research needs to identify the mechanism at play in this model.

Finally, we investigated if traveling waves emerge and follow instrength gradients in a more realistic cortical Jansen-Rit model (Supplementary Fig. 4a–d and Supplementary Movie 9) where a pyramidal neuron population excites inhibitory and excitatory interneuron populations that provide feedback. Each region communicates with other brain regions through the pyramidal population. We set the Jansen-Rit regions to achieve approximately 10 Hz local field potential oscillations within the network (see Methods and Supplementary Table 1). This model expressed traveling waves for 71.8% of time across all simulations and of those waves 41.7% followed the instrength gradient (Supplementary Fig. 4a). The average flow potential significantly correlated with instrength ($r = -0.63$, $p < 0.01$) suggesting that traveling waves emerge and follow instrength gradients in a more realistic model and can be understood with the weakly-coupled oscillator mechanism.

Further, our results were robust in cortical network models with added noise (Supplementary Fig. 4e–g; Supplementary Movie 10) and random IF dispersion (Supplementary Fig. 4h–j; Supplementary Movie 11).

Returning to our original cortical network model, we studied how IF, conduction speed, and global coupling scaling affect traveling wave emergence and direction (Fig. 6 and Methods). We explored four IFs roughly reflecting delta, alpha, beta and gamma rhythms (1, 10, 20, 40 Hz); conduction speeds from 1 to 10 m/s inspired by experimental estimates from white matter fibers[39–41]; and global coupling scalings from $10^{-5}$ to 10. We calculated proportions of traveling waves and

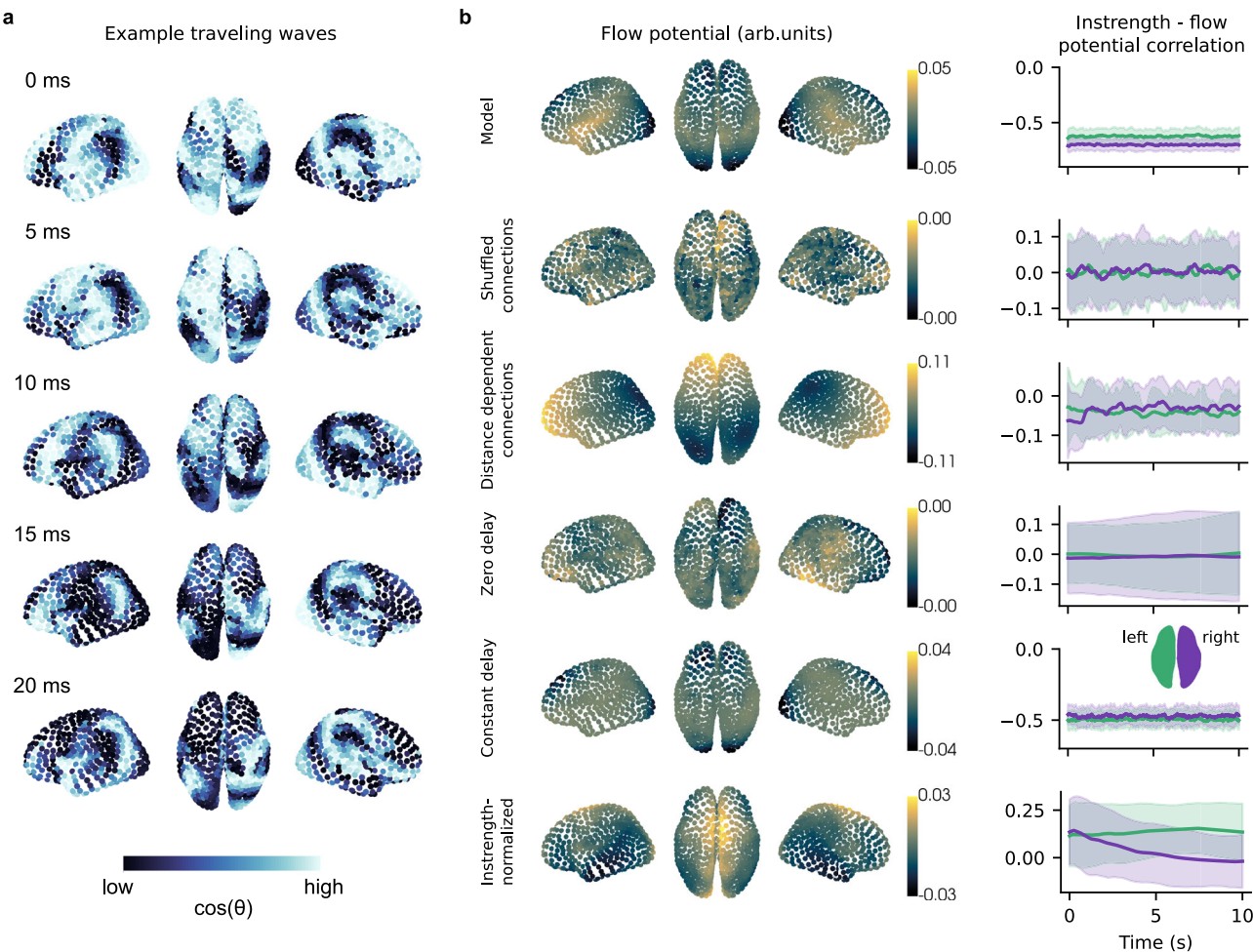

**Fig. 5 | Traveling waves emerge in the cortical network model and follow the instrength gradient of the human connectome. a** Example time series showing traveling waves emerging in the cortical network model (Network activity was defined as the cosine of the instantaneous phase; see Supplementary Movie 3). **b** Average flow potentials show how emerging waves propagate across the cortical network model and controls (left; 100 simulations of 10 s duration initialized with random phases and 1 s initial transients removed). The instrength – flow potential correlation shows how wave propagation changes in time (right; thick lines and shaded areas represent averages and standard deviations across simulations). Model: the average flow potential of the original cortical network model suggests that emerging waves propagate from temporal and parietal to frontal and occipital regions (see Supplementary Movie 3). The instrength – flow potential correlation indicates a strong negative correlation across time and simulations for both cortical hemispheres. Shuffled connections: after shuffling the connection weights within existing connections the average flow potential appears random. The corresponding instrength – flow potential correlation suggests that waves do not consistently propagate along the original instength gradient (see Supplementary

Movie 4). Distance dependent connections: synthesizing a connectome based on exponentially decaying connection strengths results in waves propagating from frontal to parietal areas (see Supplementary Movie 5). These waves do not follow the empirical instrength gradient of the human connectome as can be seen from the instrength – flow potential correlation. See also Supplementary Fig. 5a for a comparison of the empirical and synthesized left hemisphere SCs and the connection strength – euclidean distance relationship used to derive the exponential model. Zero delay: if time delays are removed emerging traveling waves do not follow the instrength gradients (see Supplementary Movie 6). Constant delay: using an average time delay instead of distance-dependent delays results in pronounced traveling waves that follow the instrength gradient as visible from the average flow potential and the instrength – flow potential correlation (see Supplementary Movie 7). Instrength-normalized: normalizing the structural connectivity by instrength created a systematic flow potential that does not resemble the original instrength gradient. The instrength – flow potential correlation shows that emerging waves do not follow the original instrength gradient (Supplementary Movie 8). Source data are provided as a Source Data file.

instrength-directed waves across ten randomly initialized simulations per parameter combination.

Traveling waves emerged (Fig. 6 - top row) and followed the instrength gradient (Fig. 6 - mid row) for a wide range of parameter combinations in all frequency bands. A negative instrength – flow potential correlation suggested that emerging waves primarily flow from low to high instrength regions (Fig. 6 - bottom row). Cortical network models did not generate traveling waves for very low or high coupling scalings (Fig. 6 - top row). Presumably, low coupling scalings did not exceed the critical coupling sufficient for partial synchronization to generate traveling waves. Very large coupling scalings resulted in full synchronization or erratic behavior. The wide range of parameter combinations where traveling waves emerged lies between

these two extremes. We observed that cortical network models with higher IF required faster conduction speeds to sustain traveling waves. Simulated traveling waves propagated at speeds consistent with empirical large-scale cortical waves (<5 m/s; Supplementary Fig. 6b)[2,3].

We showed that traveling waves emerge and follow instrength gradients for a variety of IFs, coupling scalings, and conduction speeds in cortical network models with empirical SC.

**The human connectome produces cortical frequency gradients**
Earlier, we showed that instrength gradients shaped EF gradients – the frequency assumed by connected oscillators – in a 2D network model. We wondered if instrength gradients in the human connectome could produce EF gradients similar to those found empirically[3,31].

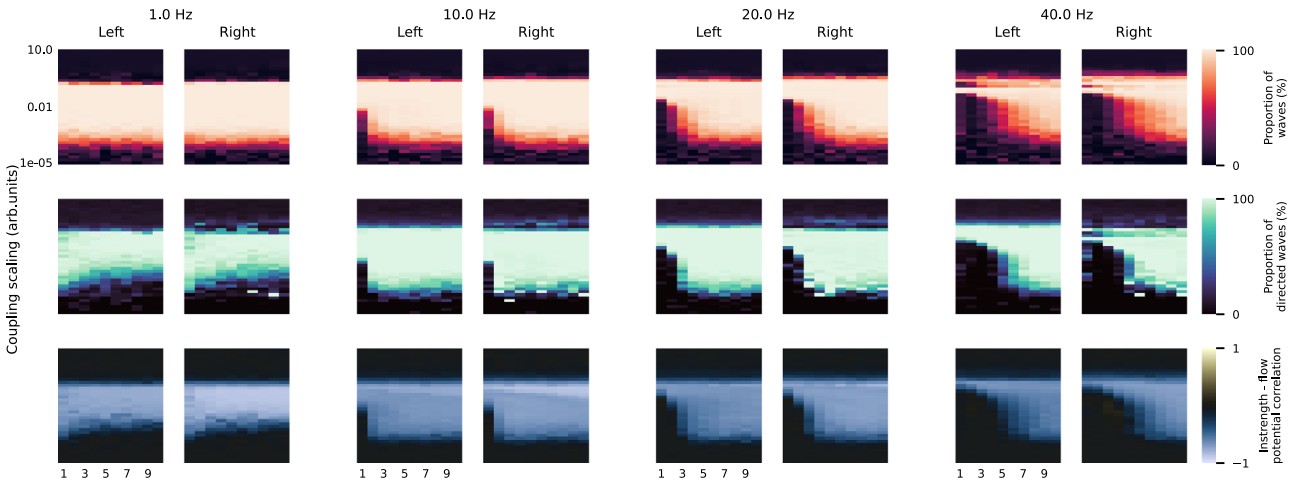

**Fig. 6 | Instrength-directed traveling waves emerge in a cortical network model within a wide range of parameters.** This figure shows the average proportion of time where traveling waves were detected (first row), the average proportion of waves that are directed by instrength (second row), and the instrength – flow potential correlation (third row) for each hemisphere (left, right). The parameters explored were the intrinsic frequency (columns), the global coupling scaling, and conduction speed. All metrics were estimated from 10 cortical network simulations with a duration of 10 s per parameter combination (phases were initialized randomly; 1 s initial transients removed). See also Supplementary Fig. 4 for estimated wave speeds and average EF across time, regions, and simulations. Source data are provided as a Source Data file.

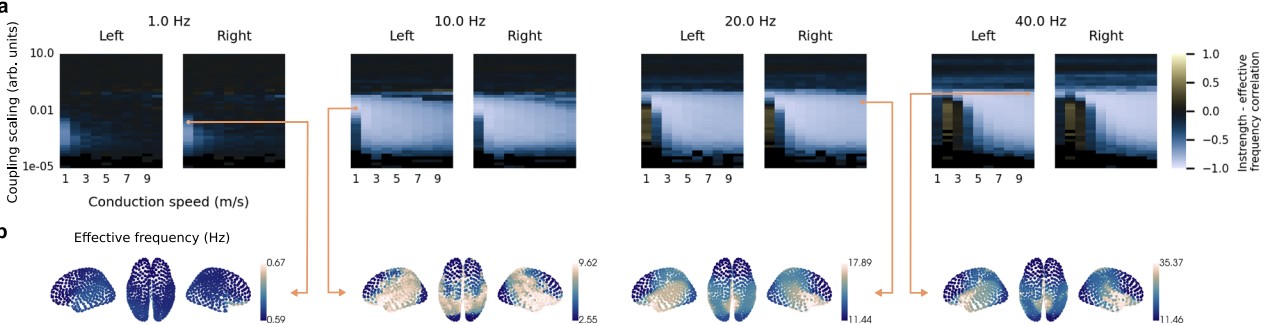

**Fig. 7 | Effective frequency gradients co-emerge with traveling waves in a cortical network model. a** Parameter exploration of average instrength – effective frequency correlation for each hemisphere (left, right). The parameters explored were the intrinsic frequency, the global coupling scaling, and conduction speed (see Fig. 6). All metrics were estimated from 10 cortical network simulations with a duration of 10 s per parameter combination (phases were initialized randomly; 1 s initial transients removed). **b** Examples of different effective frequency patterns (thresholds at 5th and 95th percentiles). Different parameter combinations resulted in gradient- or cluster-like effective frequency patterns. See also Supplementary Fig. 6a for estimated average effective frequency across time, regions, and simulations. Source data are provided as a Source Data file.

To study this question, we calculated the simulated average EF of each region for the same parameter combinations introduced earlier (see previous section). We found that EF patterns negatively correlated with instrength gradients for most parameter combinations where traveling waves emerged (compare Fig. 6 and Fig. 7a) suggesting that they propagated from fast to slow oscillators. The EF pattern differed from the instrength gradient for most delta band models that produced waves, possibly affected by the nearby non-oscillatory regime.

Inspecting EF patterns at distinct parameter combinations revealed diverse arrangements from smooth gradients closely resembling the instrength gradient (Fig. 7b - 10 Hz) to more clustered patterns (Fig. 7b - 1, 20 and 40 Hz) and from narrow (Fig. 7b - 1 Hz) to wide (Fig. 7b - 40 Hz) EF ranges. Notably, these patterns existed in all IF bands. Some of the alpha band EF patterns closely resembled the EF pattern observed in ECoG recordings during a memory task (Fig. 7b - 10 Hz)[3]. Our cortical network model also generated EF gradients of similar range but different structure compared to previously observed resting-state MEG EF gradients[31].

We found that EF patterns did not emerge in control models with randomly shuffled connections or zero delay (using the cortical network models presented in Fig. 5). Additionally, the original instrength gradient differed from the EF gradient of the control model with euclidean distance-dependent connection strength relationship ($r = -0.07$, $p = 0.638$).

Our findings show that large-scale cortical EF gradients could be a network effect resulting from conduction delays and instrength gradients.

## Simulated instrength-directed traveling waves and smooth frequency gradients are consistent with MEG resting-state functional connectivity

Connectome instrength gradients are relatively stable over time because white matter changes on slow timescales[48,49]. We hypothesize that permanent instrength gradients affect traveling waves across brain states but their influence is probably pronounced during resting-state without task-dependent modulation.

Previous studies have found that traveling waves coordinate functional connectivity (FC)[10,50]. We investigated if instrength-directed traveling waves and EF gradients are consistent with this framework by fitting our cortical network model to resting-state MEG FC of 80

healthy HCP subjects. We reconstructed the source activity at the 1000 Schaefer atlas regions from MEG sensor recordings filtered in the alpha, beta and gamma bands (see Methods). Next, we computed frequency-specific average FCs across subjects using the phase locking value (PLV)[51] – a measure of neural synchrony. Similarly, we estimated simulated PLV-FC for our parameter exploration presented above (excluding the delta band due to HCP filter settings). The resting-state alpha PLV-FC and an example simulated alpha PLV-FC are shown in Fig. 8b. We found that across frequency bands simulated and empirical PLV-FC fit increased if directed traveling waves emerged (Fig. 8a). The best fitting cortical network models had a PLV-FC correlation above 0.56 (alpha: 0.564, beta: 0.591, gamma: 0.594; Fig. 8c–e) and instrength correlated strongly with flow potential (alpha: −0.66, beta: −0.65, gamma: −0.65) and EF patterns (alpha: −0.85, beta: −0.86, gamma: −0.87). In contrast to clustered EF patterns that some cortical network models produced (Fig. 7b) the best fitting models produced smooth alpha, beta, and gamma EF gradients (Fig. 8c−e).

Signal volume conduction can result in spurious zero-lag phase relations with high PLV. We additionally analyzed FC based on the phase lag index (PLI)[52], which ignores spurious but also true zero-lag interactions. Resting-state alpha PLI-FC decreased notably compared to the PLV (Fig. 8g, b). We found that simulated and empirical PLI-FCs strongly correlated in the alpha band ($r = 0.46$), while beta and gamma bands had weaker correlations (beta: 0.36; gamma: 0.14; Fig. 8f). This suggests that non-zero-lag phase interactions could be frequency-specific; we investigated this further in the next section.

We also explored if simulated cortical network dynamics exhibit metastability and found similar synchronization variability and state dwell times between the best fitting cortical network models and resting-state MEG (Supplementary Fig. 7)[27,53–55].

We found that our cortical network model achieved high correlation with empirically derived PLV-FCs if directed traveling waves emerged and smooth EF gradients were produced across the alpha, beta, and gamma bands. PLI-FC fit decreased from alpha to beta to gamma band. Our findings suggest that zero-lag FC across frequency bands could be coordinated by cortical traveling waves following SC instrength gradients, while non-zero-lag FC is more specific to the alpha band.

## Connectome subnetworks explain effective frequency gradients and traveling wave direction in alpha and beta bands

Mahjoory and colleagues[31] found large-scale EF gradients in resting-state MEG; alpha EF gradients increased along the anterior-posterior axis while beta EF gradients decreased. Does our cortical network model with instrength gradients generate EF patterns consistent with those observations?

We computed resting-state MEG EF gradients in the alpha and beta bands (see Methods). We found an alpha EF gradient increasing from occipital to prefrontal areas, while a beta EF gradient increased from prefrontal to occipital regions (Fig. 9a). Our findings corroborate the EF gradients that Mahjoory and colleagues observed[31]. Next, we explored the fit between empirical and simulated EF for the same coupling scalings and conduction speeds as before. We used the concordance correlation coefficient (CCC) to evaluate empirical and simulated EF fit (see Methods). CCC comprehensively measured if the empirical and simulated EFs were correlated and of similar magnitude.

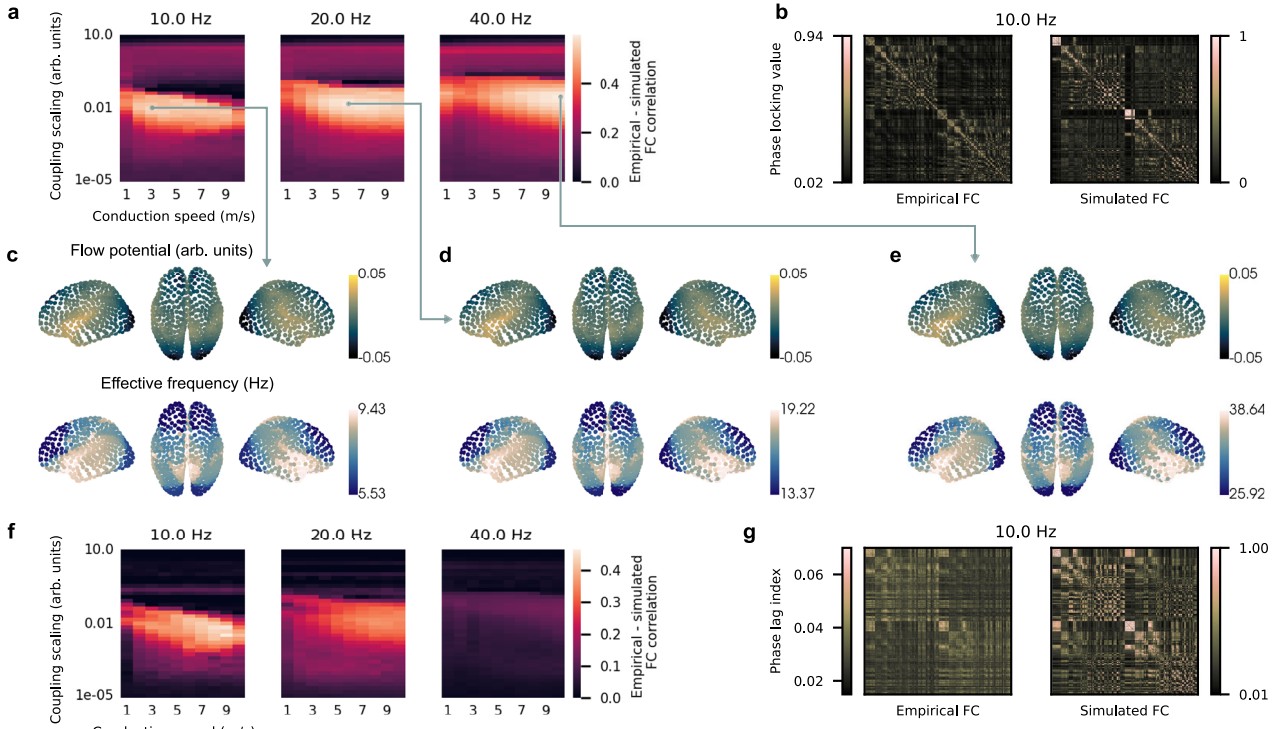

**Fig. 8 | Fitting cortical network models with MEG functional connectivity yields instrength-directed traveling waves and smooth effective frequency gradients.** **a** Average correlation between empirical MEG and simulated functional connectivity (FC) for different intrinsic frequencies, coupling scalings, and conduction speeds (see Figs. 6 and 7; filter settings of the Human Connectome Project pipeline excluded the delta band from this analysis). The empirical and simulated FCs were estimated using the phase locking value (PLV)[51]. The average empirical PLV-FC was determined from source-reconstructed resting-state MEG activity of 80 subjects that participated in the Human Connectome Project. The average

simulated PLV-FC was calculated from 10 simulations of the cortical network model (phases were randomly initialized; 1 s initial transients removed). **b** Empirical and simulated (best fit) PLV-FC for the alpha band (10 Hz). **c−e** Average flow potentials and effective frequency patterns (thresholds at 5th and 95th percentiles) for the best fitting cortical network models with 10, 20, and 40 Hz intrinsic oscillation frequency. **f** Average correlation between empirical MEG and simulated FC estimated using the phase lag index (PLI)[52]. **g** Example empirical and simulated PLI-FC for the alpha band (10 Hz). Source data are provided as a Source Data file.

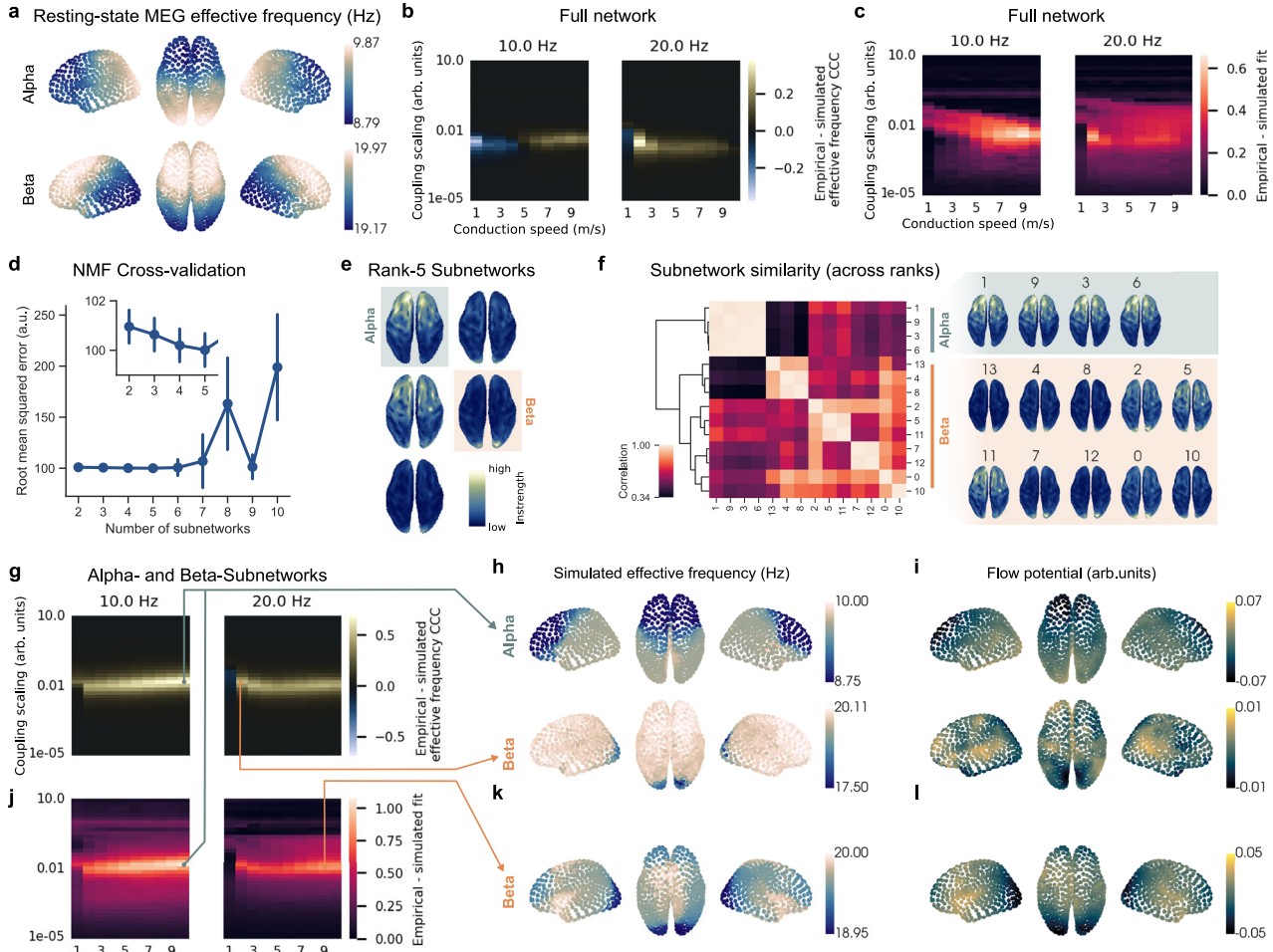

**Fig. 9 | Alpha- and beta-subnetworks match resting-state MEG effective frequency gradients and functional connectivity. a** Average alpha and beta effective frequency gradients measured across HCP resting-state MEG of 80 subjects. **b** Parameter exploration of cortical network model using the full network (structural connectivity shown in Fig. 4) and the concordance correlation coefficient (CCC) between empirical (see **a**) and simulated effective frequency. **c** Parameter exploration of the full network with the combined PLI-FC and effective frequency fit. **d** Cross-validation of root mean squared error between training and test data-points estimated by nonnegative matrix factorizations (NMF) with different numbers of subnetworks (blue dots and bars represent the mean and standard deviation across 100 randomly initialized NMF runs). The inset shows a zoom-in of the root mean squared error of rank-2 to -5 NMFs. **e** Instrength patterns of the rank-5 NMF subnetworks projected onto the cortical surface with highlighted alpha- and beta-subnetworks used for simulations. **f** Pairwise correlation matrix between

subnetwork components across rank-2 to -5 NMFs (total of 14 components; see inset in **d**) ordered in hierarchical clusters (dendrogram in black). The two high-level clusters show subnetwork instrength patterns decreasing and increasing along the anterior-posterior axis corresponding to putative alpha- and beta-subnetworks. **g** Empirical − simulated effective frequency CCC for parameter exploration of alpha- and beta-subnetworks (compare with **b**). **h, i** show the average effective frequency and flow potentials estimated from alpha- and beta-subnetwork simulations at the best effective frequency CCC fit (across 10 randomly initialized simulations). **j** Empirical − simulated PLI-FC + effective frequency CCC fit for parameter exploration of alpha- and beta-subnetworks (compare with **c**). **k, l** show the simulated average effective frequency at the best fit for the beta-subnetwork in (**j**). The best fit for the alpha-subnetwork coincided with the one obtained for the effective frequency CCC fit in (**g**). Source data are provided as a Source Data file.

If two measurements are fully correlated and have the same magnitude the CCC is 1, if they are fully anti-correlated and have the same magnitude it is −1, and if they are not correlated and of different magnitude it is 0. We found parameter regions where empirical and simulated EFs fit best (Fig. 9b).

Alpha EF fit peaked with a CCC of 0.2 at a conduction speed of 9 m/s and medium coupling scaling. This model's alpha EF gradient strongly correlated with the instrength gradient ($r = -0.73$; Fig. 7a) but frontal EF was lower than occipital EF (Supplementary Fig. 8a). Instrength-directed alpha traveling waves emerged across all time-points and simulations (Fig. 6 and Supplementary Fig. 8a).

Beta EF fit peaked with a CCC of 0.38 at 2 m/s and medium coupling scaling (Fig. 9b) and the emerging EF gradient increased from posterior to anterior sites (instrength − EF correlation = −0.39; Supplementary Fig. 8a). We detected beta traveling waves in 38% of timepoints

but only 6% were instrength-directed (Fig. 6). The wave flow potential suggested frontal and temporal sources along with sinks in parietal areas and the temporal poles (Supplementary Fig. 8a) but beta traveling waves were not noticeable in corresponding Supplementary Movies.

Next, we studied the combined PLI-FC and EF fits and found similar best fit parameters to the ones obtained by EF fit only (alpha fit = 0.66 at 9 m/s and slightly lower coupling scaling; beta fit = 0.57 at 2 m/s and same coupling scaling; Fig. 9c) resulting in similar EF patterns and wave potentials for both models. In sum, while the cortical network model partially explains the empirical EF gradients, our results were ambiguous for emerging beta traveling waves.

We next explored if SC subnetworks could better explain experimentally observed opposing alpha and beta EF gradients and traveling waves[31,56]. Such putative subnetworks could be cortical layer- or frequency-specific[57,58].

To identify SC subnetworks, we used nonnegative matrix factorization (NMF) an unsupervised algorithm that decomposes nonnegative high-dimensional data into nonnegative low-dimensional components and associated loadings (see Methods)[59]. Because of the nonnegativity constraint, we can interpret NMF components as additive parts of a whole, for example subnetworks of the full SC network.

We constructed a large matrix where each column was associated with the flattened SC of individual HCP subjects (see Methods). Given the subjects' SCs, NMF finds subnetworks that are common across SCs. Determining the optimal number of components − in our case subnetworks − remains challenging. If the number of components is too small, they might miss important detail and if it is too large, they capture noise. We determined the number of subnetworks using rank-2 to -10 NMF cross-validation that randomly zeroed and imputed datapoints for training and testing, respectively[60,61] (Fig. 9c). Five subnetworks minimized the error between held-out original values and imputed datapoints. Some of these subnetworks' instrength gradients matched our expectations for alpha- and beta-specific SCs (Fig. 9d).

To validate subnetwork stability, we compared their similarity across rank-2 to -5 NMFs (14 subnetworks in total) – the ranks with low reconstruction error (Fig. 9c). Hierarchical clustering identified two groups of subnetworks whose instrength patterns decreased or increased along the anterior-posterior axis as would be expected for alpha- and beta-subnetworks (Fig. 9e). In sum, NMFs found consistent alpha- and beta-subnetworks across ranks that explained up to 63% of SC variance.

We built two rank-5 NMF subnetwork models to test if they generated EF gradients and traveling wave directions consistent with empirical observations. We chose the putative alpha- and beta-subnetworks with the strongest positive and negative correlation between instrength and posterior-anterior position (Fig. 9d). Combining the alpha- and beta-subnetworks explained 36% of SC variance; this network's instrength strongly correlated with the total SC's instrength ($r = 0.96$, $p < 0.01$). The alpha- and beta-subnetwork models improved the EF fit drastically compared to the original cortical network model (compare Fig. 9b, f). The alpha band CCC was 0.69 at 10 m/s and medium coupling scaling, while the beta band CCC was 0.50 at 2 m/s and medium coupling scaling. Empirical and simulated EFs in the alpha and beta bands at peak CCC were similar with opposing anterior-posterior gradients (compare Fig. 9a, g). While the alpha wave potential suggested posterior to anterior propagation (Fig. 9i top), beta wave potentials were more complex (Fig. 9i bottom). Example Supplementary Movies of beta waves did not show noticeable waves. We replicated these findings using the rank-2 NMF subnetworks – with fewer parameters and similar reconstruction error (Fig. 9d inset and Supplementary Fig. 8c−f).

Using the combined FC and EF fit, we found parameters for the rank-5 alpha- and beta-subnetworks that improved the fit by 62% and 47% compared to the original cortical model (Fig. 9j). The parameters for the alpha network coincided with the ones found with the EF CCC and this model's empirical – simulated PLI-FC correlation peaked at 0.38 (Fig. 9g). This network generated traveling waves 88.7% of time across 100 simulations and the alpha-subnetwork's instrength gradient directed all emerging waves (see Supplementary Fig. 4a and Supplementary Movie 12). The subnetwork's instrength gradient correlated significantly with the average wave flow potential ($r = -0.75$, $p < 0.01$) and effective frequency pattern ($r = -0.77$, $p < 0.01$). The beta network model fit best at an increased conduction speed of 9 m/s with an almost unchanged EF CCC of 0.49, while the PLI-FC fit increased from 0.31 to 0.40. This beta-subnetwork model showed a less pronounced anterior-posterior EF decrease compared to the best EF CCC fit (Fig. 9k). The wave potential suggested beta waves travel from frontal and temporal to posterior but also anterior sites (Fig. 9l). Traveling waves emerged 87.8% of time and all of them were instrength-directed

(see Supplementary Fig. 4a and Supplementary Movie 13). The beta-subnetwork's instrength gradient correlated significantly with the average wave flow potential ($r = -0.78$, $p < 0.01$) and effective frequency pattern ($r = -0.84$, $p < 0.01$). We achieved comparable results using the 2-rank NMF subnetworks (Supplementary Fig. 8g−i).

We showed that the full SC network can be decomposed into additive subnetworks whose instrength gradients generated alpha- and beta-band EF gradients and traveling wave directions matching empirical data while maintaining high PLI-FC.

## Discussion

We hypothesized that SC instrength gradients of the human connectome affect cortical traveling wave direction and EF gradients. To test this, we studied a 2D network of weakly-coupled oscillators that allowed traveling wave emergence. When we synthesized an instrength gradient, waves traveled from lower to higher instrength oscillators while EF gradients emerged. We showed that this mechanism can co-exist and interact with the previously studied IF gradient mechanism (Fig. 1b)[15]. Analytical expressions derived in earlier studies suggest that higher instrength nodes phase-lag, while lower instrength nodes phase-lead in coupled oscillator models[62–64]. When combined with spatial instrength gradients, this analytical framework could be a theoretical foundation for the instrength gradient mechanism.

After demonstrating that instrength gradients direct traveling waves in a 2D network model, we explored if gradients exist in the human connectome and found that instrength increased from temporal and parietal towards frontal and occipital regions. In contrast to previous studies[65–67], we quantified instrength patterns statistically with spectrospatial mode analysis and found similar gradients across different cohorts and parcellations. We only studied SCs with similar-sized parcels, high spatial resolution (≥400 regions), and across many subjects (≥70 subjects) to ensure high quality estimates. While our instrength patterns were relatively consistent compared to other studies[65–67], differing processing pipelines could explain variations between SCs. For example, normalizing the streamline count between brain regions using either brain region size or fiber lengths can impact instrength patterns[65]. Additional processing such as thresholding individual or group averaged SCs can further affect instrength patterns[44]. Gajwani and colleagues[67] extensively studied how 1760 distinct processing pipelines for diffusion-weighted MRI affected instrength patterns. They found that different tractography algorithms and parcellations had the largest influence on instrength topographies. Developing methods and processing pipelines that reliably estimate connection strengths will help understanding how instrength gradients direct traveling waves and shape EF gradients.

We showed that instrength gradients existing in the human connectome affect traveling wave direction in a cortical network model. Emerging traveling waves propagated along instrength gradients from temporal and parietal (low instrength) to frontal and occipital areas (high instrength). Shuffling the connection strengths and thereby corrupting their distance-dependence abolished traveling waves. This finding corroborates that distance-dependent connection strengths are crucial for waves to emerge (Fig. 1a)[27]. Removing time delays between network regions resulted in varied traveling wave directions. Thus, neglecting finite time delays in cortical network models has drastic consequences for the spatiotemporal dynamics of simulated activity. Interestingly, we found that instrength-directed traveling waves still emerged in a cortical network model with uniform average time delays suggesting that large-scale traveling wave directions could be robust to specific changes of conduction delays, for example in response to myelin plasticity[68]. Pang and colleagues[69] showed that a neural field model encoding the cortical geometry generated wave-like activity and explained fMRI functional connectivity. They further found that geometric eigenmodes explained fMRI variability better compared to connectome eigenmodes. This difference nearly

vanished when they used a stochastic exponential distance rule to synthesize a connectome. Intriguingly, an instrength gradient increasing from frontal and temporal to parietal areas emerged in our control model with deterministic exponential distance rule. We found that this instrength gradient reliably directed traveling waves as predicted by our hypothesis. Future research could address how the cortical geometry constrains instrength patterns, traveling wave direction, and EF gradients.

While Roberts and colleagues[27] focused on metastability of cortical waves, they also analyzed if sink and source locations were associated with instrength in a cortical model with empirical SC. They found that instrength correlated weakly with source occurrences ($r = 0.09$) but not with sinks. Their results might be due to eliminating instrength patterns by instrength-normalizing their SC. Intriguingly, they also found that rotating waves frequently emerged in their cortical network model. Supplementary Movies of our instrength-normalized cortical network model showed prominent rotating waves too. This was also the case in the 2D control network with instrength-normalized connectivity and when interacting IF and instrength gradients were balanced. We hypothesize that mechanisms underlying rotating waves take over whenever IF or instrength gradients are either weak or balanced. Our experiment on interacting IF and instrength gradients suggests that the brain could dynamically switch between different wave types by modulating IF gradients.

Exploring our cortical network model's parameters showed that traveling waves emerge across a range of global coupling scalings and conduction speeds (or time delays) within four common frequency bands. Additionally, we found that traveling waves followed the instrength gradient whenever they emerged. We speculate that instrength gradients affect traveling wave direction in concert with IF gradients across brain states.

Do empirical cortical traveling waves follow this trajectory in humans? Halgren and colleagues[9] found that large-scale alpha ECoG waves traveled from parietal and temporal to posterior and anterior sites during resting-state. Zhang and colleagues[3] also found alpha and theta ECoG waves traveled from parietal and temporal to frontal and occipital regions during a memory task. Stolk and colleagues[56] found that ECoG alpha and beta waves traveled from posterior to anterior and anterior to posterior sensorimotor cortex during a motor imagery task. Their observations align with the simulated instrength-directed traveling waves identified in our work. While we hypothesize that instrength-directed waves are most pronounced during resting-state, their impact may persist during tasks because white matter remains relatively stable over time[48,49]. In contrast, studies focusing on resting-state EEG and MEG found alpha waves traveling along the anterior-posterior axis with varying direction[70–74]. Differing traveling wave directions in ECoG compared to M/EEG might arise from signal mixing or low spatial resolution[74]. Recent studies found that large-scale fMRI infra slow waves in humans and gamma band-limited power waves in macaque[10,75] followed the principal functional connectivity gradient[20], which differs from the instrength gradient found in our study. Thus, empirical traveling waves could be directed by instrength gradients in certain frequency bands (e.g., alpha and theta) but not others (e.g., infra slow waves and gamma). Differences in traveling wave propagation direction between empirical and simulated traveling waves could also arise from differing MRI processing pipelines as outlined previously[44,67].

We further showed that instrength gradients not only direct emerging traveling waves but also generate EF patterns in cortical network models. Evidence is accumulating that smooth frequency gradients across the cortex are prevalent[3,29,31]. While investigating alpha and theta traveling waves during a memory task, Zhang and colleagues[3] found an EF gradient with higher frequencies in parietal and temporal regions and lower frequencies in anterior and posterior regions resembling the EF gradient that emerged in our cortical

network model both in structure and magnitude (compare Fig. 8c here with Fig. 2A in ref. 3). Mahjoory and colleagues[31] found a resting-state alpha EF gradient increasing from anterior to posterior sites contrasting our simulated EF gradient increasing from frontal and occipital to temporal and parietal areas. Differences between those studies could have arisen from different recording modalities (ECoG vs. MEG), processing pipelines (MEG source reconstruction and cortical parcellation), or brain states (rest vs. task).

Furthermore, Mahjoory and colleagues[31] found that EF gradients differed noticeably between theta, alpha and beta bands. We corroborated opposing alpha and beta EF gradients during resting-state MEG of HCP subjects. In contrast, our cortical network models generated similar EF gradients across frequency bands. These differences might be explained by cortical layer-specific frequency and connectivity profiles[31,66,76,77]. While the origin of alpha oscillations is still debated (infragranular[77] vs. supragranular[9] vs. network interactions[71,78]), a putative alpha-subnetwork with an instrength gradient decreasing from anterior to posterior sites and a parallel beta-subnetwork with instrength gradient increasing from anterior to posterior regions could explain EF gradients observed by Mahjoory and colleagues[31]. We showed that an unsupervised algorithm finds such putative alpha- and beta-subnetworks within the full SC network. Simulating cortical network models with these subnetworks improved the empirical and simulated EF fit markedly. We assumed that putative alpha- and beta-subnetworks act independently in our cortical network model but whether frequency-specific subnetworks interact or operate largely independently remains debated[58,79]. In a network with cross-frequency interactions, alpha oscillations could dynamically hijack the full network explaining the distinct traveling wave directions and EF gradients observed in ECoG vs. M/EEG. Our findings suggest that frequency-specific structural subnetworks could exist, but the spatial resolution of MRI currently limits more direct evidence. Advances in neuroimaging are needed to conclusively identify cortical-layer and frequency-specific subnetworks.

Previous studies proposed that traveling waves could coordinate FC networks[10,50,75]. We fit our cortical network model to resting-state MEG PLV-FC and found that the best fitting models produced instrength-directed waves corroborating that they could coordinate FC. Additionally, these cortical network models generated smooth EF gradients resembling observations from ECoG recordings during a memory task[3]. We did not model volume conduction in our cortical networks and thus, high PLV between empirical and simulated source activity could reflect true zero-lag interactions, which are physiologically meaningful[64,80]. To fully outrule volume conduction effects, we investigated non-zero-lag interactions with PLI-FC, which was largest in the alpha band and less pronounced in beta and gamma bands suggesting that instrength-directed alpha traveling waves could coordinate non-zero-lag FC. Simulated activity from putative alpha- and beta-subnetworks maintained a similar PLI-FC fit while also generating EF gradients consistent with resting-state MEG[31].

We used the tractography-derived number of streamlines as a proxy for coupling strength between brain regions. Cortical traveling wave direction and EF patterns are likely subject to additional factors such as cortical gradients of neuron density, synaptic spine count, receptor distributions, myelin content, cortical thickness, and excitation-inhibition ratio[24,81–87]. While some cortical gradients may vary on faster timescales large-scale SC gradients remain relatively stable over longer time periods such as many days, weeks or years[48,49]. Thus, SC gradients could contribute to traveling wave direction across brain states, e.g., similar traveling wave directions have been observed in the alpha band during rest and memory-tasks[3,9]. Other studies have shown that traveling wave direction changes rapidly in response to tasks[70,71,88]. We hypothesize that external or self-generated stimuli dynamically affect traveling waves by modulating IF[89]. For example, a stimulus arriving in the visual cortex could accelerate local

oscillations[90] that interact with large-scale stable coupling and IF gradients to achieve cortex-wide processing through traveling waves. Alternatively, stimuli could induce large-scale IF gradients to direct cortical traveling waves, this could be achieved through thalamocortical loops preparing the cortex for incoming stimuli. We explored this latter mechanism in our 2D network model and found that stable instrength and dynamic IF gradients could cooperatively direct traveling waves. Alternatively, inter-regional coupling strength could be dynamically controlled – for instance by modulating the long-range excitation-inhibition ratio[86]. To explain behavior, deeply understanding stable and dynamic coupling and IF gradients will be necessary.

Cortical traveling waves are ubiquitous and relate to EF gradients. We showed that a shared mechanism − namely instrength gradients − could account for both phenomena. We found that instrength gradients exist in the human connectome and affect traveling wave direction and EF gradients in cortical network models. Simulated instrength-directed traveling waves coordinated FC to fit resting-state MEG and matched experimentally observed traveling wave directions[3,9,56]. Simulated EF gradients aligned with resting-state MEG EF gradients; particularly in alpha- and beta-subnetworks found within the human connectome. While we investigated the instrength gradient mechanism on macroscale cortical networks, our findings could generalize to microscale coupling strength gradients that could direct waves emerging in spiking neural networks[28]. We proposed many hypotheses throughout our work with the potential to advance research on cortical gradients, traveling waves, and neural oscillations. In sum, our findings suggest that SC instrength gradients affect cortical traveling wave direction and shape EF gradients.

## Methods

### Experimental data
We used the publicly available data of 785 subjects that participated in the Human Connectome Project (HCP; S900 release)[42] and had complete MRI data including structural MRI (T1w and T2w), diffusion-weighted MRI and all four sessions of resting-state fMRI[86]. All participants provided informed consent as part of the HCP and the study protocol was approved by the WU-Minn HCP Consortium's institutional review boards. The processing of these data was approved by the medical ethical committee of the Charité Medical Center in Berlin. Nine subjects were excluded because of missing files that were necessary for our processing pipeline. We used the data of the remaining 776 healthy subjects (number of subjects/age range: 160/22–25, 339/26–30, 271/31–35, 6/36 + ; 432 female and 344 male; self-reported) to construct the average structural connectivity (see Estimation of structural connectivity).

We further investigated the instrength distributions in different cohorts and parcellations. We analyzed publicly available data from 70 young healthy subjects (age: 28.8 ± 9.1 years, 27 female; https://zenodo.org/record/2872624) to compute the instrengths for each region of the Lausanne atlas[65]. The Schaefer parcellation with 400 regions was estimated from 369 healthy subjects (age: 42.7 ± 17.9 years, 243 female) from the Enhanced Nathan Klein Institute Rockland Sample (http://rocklandsample.org/). The dataset is publicly available from EBRAINs (https://search.kg.ebrains.eu/instances/3f179784-194d-4795-9d8d-301b524ca00a). The random parcellation with 500 equally sized regions was estimated by Arnatkeviciute et al.[91] from 972 healthy participants of the HCP S1200 release (age: 28.7 ± 3.7, 522 female; self-reported) and is publicly available from Zenodo (https://zenodo.org/record/4733297).

Furthermore, we studied if cortical network models that produce traveling waves and frequency gradients are consistent with empirical data. For this analysis, we used magnetoencephalographic (MEG) resting-state data of 89 subjects that participated in the Human Connectome Project (see MEG resting-state preprocessing and source reconstruction)[42,92].

### Estimation of structural connectivity
The pipeline for structural connectivity (SC) estimation was based on MRtrix (https://www.mrtrix.org/) and FreeSurfer (https://surfer.nmr.mgh.harvard.edu/). We segmented the HCP subjects' structural MRI images based on tissue type (gray matter, white matter, and cerebrospinal fluid) and used the resulting segmentation for multi-shell multi-tissue constrained spherical deconvolution to estimate fiber density distributions from subjects' diffusion-weighted MRI images[86]. Next, we generated anatomically-constrained tractograms using the probabilistic iFOD2 algorithm with 25 million streamline seeds, an FOD amplitude cut-off of 0.06, and restricted fiber lengths to 250 mm. Then, we resampled the Schaefer atlas defined on fsaverage (retrieved from: https://github.com/ThomasYeoLab/CBIG/tree/master/stable_projects/brain_parcellation/Schaefer2018_LocalGlobal) to each subject's individual space and mapped this surface-based atlas to the volumetric image using FreeSurfer. Subsequently, we computed the SC weights between all regions from the SIFT2 filtered tractograms using MRtrix. Additionally, we computed the SIFT2 filtered mean streamline lengths between each Schaefer brain region. Finally, we formed the average SC across subjects using consistency-based thresholding[44]. This method preserves connections that are consistent across subjects while removing spurious ones and it reproduces the exponentially decaying distance – connection strength relationship that is frequently observed across species[45]. We specified to retain 15% of consistent connections for the group average SC. Additionally, we averaged the fiber lengths across subjects for the retained connections. Notably, we could reproduce all our findings with a group average SC thresholded at a connection density of 30%.

We further investigated the instrength distributions in different cohorts and parcellations to make sure that the finding of instrength gradients is robust. We used the publicly available SC of 70 young healthy subjects (age: 28.8 ± 9.1 years, 27 females) estimated with deterministic tractography (https://zenodo.org/record/2872624#.Y-JarOzMKDU) to compute the instrength for each region of the Lausanne atlas[65]. Hagmann and colleagues[65] detailed the MRI processing and tractography (https://zenodo.org/record/2872624#.Y-JarOzMKDU). The Schaefer parcellation with 400 regions was estimated from 369 subjects (age: 42.7 ± 17.9 years, 243 females) from the Enhanced Nathan Klein Institute Rockland Sample (eNKI) with the mean streamline count as connection strengths. Processing details and the dataset are publicly available from EBRAINs (https://search.kg.ebrains.eu/instances/3f179784-194d-4795-9d8d-301b524ca00a). The random parcellation with 500 equally sized regions was estimated by ref. 91 from 972 participants of the HCP S1200 release (age: 28.7 ± 3.7, 522 females) and is available from Zenodo (https://zenodo.org/record/4733297#.Y6w_lOzMJb8). Processing details can be found in ref. 91.

### Identification of instrength gradients in the human connectome
We calculated the instrength for each brain region by summing the incoming connection weights of the SC matrix. We identified instrength gradients across the cortex with modal spectrospatial analysis, which extends Fourier analysis to meshes with arbitrary topology[47]. First, we decomposed the Laplace-Beltrami operator to get a hundred eigenmodes and eigenvalues of a mesh that characterizes the topology between brain parcels (see Constructing a mesh for discrete operators). Next, we calculated the power spectrum by projecting the spatial instrength pattern onto the eigenmodes and normalizing by the total power. Spatial frequencies corresponding to each eigenmode were approximated by $\sqrt{\lambda}/2\pi$, where $\lambda$ are the eigenvalues[47]. We detected instrength gradients with permutation tests by randomly shuffling the instrength pattern 10,000 times and computing the corresponding spectra. For each mode, the fraction of

spectral powers that exceeded the original power corresponds to the p-value. We identified significant modes by comparing the Bonferroni-corrected p-values to a significance level of α = 0.01. We applied this analysis to group- and subject-level instrength patterns of the left hemisphere of the Schaefer parcellation and all control connectomes (eNKI cohort had average connectome only; see Supplementary Fig. 3).

## 2D Kuramoto network model

We studied if traveling waves emerge and follow an instrength gradient in a 30 × 30 network model of Kuramoto oscillators assuming a side length of 140 mm and unconstrained boundaries. The standard Kuramoto model[36] was modified to include distance-dependent connectivity and conduction delays:

$$\frac{d\theta_i}{dt} = \omega_i + \frac{K}{N}\sum_{j=1}^{N} a_{ij}\sin(\theta_j(t - \tau_{ij}) - \theta_i(t)) \tag{1}$$

Where $\theta_i$ is the phase of oscillator $i$, $t$ is time, $\omega$ is the intrinsic oscillation frequency (IF), $K$ the global coupling scaling, $N$ the number of oscillators, $a_{ij}$ the connection strength between oscillators $i$ and $j$, and $\tau_{ij}$ is the delay between oscillators $i$ and $j$. We used exponentially decaying connection strengths

$$a_{ij} = \frac{1}{2\sigma}e^{-\frac{d_{ij}}{\sigma}} \tag{2}$$

Where $\sigma = 10\,mm$ is the length scale and $d_{ij}$ is the euclidean distance from oscillator $i$ to $j$. Additionally, we created a distance-dependent connection probability by connecting two oscillators if their euclidean distance was smaller than a random sample drawn from an exponential distribution. We parameterized its probability density function to achieve a network connectivity of approximately 10% (scale parameter = 17 mm). Next, we removed self-connections and normalized all connections to have uniform instrength across the network. We imposed an instrength gradient by weighting the connection strengths with 2D-gaussians placed in the upper right and lower left of the network, respectively. Each gaussian was defined by

$$f_j(\mathbf{x_i}) = \frac{1}{\sqrt{(2\pi)^2 \det(\boldsymbol{\Sigma})}}\exp\left(-\frac{1}{2}\left(\mathbf{x_i} - \boldsymbol{\mu_j}\right)^T \boldsymbol{\Sigma}^{-1}\left(\mathbf{x_i} - \boldsymbol{\mu_j}\right)\right) \tag{3}$$

Where $\mathbf{x_i} \in \mathbb{R}^2$ are the coordinates for oscillator $i$, $\boldsymbol{\mu_j} \in \mathbb{R}^2$ is the center of the $j$ th gaussian, and $\boldsymbol{\Sigma} \in \mathbb{R}^{2\times2}$ is the covariance matrix. We chose

$$\boldsymbol{\mu_1} = \begin{bmatrix} 40 \\ 40 \end{bmatrix}, \boldsymbol{\mu_2} = \begin{bmatrix} 100 \\ 100 \end{bmatrix}, \boldsymbol{\Sigma} = \begin{bmatrix} 300 & 0 \\ 0 & 300 \end{bmatrix} \tag{4}$$

Resulting in appropriate positioning and width of the gaussians for the network size. By min-max normalizing the difference $f_1(\boldsymbol{x_i}) - f_2(\boldsymbol{x_i})$ to range between -1 and 1 we got a gradient template g($\mathbf{x_i}$) that we used for all 2D network models. The final connection strengths $c_{ij}$ that sum up to the desired instrength were calculated as

$$c_{ij} = a_{ij}(\alpha g(\mathbf{x_i}) + \beta) \tag{5}$$

where $\alpha = 2$ is a scaling factor and $\beta = 4$ is an offset that we chose heuristically to ensure that waves emerge and follow the instrength gradient reliably and that all $c_{ij} \geq 0$. This instrength gradient resulted in a sink at $\boldsymbol{\mu_1}$ and a source at $\boldsymbol{\mu_2}$. We calculated the delays between oscillators $i$ and $j$ as

$$\tau_{ij} = \frac{d_{ij}}{v} \tag{6}$$

Where $v$ is the conduction speed and $d_{ij}$ the euclidean distance between $i$ and $j$. We chose a biologically realistic conduction speed of 3 m/s[39–41].

The control model was created in the same way except that the connection strengths were scaled with the average instrength of the gradient model, thereby creating a uniform instrength distribution that allows the emergence of traveling waves without directional bias. However, the oscillators settled in a synchronized state and thus, we reduced the global coupling scaling ($K_{gradient} = 10$ and $K_{uniform} = 0.3$) to achieve a similar synchronization to the gradient model making the network activity comparable.

We set all oscillators to an IF of 10 Hz, initialized the phases uniform randomly and simulated the model 100 times for 11 s with a 4th order Runge-Kutta algorithm with an integration time step of 1 ms in The Virtual Brain[93]. We removed the first second of all simulations taking into account the transition time from random initial conditions to traveling waves.

## 2D Kuramoto network model with instrength and intrinsic frequency gradients

We investigated if instrength and IF gradients interact to direct traveling waves. To do so, we built on the 2D instrength gradient model introduced earlier. We added an IF gradient generated from the same gradient template g($\mathbf{x_i}$) that we used to construct the instrength gradient (see Fig. 3a). We chose to scale the IF gradient from 0 to 1.5 (IF ranges from 8.5 to 11.5 Hz at maximum scaling) in steps of 0.05 (IF range increment of 0.1 Hz) superimposed on a fixed instrength gradient and simulated 100 randomly initialized models per scaling factor. We investigated gradient template – flow potential Spearman correlation for all scaling factors. In this model, a negative correlation indicates instrength-directed waves, while a positive correlation indicates IF-directed waves. We also investigated the gradient template – EF Spearman correlation, where a negative correlation indicates that EF patterns are shaped by the instrength gradient, while a positive correlation means that they are shaped by the IF gradient.

## Cortical network model

We built a cortical network model with 1000 Schaefer atlas regions[43]. Each brain region was represented by the Kuramoto model introduced earlier and connected by setting $a_{ij}$ to the tractography-estimated connection strengths (see Estimation of structural connectivity). We used an IF $\omega_i$ of 10 Hz, a conduction speed $v$ of 3 m/s, and a coupling scaling $K$ of 0.01. We calculated the time delays between cortical regions by dividing the empirically estimated fiber lengths by the conduction speed $v$. This also results in distance-dependent conduction delays because fiber length and euclidean distance are strongly correlated ($r = 0.8$, $p < 0.01$; see Fig. 4b, c and Supplementary Fig. 5). The model was simulated 100 times with random uniform initial phases for 11 s using the 4th order Runge-Kutta algorithm with an integration time step of 1 ms in The Virtual Brain[93]. We removed one second of transient activity at the beginning of all simulations.

## Control model with randomly shuffled connection strengths

We randomly shuffled the empirically estimated connection strengths within existing connections. This model preserved the network topology and conduction delays while destroying the fiber length – connection strength relationship as well as the instrength gradient. Notably, random shuffling of the connection strengths also destroys the euclidean distance – connection strength relationship. We verified this by fitting the euclidean distance – shuffled connection strength relationship with a linear, exponential, and power law model (coefficients of determination: $r^2_{linear} < 0.001$, $r^2_{\exp} < 0.001$, $r^2_{power} < 0.001$; see next section for further details). At the original coupling scaling of 0.01 all cortical regions synchronized fully, so we adjusted the global

coupling scaling ($K = 0.003$) to achieve a synchronization similar to the original model.

## Control model with preserved distance - connection strength relationship

We first computed the position of the 1000 Schaefer regions[43] using the centers of mass of the FreeSurfer fsaverage5 inflated cortical surface (https://github.com/ThomasYeoLab/CBIG/tree/master/stable_projects/brain_parcellation/schaefer2018_LocalGlobal). Next, we calculated the euclidean distance based on these positions for connections that existed in the SC (left hemisphere only). Visualizing the euclidean distance – connection strength relationship suggested that connection strength drops off with distance following an exponential or power law (see Supplementary Fig. 5b).

Accordingly, we fit an exponential and a power law model to the euclidean distance – connection strength relationship using non-linear least squares. We found the exponential model to be a better fit ($r^2_{exp} = 0.37$ vs. $r^2_{power} = 0.34$):

$$\widehat{a_{ij}} = \alpha e^{-\lambda \delta_{ij}} \tag{7}$$

Where $\hat{a}_{ij}$ is the estimated connection strength, $\delta_{ij}$ is the euclidean distance between brain regions $i$ and $j$, $\alpha$ is a scaling constant, and $\lambda$ is the decay rate. The free parameters were estimated to be $\alpha = 839.11$ and $\lambda = 0.08\ mm^{-1}$. Subsequently, we used this exponential model to synthesize a surrogate SC for the left and right hemispheres (interhemispheric connections were retained from the original SC) that preserves the distance – connection strength relationship but destroys the instrength gradients. We computed the corresponding conduction delays by dividing the euclidean distance with a conduction speed of 3 m/s.

## Control model with zero conduction delays

For this control model, we set the conduction delays to zero to investigate their role in the emergence and guidance of traveling waves. This control model preserves the tractography-derived SC including connection topology and strengths, as well as the instrength gradients. We adjusted the global coupling scaling ($K = 0.0001$) to achieve a similar synchronization to the original cortical network model and prevent full synchronization.

## Control model with instrength-normalized structural connectivity

We tested if hub-structure could explain the traveling wave direction observed in our cortical network model by instrength-normalizing the SC. To do so, we divided the SC matrix row-wise by the region's instrength and multiplied it by the mean instrength of the original SC. Everything else remained equal to the original cortical network model.

## Control model with Jansen-Rit neural masses

We tested if the SC allows the emergence of traveling waves in a more realistic cortical network model consisting of Jansen-Rit neural masses. We further investigated if traveling wave direction and oscillation frequency are modulated by the instrength gradients.

In this model, each region consisted of populations of excitatory pyramidal neurons as well as inhibitory and excitatory interneurons. The pyramidal neuron population excites the interneuron populations and receives inhibitory and excitatory feedback in turn. We interpreted the postsynaptic activity at the pyramidal population as the local field potential of the region. This local field potential is the difference of postsynaptic potentials induced by the excitatory and inhibitory interneuron populations.

We set the parameters of the model such that ~10 $Hz$ oscillations emerged in the connected cortical network (all parameters are shown in Supplementary Table 1). We simulated the model 100

times for 12 s with a 4th order Runge-Kutta algorithm with an integration time step of 1 ms in The Virtual Brain[93]. For further analyses, we bandpass filtered the local field potentials between 5 and 15 Hz (8th-order forward-backward Butterworth filter using cascaded second-order sections). We removed the first and last seconds of all simulations to remove simulation transients and filtering edge effects.

## Parameter exploration of cortical network model

For the parameter exploration of the cortical network model, we used intrinsic oscillation frequencies $\omega_i \in \{1,10,20,40\}$ Hz, conduction speeds $v$ from 1 to 10 m/s in increments of 1 m/s, and global coupling scalings $K$ from $10^{-5}$ to 10 spaced logarithmically with 39 steps. We ran 10 simulations for each parameter combination with random uniform initial phases for 11 s using the 4th order Runge-Kutta algorithm with an integration time step of 1 ms in The Virtual Brain[93]. We removed one second of transient activity at the beginning of all simulations.

## MEG resting-state preprocessing and source reconstruction

Our preprocessing steps roughly followed the Human Connectome Project MEG processing pipeline[92]. We used the MNE-Python software for processing these data[94]. We used the first resting-state session of 89 HCP MEG subjects for our analyses. First, we regressed out the MEG reference channels followed by removing bad channels and segments (provided with the HCP data). Then, we bandpass filtered the MEG signals between 1.3 and 150 Hz (8th-order forward-backward Butterworth infinite impulse response filter) and removed line noise with a zero-phase notch filter at 60 and 120 Hz (zero-phase finite impulse response filter; Hamming window with 0.0194 passband ripple and 53 dB stopband attenuation; lower transition bandwidth: 0.50 Hz; upper transition bandwidth: 0.50 Hz). Next, we used the ICA components provided with the HCP data to remove EOG, ECG, and other artifacts. Finally, we extracted the longest continuous artifact-free segment of resting-state activity per subject with a minimum duration of 80 s to ensure uninterrupted data streams. After excluding nine subjects that did not have more than 80 s of continuous artifact-free recordings, we were left with a total of 80 subjects (number of subjects/age range: 15/22−25, 33/26−30, 32/31−35; 40 females and 40 males). We removed five seconds at the beginning and end of these segments to avoid filter edge effects. We applied the same processing to the empty room recordings but used the full length of the data. To reduce the computational burden, we resampled the recordings to a frequency of 100 Hz after which we removed another five seconds at each end to avoid filter edge effects.

To prepare for source reconstruction, we extracted the head model and the MRI-MEG coregistration transformation matrix from the anatomical data of the subjects using the HCP MNE toolbox (https://mne.tools/mne-hcp). Next, we set up a source model by decimating the FreeSurfer fsaverage gray-white matter surface using recursively subdivided octahedrons, resulting in 8196 freely oriented sources. The sources were then morphed into the subject's native space. Subsequently, we constructed a single layer inner skull BEM model with a conductivity of 0.3 S/m followed by computing the forward model. Furthermore, we estimated the data and noise covariances from the respective recordings. We used the previous results to compute the linearly constrained minimum variance spatial filters for the dipole orientation that maximizes power (regularization constant of 0.05 and unit-noise gain normalization). Then, we aggregated the activities by averaging all source time series within each parcel of the Schaefer atlas. Lastly, we bandpass filtered the resulting source activities within the respective frequency bands (alpha: 5−15 Hz, beta: 15−25, gamma: 35−45; 4th-order forward-backward Butterworth filter using cascaded second-order sections).

## Fitting cortical network models to resting-state functional connectivity

We studied if cortical network models expressing instrength-directed traveling waves and EF gradients are consistent with MEG-derived resting-state FC.

We estimated the FC from the resulting source activities for each frequency band separately. We chose the phase locking value (PLV)[51] as a measure for FC because it is frequently used and our cortical network model simulates instantaneous phases. The PLV between regions $k$ and $l$ is defined as

$$\text{PLV}_{kl} = \frac{1}{N}\left|\sum_{n=0}^{N} e^{j(\theta_k(n)-\theta_l(n))}\right| \tag{8}$$

Where $N$ is the number of time points, $\theta_k(n)$ is the instantaneous phase of region $k$ at time point $n$, and $|\cdot|$ the magnitude operator. Thus, the PLV captures consistent phase differences between pairs of regions across time. The PLV is 1 if the regions show consistent phase differences and 0 if their activity is completely incoherent.

We further used the phase lag index (PLI)[52], which is insensitive to zero-phase lag interactions and thus also to possible volume conduction. PLI quantifies the asymmetry of the phase difference distribution between signals. It is 1 if the two regions have consistent non-zero phase differences and 0 if they have zero or random phase differences. The PLI between regions $k$ and $l$ is defined as

$$\text{PLI}_{kl} = \left|\frac{1}{N}\sum_{n=0}^{N}\text{sign}\left[\text{Im}\left(e^{j(\theta_k(n)-\theta_l(n))}\right)\right]\right| \tag{9}$$

Where $\text{sign}[\cdot]$ is the sign-function and $\text{Im}(\cdot)$ is the imaginary part.

To construct the resting-state FC, we estimated the PLV and PLI for all pairs of regions. We used the same method to estimate the simulated FC for all cortical network models of the parameter exploration (see Parameter exploration of cortical network model) in the alpha, beta, and gamma frequency bands. We set the diagonal elements of the FC matrix to zero. Finally, we used the Pearson correlation coefficient to assess the fit between simulated and empirical FC for each parameter combination.

## Fitting cortical network models to resting-state MEG effective frequency

We fit cortical network models to resting-state MEG EF. To do so, we computed the empirical EFs within each frequency band extracted from the MEG source reconstruction and averaged them across subjects (see Effective frequency estimation). We further computed the average EF maps across simulations for various parameter combinations. We explored the agreement between simulated and empirical EF using the concordance correlation coefficient (CCC)

$$\rho_c = \frac{2\rho\sigma_{sim}\sigma_{emp}}{\sigma_{sim}^2 + \sigma_{emp}^2 + \left(\mu_{sim}-\mu_{emp}\right)^2} \tag{10}$$

where $\rho$ is the Pearson correlation coefficient between the simulated and empirical EF, $\sigma_\cdot$ is the simulated or empirical EF standard deviation, and $\mu_\cdot$ is the simulated or empirical EF mean. CCC is a metric of agreement between two measurements and is 1 if the measures agree perfectly, -1 if they disagree perfectly, and 0 if they do not agree beyond chance. We chose CCC because it not only assesses if simulated and empirical EFs are correlated but also if they are of similar magnitude.

## Decomposing the structural connectivity into subnetworks

Tractography is indifferent to subnetworks underlying the total SC. Nonnegative matrix factorization[59] (NMF) – an unsupervised

decomposition algorithm – allowed us to identify common subnetworks across the HCP subjects total SCs. NMF decomposes a non-negative matrix $\mathbf{V}$ into two low-rank nonnegative matrices $\mathbf{W}$ and $\mathbf{H}$

$$\mathbf{V} \approx \mathbf{WH} \tag{11}$$

$\mathbf{V}$ is a $n \times m$ matrix, where $n$ is the number of features and $m$ is the number of samples. $\mathbf{W}$ and $\mathbf{H}$ are $n \times r$ and $r \times m$ matrices, where $r$ is the number of components. The components of $\mathbf{W}$ are feature patterns common across the samples, while the coefficients in $\mathbf{H}$ express how much each feature pattern contributes to each sample.

We built a data matrix $\mathbf{V}$ where each sample was the upper triangular matrix of an individual subject's SC flattened into a column vector. Then, we removed all connections across subjects that did not survive consistency-thresholding for the average SC (see Estimation of structural connectivity), which resulted in a $75000 \times 776$ nonnegative data matrix $\mathbf{V}$. Notably, using the full upper triangular matrices of all subjects' SCs ($\mathbf{V}$ with dimensions $499500 \times 776$) resulted in similar subnetworks (see Supplementary Fig. 8b). We used the reduced data matrix in our simulations and analyses to speed up computations. We determined the number of components using a cross-validation method that first randomly sets 5% of the datapoints to zero, then computes the NMF, which is eventually used to impute the missing data[60]. The mean squared error between the imputed datapoints and the original held-out values was computed for NMFs with 2 to 10 components. We repeated this procedure 100 times because the NMF low-rank matrices require random initialization resulting in different outcomes across runs. We found that 5 components minimized the average mean squared error across all 100 runs. Finally, we used the 5-rank NMF with the lowest mean squared error across 100 random initializations. We calculated the average subnetwork SCs by reconstructing $\mathbf{V}$ from each component, followed by averaging across subjects. We used the RcppML package for the NMF and cross-validation[61].

We identified putative alpha- and beta-subnetworks by finding the subnetworks whose instrengths correlated most strongly with the anterior-posterior position. The subnetwork with instrength decreasing along the anterior-posterior axis was chosen to be the putative alpha-subnetwork, and the one with increasing instrength the putative beta-subnetwork. These networks were used in the parameter exploration presented in the main text. The cross-validation revealed that NMFs with fewer than 5 subnetworks had similar reconstruction errors compared to the best model. Thus, we wondered if these NMFs also identified similar subnetworks. To test this, we first computed the pairwise Pearson correlation between all feature patterns of the rank-2 to 5 NMFs (total of 14 subnetworks). Hierarchical clustering of the resulting correlation matrix using the Ward algorithm, revealed two high-level clusters. The instrength patterns of subnetworks belonging to one of the two clusters showed that the NMFs consistently found alpha- and beta-subnetworks.

## Detecting traveling waves

We assumed that traveling waves propagate from sources to sinks[27] and wave detection is equivalent to identifying these sources and sinks. To do this, we started with the instantaneous phases from our network models (we estimated the instantaneous phases using the complex argument of the analytic signal obtained through the Hilbert transform for the Jansen-Rit model) and quantified how phase changes across space by computing the spatial phase gradient on a mesh derived from the positions of the 2D network or from the 1000 Schaefer brain regions (see Computing spatial phase gradients). Then, we defined a 3-ring neighborhood around a region and calculated its idealized and normalized gradient based on the region's geodesic distance. This local idealized gradient describes a wave expanding from a source. We calculated the average angular similarity between this idealized gradient and the empirically derived normalized

negative phase gradient (wave propagation direction) to measure their alignment within a region's neighborhood:

$$\text{angular similarity} = 1 - 2\alpha/\pi \tag{12}$$

where $\alpha$ is the angle between the idealized and empirical phase gradients. An angular similarity of 1 indicates full alignment, 0 orthogonality, and -1 full alignment with opposite directions. Notably, the negative phase gradient points into the direction of wave propagation. Thus, if the average angular similarity in the neighborhood around a region is negative, the phase gradients are consistent with a wave sink and if it is positive, they are consistent with a wave source.

Finally, we used permutation testing to assess the statistical significance of sources and sinks. Here, we randomly shuffled the data 1000 times across space (instantaneous phases for Kuramoto models and local field potentials for Jansen-Rit model), repeated the processing above, and computed a null distribution of global maxima found from the absolute values of the angular similarities of all regions. Then, we determined the $p$-value by finding the proportion of shuffled angular similarity maxima that were at least as extreme as the absolute value of the original angular similarity of each region. Using the global maximum of our statistic accounts for the multiple testing problem. Lastly, we identified significant sources and sinks if $p < \alpha$, with a significance level of $\alpha = 0.01$ for the 2D network model and the cortical network model. To keep the computational load manageable for the parameter exploration, we reduced the number of random permutations across space to 100 and adjusted the significance level to $\alpha = 0.05$. Additionally, we downsampled all simulated time series by a factor of five before applying this method.

### Computing the wave flow potential
We visualized how waves propagated across our network models and quantified their relationship with instrength gradients with the wave flow potential, a scalar field defined on our network models that can be interpreted as a landscape in which waves flow from peaks (sources) to valleys (sinks). To compute the wave flow potential, we decomposed the spatial phase gradient (see Computation of spatial phase gradient) using the Helmholtz-Hodge Decomposition for vector fields defined on 2D manifolds embedded in 3D[95]:

$$\boldsymbol{\xi} = \nabla \mathbf{D} + \mathbf{J}\nabla \mathbf{R} \tag{13}$$

where $\boldsymbol{\xi}$ is a vector field (spatial phase gradients), $\nabla$ is the gradient operator, $\mathbf{D}$ and $\mathbf{R}$ are curl-free and divergence-free potentials, and $\mathbf{J}$ is a rotation $\mathbf{J}v = (-v_2, v_1)$ with $\mathbf{v} = (v_1, v_2)$. In our specific case the curl-free potential $\mathbf{D}$ is the wave flow potential, obtained by solving the Poisson equation:

$$\Delta \mathbf{D} = \nabla \cdot \boldsymbol{\xi} \tag{14}$$

where $\boldsymbol{\xi}$ is the spatial phase gradient, $\Delta$ the Laplacian and $\nabla\cdot$ the divergence operator. We solved this equation with the sparse Cholesky decomposition using the scikit-sparse Python package (https://github.com/scikit-sparse/scikit-sparse) and the corresponding discrete operators (see Computing spatial phase gradients). The interpretation of the resulting wave flow potential $\mathbf{D}$ is that waves travel from higher to lower potentials. We applied this analysis to the left and right hemispheres separately. Additionally, we downsampled all simulated time series by a factor of five before applying this method.

### Detection of instrength-directed traveling waves
We detected instrength-directed traveling waves by finding time points where the instrength gradient correlated significantly with the flow potential (see Computing the wave flow potential). To do so, we established a null distribution of Spearman correlations using 1000

random spin permutations[90] (100 for parameter explorations) of the original data and repeated the analysis described in the section Computing the wave flow potential. The p-value is the proportion of permutations that were at least as extreme as the original correlation. We detected significant instrength-directed waves by $p < 0.01$ ($p < 0.05$ for parameter exploration). This analysis was conducted separately for the left and right hemispheres and we downsampled all simulated time series by a factor of five before applying this method.

### Computing spatial phase gradients
We computed the spatial phase gradients from the instantaneous phases $\theta_i$ of our network models using libigl's discrete gradient operator (https://libigl.github.io/) that works on a triangle mesh (see Constructing a mesh for discrete operators). We handled phase unwrapping similar to ref. 27 by applying the discrete gradient operator to $e^{j\theta_i}$ ($j$ is the imaginary unit) which results in complex gradients defined on the mesh triangles. Then, we multiplied these complex gradients by $-je^{-j\phi_k}$ where $\phi_k$ is the barycentric interpolation of $e^{j\theta_i}$ to triangle $k$ for all $i$ defining triangle $k$. Finally, the real part of this result is the phase gradient defined on mesh triangles. We applied the discrete gradient operators on the left and right hemispheres separately.

### Constructing a mesh for discrete operators
The discrete operators that we used to estimate the spatial phase gradient, Laplacian, and divergence, operate on triangle meshes defined by vertices and faces (triplets of vertices). For the 2D network model, we defined the oscillator positions as mesh vertices and created the corresponding triangle faces with Delaunay triangulation. For the cortical network model, we started with the inflated FreeSurfer fsaverage mesh (https://surfer.nmr.mgh.harvard.edu/) and defined mesh vertices as the centers of mass of all vertices belonging to a Schaefer region. Furthermore, we preserved the neighborhood topology between regions to define the mesh triangle faces. We created these meshes for each cortical hemisphere separately.

### Effective frequency estimation
We quantified EF by calculating the instantaneous frequency from the analytic signal:

$$f_{inst} = \arg\left(s_a(n)\bar{s}_a(n-1)\right) \tag{15}$$

where $s_a(n) \in \mathbb{C}$ is the analytic signal at the discrete sample $n$, $\bar{s}_a$ is the complex conjugate of the analytic signal, and $\arg(\cdot)$ is the complex argument. We used the exponential form of the analytic signal $s_a(n) = e^{j\theta(n)}$ for the Kuramoto network models. For estimating EF from resting-state MEG, we computed the analytic signal by Hilbert transforming the bandpass filtered source signals. We calculated the EF maps for each simulation by taking the median instantaneous frequency across time. Average EF maps were calculated by taking the mean of the median instantaneous frequency maps across simulations or subjects.

### Spin permutation tests
We assessed the spatial relationship between various 2D and cortical maps (e.g., instrength, EF, flow potential) with Spearman correlation. We identified statistical differences using permutation tests with 10,000 random rotations of the original data across space (brainspace.readthedocs.io)[96]. Spin permutation tests preserve most of the original data's features including spatial autocorrelation; Hence, they are suitable for assessing spatial correspondence between maps[97]. We devised a similar strategy for the 2D network model statistics: we randomly translate 2D maps along both dimensions followed by randomly rotating them. Missing values were replaced by border

reflection. The proportion of permutations that resulted in a correlation at least as extreme as the original correlation determines the p-value. The correlation was deemed significant if $p < 0.01$.

## Reporting summary

Further information on research design is available in the Nature Portfolio Reporting Summary linked to this article.

## Data availability

Human Connectome Project data are publicly available at https://db.humanconnectome.org. The Schaefer atlas parcellation with 1000 regions is publicly available at https://github.com/ThomasYeoLab/CBIG/tree/master/stable_projects/brain_parcellation/Schaefer2018_LocalGlobal. The structural connectivity based on the Lausanne atlas is publicly available at https://zenodo.org/record/2872624. The structural connectivity based on the 400 region Schaefer atlas is publicly available at https://search.kg.ebrains.eu/instances/3f179784-194d-4795-9d8d-301b524ca00a. The structural connectivity based on a random parcellation with 500 regions is publicly available at https://zenodo.org/record/4733297. Source data are provided with this paper.

## Code availability

All custom code used in this study is freely available at https://osf.io/daq54. Custom code was written in Python version 3.8.12 using multiple packages (The Virtual Brain 2.3, scipy 1.9.0, numpy 1.23.1, matplotlib 3.7.2, pyvista 0.41.1, LibIGL-Python-bindings 2.2.1, MNE-Python 1.0.3, MNE-HCP 0.1.dev12, BrainSpace 0.1.4, scikit-sparse 0.4.6) and R programming language version 4.3.1 using multiple packages (RcppML 0.5.5, reticulate 1.34.0). FreeSurfer 7.1.1 and 7.1.0, MRtrix 3.0 and 3.0.2, and FSL 6.0 were used for MRI processing.

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

## Acknowledgements

Data were provided [in part] by the Human Connectome Project, WU-Minn Consortium (Principal Investigators: David Van Essen and Kamil Ugurbil; 1U54MH091657) funded by the 16 NIH Institutes and Centers that support the NIH Blueprint for Neuroscience Research and by the McDonnell Center for Systems Neuroscience at Washington University. P.R. acknowledges funding from the following sources: Digital Europe Grant TEF-Health # 101100700, H2020 Research and Innovation Action Grant Human Brain Project SGA2 785907, H2020 Research and Innovation Action Grant Human Brain Project SGA3 945539, H2020 Research and Innovation Action Grant EOSC VirtualBrainCloud 826421, H2020 European Innovation Council PHRASE 101058240, H2020 European Research Council Grant ERC BrainModes 683049, JPND ERA PerMed PatternCog 2522FSB904, Berlin Institute of Health & Foundation Charité, Johanna Quandt Excellence Initiative, German Research Foundation SFB 1436 (project ID 425899996), German Research Foundation SFB 1315 (project ID 327654276), German Research Foundation SFB 936 (project ID 178316478), German Research Foundation SFB-TRR 295 (project ID 424778381) German Research Foundation SPP Computational Connectomics RI 2073/6-1, RI 2073/10-2, RI 2073/9-1, DFG Clinical Research Group BECAUSE-Y 504745852; Horizon Europe: EBRAINS 2.0 101147319, Virtual Brain Twin 101137289, Research and Innovation Action Grant AISN 101057655, Research Infrastructures Grant EBRAINS-PREP 101079717, Research Infrastructures Grant EBRAIN-Health 101058516.

## Author contributions

Conceptualization, D.P.K. and P.R.; Methodology, D.P.K. and M.S.; Software, D.P.K. and M.S.; Validation, D.P.K.; Formal Analysis, D.P.K.; Investigation, D.P.K.; Resources, P.R.; Writing – Original Draft, D.P.K.; Writing – Review & Editing, D.P.K., M.S. and P.R.; Visualization, D.P.K.; Supervision, P.R.; Project Administration, P.R.; Funding Acquisition, P.R.

## Funding

## Competing interests

The authors declare no competing interests.
