## [Peer Review File · Nature Communications]

Human connectome topology directs cortical traveling waves and shapes frequency gradientsReviewer #1 (Remarks to the Author):

This study shows computational simulations on instrength-gradients hypothesis for generating cortical traveling waves with frequency gradients. All simulations appear properly implemented and the results also appear reasonable. However, unfortunately, the current results are not sufficient for proving the instrength-gradient hypothesis in terms of the following points. (1) The instrength-gradient hypothesis does not reject the frequency-gradient hypothesis proposed in the early study (Ermentrout and Kopell, 1984) and it rather appears a kind of paraphrase of the frequency-gradient hypothesis. (2) The instrength-gradient hypothesis does not agree with the experimental evidence showing frequency-band-dependent directionality of the frequency gradient (Mahjoory et al., 2020). Please revise the manuscript to show clear proof of the instrength gradient hypothesis in terms of the theoretical mechanism and the explainability of the pre-existing experimental evidence.

Major points

(1a) "Instrength" or "intrinsic frequency"?

Early theoretical study has shown that connected oscillators with intrinsic-frequency gradients result in the formation of traveling waves (Ermentrout and Kopell, 1984) and it is also discussed as a key mechanism generating the cortical traveling waves (Zhang, et al., 2018). The current results did not evaluate the influence of the intrinsic frequency gradient, but such estimation would be technically possible in Kuramoto model where the connection strength (a_{ij}) and intrinsic frequency (ω_i) are separately described. It is encouraged to clarify what type of spatial gradient would be more likely to explain the cortical traveling waves.

(1b) Additional control model using the normalized connection weight ("Instrength" or "Hub-structure"?)

In the current manuscript, there is no control model using the normalization of connection weights (i.e., connection from oscillator j to i , a_{ij} , is replaced to $a_{ij}/\sum_i(a_{ij})$), but it has been often used in many connectome-based models (c.f., Roberts, et al., 2019). To clarify the importance of instrength but not to connection pattern (i.e., Hub-structure) on the traveling waves, it is necessary to include this type of control model. This point is also associated with the comment (1a).

(1c) Fluctuation of traveling waves?

Fluctuations in the cortical traveling waves have been thought to be one of the fundamental properties for the understanding of the phenomena and its functional importance, and many theoretical models have also focused on it (Roberts, et al., 2019). The current manuscript does not evaluate these properties, but it is still important to show that the instrength-gradient hypothesis leads similar (or different) properties of the fluctuation with those in the pre-existing hypothesis and the experimental evidence. It would be one fundamental support of the current hypothesis. Please consider including these discussions in the revised manuscript.

(2) Frequency-band-dependent directionality of the frequency gradient

Experimental evidence have shown that the cortical traveling wave dominantly appear in theta-alpha-band oscillation (Zhang, et al., 2018) and that the directions of peak-frequency-gradient are highly dependent on their frequency band (i.e., Alpha-band gradient direction is opposite to beta-band gradient direction) (Mahjoory et al., 2020). As shown in Figure 5, the current hypothesis does not agree with these evidence; i.e., the current mechanism principally produce cortical waves independent to their frequency band (when the conduction speed is around 7 m/s (which is still within a physiologically reasonable value)). A short, interesting discussion on cortical layer-specific structural connectivity is described (line 480-), however, it does not appear convincingly; i.e., the layer-specific oscillation in the experiments does not seem to be organized with gradual frequency-dependent profiles (i.e., Halgren, et al.(2019) showed dominantly alpha oscillation the layers. Bastos, et al. (2018) showed that alpha/beta band is dominant in deep layers and gamma-band is dominant in superficial layer). Please solve the disagreements of the experimental evidence with the current hypothesis and results.

Minor points

(1) The current manuscript does not clearly describe why the understanding of the neural

mechanism generating traveling waves is important. Please add some descriptions on previous arguments or its necessity on functional point of view.

(2) The section on the 2-D network model appears trivial to me from the given Kuramoto model where the phase of oscillators receiving stronger connection weights results in earlier phase.

Please revise the manuscript to show the novel points of these results or to move them to Supplemental information.

(3) What is the boundary constraints in the 2D network model?

(4) Please describe the absolute values (and their distribution) of instrength in the connectome-based model and discuss their associates with experimental evidence (on cortical thickness, neuron density, etc.). This would be a support of the instrength gradient hypothesis.

Reviewer #2 (Remarks to the Author):

This study proposes a complementary mechanism for the generation of neural traveling waves using gradients of connection strength between neural networks, which is a different framework compared to frequency gradients suggested by other work. The researchers tested their hypothesis using a 2D oscillator model and human connectome data, which they described as indicating that the brain's connectivity shows patterns of gradients consistent with their model. Although there are some interesting ideas in this paper, I do not find this contribution to be compelling because the paper overstates the novelty of their model, inadequately describes the existing literature, and does not fully consider alternative hypotheses.

Main comments:

The paper claims that in their simulations traveling waves only emerges in models where there is a high "instrength" gradient—I think this is quite misleading! Although the text Figure 1B claims there are no any stable traveling waves in the uniform instrength simulation, however the spatio-temporal patterns generated in this situation indeed look very much like actual electrophysiology recordings that is reported by earlier studies! There seem to be spatiotemporal wave propagation patterns that resemble spatially complex patterns of traveling waves. I think the authors really mean to state that these patterns are not linear plane waves... But nonetheless the demonstration for some type of traveling wave here seems to be important because it contradicts some of the author's claims. Relatedly, the paper should provide a clear quantitative definition for measuring traveling waves and identifying their robustness.

The paper claims that gradients of connectivity give rise to traveling waves. But the paper does not even show statistically that these gradients exist in individual subjects' data. To address this concern, the authors should compare pairs of regions and demonstrate the significance of gradient differences, particularly for longer distances between the regions. It is crucial to show the existence of this gradient at the single-subject level as well, considering the presence of traveling waves in individual subjects.

I think the paper overstates their claims of novelty regarding their modeling. It is known in physics that traveling waves are a general form of solution for Kuramoto models. Previous studies have shown that traveling waves can occur even in the absence of instrength gradients, which makes me concerned about the authors' claim that traveling waves specifically emerge in Kuramoto models with non-uniform instrength spatial distribution. For example, see Iatsenko et al., 2013. At the very least, the authors should emphasize why the emergence of traveling waves from a 2D Kuramoto model is surprising or distinct from previous findings.

I was confused by the author's claims on line 67 that there were no observations of frequency gradients in earlier papers. The authors themselves cited at least two papers mentioning such ideas (Zhang et al and Mahjoury et al). Doesn't this undermine the entire motivation for this project?

The paper is hard to understand because many key terms are not explained or not used in

standard ways, beginning in the abstract. First, the term "structural connectivity gradient" used in line 15 is unclear and requires clarification—a precise definition should be provided. Additionally, the abstract contains redundancy and lacks a clear statement regarding the problem addressed or the novelty of the study. Furthermore, the use of the term "in-strength" is problematic. Is this presumed to refer to the weighted in-degree, considering the synaptic strengths, rather than just the counts? However, this assumption should be explicitly explained and defined to avoid ambiguity. Overall, the abstract needs improvement in terms of clarity, conciseness, and providing a more compelling overview of the research.

The overlap of the paper's findings in relation to the study by Breakspear's group in Nature Communications is questionable. The discussion paragraph (line 452) fails to elucidate how the current results contribute to a better understanding or provide additional insights beyond that work. The paper needs to explain how this paper goes beyond this paper and others)—such clarification, the significance of the research remains uncertain.

Minor comments:

[116] This parts of the text discussing 100-100% ranges are confusing. Is there never any noise in these models? These 100-100% ranges are not described in a conventional way and seem to be biologically implausible.

[191]: What does the word "should" mean here? There should be a citation to support this point, and in any case this type of claim is not appropriate for a caption. There are lots of situations where traveling waves might not emerge even with distant dependent connectivity and I think getting the details right is important here.

[Figure 2]: The average instrength plot is potentially interesting but it requires a statistical test to show if the observed gradient pattern is significant, especially at the single subject level.

Reviewer #3 (Remarks to the Author):

The authors demonstrate that traveling waves (TW) propagate along a gradient of the local sums of incoming connection strengths (i.e., instrength gradients) in a simulated weakly coupled oscillator network. They further showed that connectivity profiles in the human connectome are distance dependent and display instrength gradients that would allow for the emergence of directed traveling waves. When fitting cortical network models to empirically derived functional connectivity profiles, the authors found strong correlations when guided TWs emerged in the model. These models also produced smooth effective frequency gradients, leading the authors to suggest that functional connectivity could be generated via TWs following instrength gradients, which simultaneously generate resting-state frequency gradients.

The paper is thorough, clearly written and potentially of interest for a broad audience. The accompanying code seems complete and runs out of the box and all data necessary to reproduce the results is publicly available. I have some concerns and remarks, listed below in no particular order, that could, however, strengthen and clarify the manuscript:

1. The concepts of "instrength gradients" is very prominent in the manuscript, but it is only briefly defined without further explanation. The paper would be easier to follow after a careful introduction to instrength gradients early on in the text.

2. The authors report that the human connectome hosts instrength gradients and that traveling waves propagate along those gradients. Given that gradients found in the connectome are structural in nature and (relatively) stable across time, can the authors relate their findings to those of task based experimental work that found traveling wave direction to depend on task

demands (e.g., Alamia, Mohan, etc.)?

3. Line 137: a p-value of 0.315 is reported as significant. Is that a typo?

4. In the section on control models (line 222ff), the authors test a model that couples connection strength to Euclidean distance while destroying original instrength gradients and show that this model produces waves that do not follow the original instrength gradients. With regards to other control models they also talk of "distance dependent conduction delays." First: Is it correct to assume that distance here also refers to Euclidian distance? And second: Could the authors comment on how their results relate to the findings of a recent publication by Pang et al (Nature, 2023), in which the authors find that geometric eigenmodes derived from the cortical surface explain a large fraction of the variance in functional connectivity?

5. Lines 66-70 read as somewhat contradictory. The authors first state that evidence for intrinsic frequency gradients in humans is lacking, to then contrast with findings in animals. My hangup is with the "in particular," followed by a bunch of evidence in human cortex, that comes after. Could this section be rephrased for clarity?

6. Line 340 "effective frequency:" Could the authors please provide a brief definition there?

7. Line 399ff: "This suggests that FC could be coordinated by cortical traveling waves following structural connectivity gradients that simultaneously generate resting-state frequency gradients." Could the authors please comment on how they get to the conclusion that instrength gradients independently lead to traveling waves and frequency gradients, and not e.g., that instrength gradients lead to frequency gradients, which in turn lead to traveling waves, especially considering that they state in lines 516ff: "...We hypothesize that external or self-generated stimuli affect traveling waves by changing oscillation frequency patterns."

Point-by-point response

We thank the reviewers for their insightful comments which significantly strengthened our work and resulted in a substantially revised manuscript along with additional simulations and analyses. Please find our comprehensive point-by-point responses below. Our responses are written in **black** and are interspersed with the reviewers' statements in **blue**. Quotes from our old and revised manuscripts are *italicized* and new additions are marked in **dark yellow**.

Other remarks:

- In response to the reviewers' comments, we improved the clarity of our introduction with regards to key concepts such as instrength, intrinsic, and effective frequency gradients (we also made smaller changes throughout the manuscript). Additionally, we now present a figure introducing these key concepts.
- We also introduced acronyms for intrinsic frequency (IF) and effective frequency (EF). IF is the frequency of an oscillator disconnected from a network of oscillators. EF is the frequency that emerges when an oscillator is connected to a network of oscillators.
- Although not requested by the reviewers, we switched the statistical detection of instrength-directed traveling waves in our computational models from random permutations to spin permutations, which preserve many properties of the original data including spatial autocorrelation. This change makes our analyses more robust against potential false positives resulting from spatial autocorrelation. Please find the details in the methods sections "Detection of instrength-directed traveling waves" and "Spin permutation tests".

Reviewer #1

This study shows computational simulations on instrength-gradients hypothesis for generating cortical traveling waves with frequency gradients. All simulations appear properly implemented and the results also appear reasonable. However, unfortunately, the current results are not sufficient for proving the instrength-gradient hypothesis in terms of the following points. (1) The instrength-gradient hypothesis does not reject the frequency-gradient hypothesis proposed in the early study (Ermentrout and Kopell, 1984) and it rather appears a kind of paraphrase of the frequency-gradient hypothesis. (2) The instrength-gradient hypothesis does not agree with the experimental evidence showing frequency-band-dependent directionality of the frequency gradient (Mahjoory et al., 2020). Please revise the manuscript to show clear proof of the instrength gradient hypothesis in terms of the theoretical mechanism and the explainability of the pre-existing experimental evidence.

We thank the reviewer for their valuable comments and address each point below.

(1) The instrength-gradient hypothesis does not reject the frequency-gradient hypothesis proposed in the early study (Ermentrout and Kopell, 1984) and it rather appears a kind of paraphrase of the frequency-gradient hypothesis.

Response 1

We agree with the reviewer, our hypothesis does not challenge intrinsic frequency gradients as a mechanism to affect traveling wave direction. Rather, we suggest an additional mechanism underlying cortical wave direction, namely instrength gradients. We postulate that instrength and intrinsic frequency gradients shape the direction of traveling waves cooperatively. We apologize that our writing was not clear and restructured our introduction accordingly on lines 55 - 66 of the revised manuscript

*„Early theoretical work has shown that distance-dependent **connectivity or time delays (Figure 1a)** give rise to traveling waves in weakly-coupled oscillator networks, a frequently used system to study synchronization phenomena.¹⁴⁻¹⁶ Further simulation studies demonstrated that traveling waves **can be directed by intrinsic frequency (IF) gradients, where IF is the frequency of an oscillatory unit disconnected from a network (Figure 1b).**¹⁵ Once coupled in a network, traveling waves propagate from high to low IF oscillators (Figure 1e). The IF gradient **mechanism has been proposed to explain the propagation direction of cortical traveling waves in experimental recordings^{3,17} but we lack evidence for IF gradients across the human cortex due to methodological challenges. In non-human animals, IF gradients have been measured invasively by slicing neural tissue into disconnected self-oscillatory units.**^{18,19}*

While we speculate that cortical IF gradients exist in humans, we propose an additional mechanism that could affect the direction of traveling waves and is accessible through non-invasive tractography, namely structural connectivity (SC) instrength gradients – the sum of incoming fiber tracks (also known as weighted in-degree; Figure 1c). Here, SC instrength gradually changes across cortical space - similar to other cortical gradients such as

functional connectivity²⁰, gene expression,²¹ receptor distributions,²² myelin content,²³ cortical thickness,²⁴ or synaptic spine density.^{25,26} We postulate that this instrength gradient directs traveling waves from low to high instrength cortical regions. While previous computational studies investigated the emergence of traveling waves (Figure 1a),^{27,28} we focus on their propagation direction (Figure 1b).“

We also included a new introduction figure that explains both the IF- and instrength-gradient mechanisms:

Figure 1. Mechanisms of wave emergence and direction in weakly-coupled oscillator networks. **a** Mechanisms of wave emergence are based on neighborhood connectivity. Illustrated is a cortical network where circles and lines represent oscillators and their connections (top). The zoomed in graph (bottom) shows a chain of oscillators and how their connection strengths decrease with distance (blue line; this could also be connection probability), while their conduction delay increases (beige line). These mechanisms create neighborhood connectivity by emphasizing local synchronization between oscillators. **b** Mechanisms of wave direction include instrength gradients (top) and intrinsic frequency (bottom). Intrinsic frequency is the frequency at which the nodes oscillate isolated from the network (connections are removed). Here, the intrinsic frequency is equal across oscillators in the instrength gradient mechanism indicated by the same oscillator color and example activity (black sine curves). In contrast, the intrinsic frequency gradient mechanism is exemplified by a gradual increase of intrinsic frequency along the anterior-posterior axis illustrated by a gradual change of oscillator color. **c** Instrength is the sum of incoming connection strengths. This is illustrated for two oscillators by the addition of connections with different strength (thick and thin black lines are high and low strength connections, respectively). The instrength increases along the anterior-posterior axis for the instrength gradient mechanism, while it is equal for the intrinsic frequency gradient mechanism. **d** Effective frequency is the frequency assumed by oscillators connected within a network (black lines present). Here, effective frequency is again illustrated by the oscillator color and activity examples. The instrength gradient mechanism generates a smooth effective frequency gradient decreasing along the anterior-posterior axis (opposite of instrength gradient), while the intrinsic frequency gradient mechanism shows clusters with gradually increasing effective frequency along the anterior-posterior axis (same as intrinsic frequency gradient). **e** Traveling waves emerge in both networks from the mechanisms described in **a** and are directed by the instrength or intrinsic frequency gradient mechanisms. Traveling waves propagate from low to high instrength oscillators and from fast to slow intrinsic frequency oscillators as illustrated by the thick black arrows and the color gradients. Both mechanisms of wave direction can interact as shown in Figure 3.

In addition, we now present a model that tests how the interaction of instrength and intrinsic frequency gradients gives rise to wave direction and effective frequency gradients in the revised manuscript (see **Response 3**).

(2) The instrength-gradient hypothesis does not agree with the experimental evidence showing frequency-band-dependent directionality of the frequency gradient (Mahjoory et al., 2020).

Response 2

We thank the reviewer for this observation and address them in our response to the reviewers more specific comments (see **Response 6**).

Major points

(1a) "Instrength" or "intrinsic frequency"?

Early theoretical study has shown that connected oscillators with intrinsic-frequency gradients result in the formation of traveling waves (Ermentrout and Kopell, 1984) and it is also discussed as a key mechanism generating the cortical traveling waves (Zhang, et al., 2018). The current results did not evaluate the influence of the intrinsic frequency gradient, but such estimation would be technically possible in Kuramoto model where the connection strength (a_{ij}) and intrinsic frequency (ω_i) are separately described. It is encouraged to clarify what type of spatial gradient would be more likely to explain the cortical traveling waves.

Response 3

We thank the reviewer for their suggestions. We hypothesize that instrength and intrinsic frequency gradients co-exist in the cortex, such that both mechanisms act simultaneously. We suggest that instrength gradients remain relatively stable (white matter changes on a slow time scale) and affect traveling wave direction across brain states while interacting with other mechanisms such as intrinsic frequency gradients. To our knowledge, there is no evidence for intrinsic frequency gradients across the human cortex (IF gradients were observed in non-human animals; Diamant and Bortoff, 1969; Ermentrout et al., 1998) but we speculate that local or global intrinsic frequency modulation is one of the mechanism behind dynamic changes of traveling wave direction in response to tasks. We have now explained this more thoroughly in our revised discussion on lines 796 - 815

"We used the tractography-derived number of streamlines as a proxy for coupling strength between brain regions. Cortical traveling wave direction and EF patterns are likely subject to additional factors such as cortical gradients of neuron density, synaptic spine count, receptor distributions, myelin content, cortical thickness, and excitation-inhibition ratio.^{24,81-87} While some cortical gradients may vary on faster timescales large-scale SC gradients remain relatively stable over longer time periods such as many days, weeks or years.^{48,49} Thus, SC gradients could contribute to traveling wave direction across brain states, e.g. similar traveling wave directions have been observed in the alpha band during rest and memory-tasks.^{3,9} Other studies have shown that traveling wave direction changes rapidly in response to tasks.^{70,71,88} We hypothesize that external or self-generated stimuli dynamically affect traveling waves by modulating IF.⁸⁹ For example, a stimulus arriving in the visual cortex could accelerate local oscillations⁹⁰ that interact with large-scale stable coupling and IF gradients to achieve cortex-wide processing through traveling waves. Alternatively, stimuli could induce large-scale IF gradients to direct cortical traveling waves, this could be achieved through thalamocortical loops preparing the cortex for incoming stimuli. We explored this latter mechanism in our 2D network model and found that stable instrength and dynamic IF gradients could cooperatively direct traveling waves. Alternatively, inter-regional coupling strength could be dynamically controlled – for instance by modulating the long-range

excitation-inhibition ratio.⁸⁶ To explain behavior, deeply understanding stable and dynamic coupling and IF gradients will be necessary.”

To systematically explore how intrinsic frequency and instrength gradients can interact to coordinate wave direction, we included an additional 2D network model experiment. We superimposed a scaled opposing intrinsic frequency gradient with an instrength gradient. Our results show that the network can switch between the two wave directions depending on the scale of the intrinsic frequency gradient. Importantly, this experiment also shows that large enough intrinsic frequency gradients can determine wave direction in the presence of instrength gradients, which could be necessary for task-dependent changes of wave direction. Please find the corresponding section in the results on lines 207 - 231

*“Previous studies proposed that IF gradients direct traveling waves in the human cortex.^{3,18} We hypothesize that instrength gradients contribute to traveling wave direction and interact with IF gradients. To investigate this, we generated superimposed instrength and IF gradients with the exact same shape from a gradient template (**Figure 3a**). Notably, the two mechanisms generate opposing wave directions from low to high instrength and high to low IF,¹⁵ respectively. We scaled the IF gradient from zero (IF is 10 Hz for all oscillators) to range from 8.5 to 11.5 Hz (gradient scaling = 1.5) with a superimposed fixed instrength gradient. The instrength gradient fully directed traveling waves without the IF gradient as assessed by gradient template and flow potential correlation (**Figure 3b** top row). This is also reflected in the average flow potential and illustrated for an example wave timeseries (**Figure 3c** and **e** top). The instrength gradient’s influence weakened with increasing IF gradient scaling until both balanced each other out (gradient scaling @ 0.75; IF gradient ranging from 9.25 to 10.75 Hz; **Figure 3b**). At this point, simulated activity varied: we observed spiral waves, plane waves, traveling waves, and full synchrony (**Figure 3e** middle). The average flow potential suggested that waves source from the periphery and sink into the network center (**Figure 3c**) but individual simulations’ flow potentials varied reflecting the diversity of waves observed. A further increasing IF gradient switched the wave direction from instrength- to IF-directed (**Figure 3b**) as shown by the average flow potential (**Figure 3c**) and an example timeseries (**Figure 3e**).*

*Earlier, we showed that instrength gradients shape EF patterns (**Figure 2e**). We next investigated if instrength and IF gradients could cooperatively shape EF patterns. We found that the correlation between gradient template and EF switched from strongly negative to positive as we increased the IF gradient scaling (**Figure 3b** and **d**). This suggests that instrength and IF gradients together shaped EF gradients in the 2D network model.”*

Figure 2. Instrength and intrinsic frequency gradients interact to determine wave direction and shape effective frequency. **a** Gradient template used to create scaled intrinsic frequency gradients (0 to 1.5) and the instrength gradient. The intrinsic frequency gradients entered the 2D network model through the intrinsic frequency term ω_i , while the instrength gradient was used to scale the connectivity matrix before it entered through the local coupling term a_{ij} . **b** The top figure shows how the gradient template and flow potential correlated depending on the intrinsic frequency gradient scaling. The bottom figure shows how the gradient template correlated with the effective frequency pattern. Both graphs are based on 100 randomly initialized simulations of 10 s duration (1 s transient removed) per gradient scaling. The thick blue lines show the mean correlation and the light blue shaded areas show the standard deviation across simulations. **c** Average flow potentials for three distinct gradient scalings describe the average propagation of traveling waves from higher to lower potentials. **d** Effective frequency maps show how the gradient scaling affects effective frequency patterns. **e** Example timeseries of activity (cosine of instantaneous phases) emerging at three distinct gradient scalings are shown. The top row shows a traveling wave following the instrength gradient. The middle row shows a spiral wave that emerged when instrength and intrinsic frequency gradients were balanced. The bottom row shows a traveling wave that follows the intrinsic frequency gradient.

and in the methods on lines 976 - 998

*“We investigated if instrength and IF gradients interact to direct traveling waves. To do so, we built on the 2D instrength gradient model introduced earlier. We added an IF gradient generated from the same gradient template $g(x_i)$ that we used to construct the instrength gradient (see **Figure 3a**). We chose to scale the IF gradient from 0 to 1.5 (IF ranges from 8.5 to 11.5 Hz at maximum scaling) in steps of 0.05 (IF range increment of 0.1 Hz) superimposed on a fixed instrength gradient and simulated 100 randomly initialized models per scaling factor. We investigated gradient template – flow potential Spearman correlation for all scaling factors. In this model, a negative correlation indicates instrength-directed waves, while a positive correlation indicates IF-directed waves. We also investigated the gradient template – EF Spearman correlation, where a negative correlation indicates that EF patterns are shaped by the instrength gradient, while a positive correlation means that they are shaped by the IF gradient.”*

The reviewer's comment has drawn our attention to concepts that needed clarification and we hope addressing those has improved our revised manuscript.

(1b) Additional control model using the normalized connection weight ("Instrength" or "Hub-structure"?)

In the current manuscript, there is no control model using the normalization of connection weights (i.e., connection from oscillator j to i , a_{ij} , is replaced to $a_{ij}/\sum_i(a_{ij})$), but it has been often used in many connectome-based models (c.f., Roberts, et al., 2019). To clarify the importance of instrength but not to connection pattern (i.e., Hub-structure) on the traveling waves, it is necessary to include this type of control model. This point is also associated with the comment (1a).

Response 4

Previously, we used an instrength-normalized connectivity in the "uniform" 2D Kuramoto network model as control and showed that waves emerge but do not follow a systematic pattern. These waves rather vary in their direction. Interestingly, rotating waves appear to be more prominent in this scheme. In response, to the reviewer's comment, we have also simulated and analyzed an instrength-normalized cortical network model. We found that traveling waves emerged in this model but did not follow the original instrength gradient. In contrast to the 2D model, we found that the average wave flow potential has sources in the occipital cortex and medial frontal cortex and sinks in the temporal lobe and lateral frontal cortex (see Figure 5b). We have investigated the wave flow potentials correlation with several network metrics such as the node-degree, betweenness, and eigenvector centrality, but could not identify a specific metric that guides these waves. We have added our findings using this control model to our manuscript on lines 375 - 387

"We also investigated a control model with instrength-normalized SC to outrule node-degree influences on wave direction (see Methods). We found that this model produced traveling waves 94.7% of times but none of them followed the original instrength gradient (Figure S4a). Notably, a systematic average wave potential indicated that emerging waves propagated from visual cortex and medial frontal sites to the temporal lobe and lateral frontal sites (Figure 5b – instrength-normalized; Movie 8) suggesting a distinct mechanism directing those waves. We tested if the systematic wave potential could be explained by node degree ($r = 0.11$, $p = 0.049$), betweenness centrality ($r = 0.1$, $p = 0.011$) or eigenvector centrality ($r = 0.14$, $p = 0.292$) but we could not identify a significant relationship. Hub-structure did neither determine instrength-directed waves nor systematic wave propagation emerging in this model. However, the emerging EF correlated significantly with the wave potential ($r = 0.39$, $p < 0.01$). Further research needs to identify the mechanism at play in this model."

And to our methods on lines 1045 - 1049

„We tested if hub-structure could explain the traveling wave direction observed in our cortical network model by instrength-normalizing the SC. To do so, we divided the SC matrix row-wise by the region's instrength and multiplied it by the mean instrength of the original SC. Everything else remained equal to the original cortical network model."

We did find that rotating waves were prominent in models with instrength-normalized connectivity or whenever the effects of intrinsic frequency and instrength gradients were balanced (see **Response 3**). This is intriguing and suggests that mechanisms of rotating waves take over whenever IF or instrength gradients are absent/weak. We added a paragraph to our discussion noting this observation on lines 719 - 725

„Movies of our instrength-normalized cortical network model showed prominent rotating waves too. This was also the case in the 2D control network with instrength-normalized connectivity and when interacting IF and instrength gradients were balanced. We hypothesize that mechanisms underlying rotating waves take over whenever IF or instrength gradients are either weak or balanced. Our experiment on interacting IF and instrength gradients suggests that the brain could dynamically switch between different wave types by modulating IF gradients.“

We thank the reviewer for this suggestion, which led to insights that could be highly valuable for further research.

(1c) Fluctuation of traveling waves?

Fluctuations in the cortical traveling waves have been thought to be one of the fundamental properties for the understanding of the phenomena and its functional importance, and many theoretical models have also focused on it (Roberts, et al., 2019). The current manuscript does not evaluate these properties, but it is still important to show that the instrength-gradient hypothesis leads similar (or different) properties of the fluctuation with those in the pre-existing hypothesis and the experimental evidence. It would be one fundamental support of the current hypothesis. Please consider including these discussions in the revised manuscript.

Response 5

We thank the reviewer for this suggestion. We used the variability of the kuramoto order parameter (KOP) to assess fluctuations in simulated activity from our cortical network model. We estimated the standard deviation of the KOP for all simulations of our parameter scan and found that there are regions with higher variability in KOP:

The range of $sd(KOP)$ found here is similar to the one by Cabral et al. (2014; see their figure 3c) using a kuramoto model, Cabral et al. (2022; see their figure 2e)

using the more complex Stuart-Landau model and Deco et al. (2017; see their figure 4) using a Hopf model.

Next, we estimated the $sd(KOP)$ in the resting-state MEG recordings and found that the median $sd(KOP)$ across all subjects for the alpha, beta, and gamma bands are 0.058, 0.056, and 0.056, respectively. Interestingly, the cortical network models that fitted the MEG functional connectivity best (assessed with PLV-FC), produced very similar median $sd(KOP)$ for the alpha, beta, and gamma bands with 0.067, 0.061, 0.048. This indicates that the synchronization fluctuations are similarly variable in the simulation compared to the empirical data.

We further assessed fluctuations using the interhemispheric cross-correlation (ICC) used in Roberts et al. (2019). We first used the ICC to estimate dwell times in our cortical network model parameter scan and found that the median dwell times varied with coupling scaling and conduction speed (models with no state transitions were masked in white)

Next, we estimated dwell times of cortical states in the resting-state MEG recordings and found that their distribution was right-skewed with a median of 270, 240, and 200 ms in the alpha, beta and gamma bands, respectively (see Supplementary Figure 7c). We compared these empirical distributions with dwell times generated by the simulations that fit the empirical PLV-FC best. Those models generated a right-skewed distribution with median dwell times of 430, 255, and 170 ms in the alpha, beta and gamma bands (see Supplementary Figure 7). A Kolmogorov-Smirnov test indicated that the empirical and simulated alpha distributions were significantly different, while the beta and gamma distributions were indistinguishable. Notably, Roberts et al. (2019) observed ICC-derived dwell times with a median of 48 ms in their simulations. They show that this matches dwell times estimated with a hidden markov model (HMM) from MEG resting-state recordings (Vidaurre et al., 2018). Longer duration HMM states have been estimated in resting-state MEG recordings by Baker et al. (2014; 100-200 ms) and Coquelet et al. (2022; 128-211 ms). Trujillo-Baretto et al. (2019) and Gohil et al. (2022) also found longer duration states with methods that correct the known HMM-bias towards short duration states. To our knowledge, there are no other brain state dwell time estimates using ICC for MEG resting-state data. The discrepancy between dwell times in Roberts et al. (2019) and our model could be due to their very short time delay of 1 ms. Notably, they used a single time delay for all connections, while we use heterogenous time delays estimated by dividing the empirical tract lengths with realistic conduction speeds between 1 and 10 m/s resulting in mean time delays of > 8 ms. The specific alpha,

beta, and gamma cortical models we used to estimate dwell times had mean time delays of 27, 13, and 8 ms, respectively. Roberts and colleagues' model produced very high wave speeds between 23 and 52 m/s contrasting with empirical findings of wave speeds < 10 m/s similar to our cortical network model. While these results are interesting, we suggest presenting them as supplementary information to keep the focus of our study on the mechanism affecting wave direction and effective frequency patterns. We mention these results on lines 525 - 527

“We also explored if simulated cortical network dynamics exhibit metastability and found similar synchronization variability and state dwell times between the best fitting cortical network models and resting-state MEG (Supplementary Fig. 7).^{27,53–55}”

We thank the reviewer for this comment that has led to findings of cortical network model fluctuations comparable to empirical and computational studies.

(2) Frequency-band-dependent directionality of the frequency gradient

Experimental evidence have shown that the cortical traveling wave dominantly appear in theta-alpha-band oscillation (Zhang, et al., 2018) and that the directions of peak-frequency-gradient are highly dependent on their frequency band (i.e., Alpha-band gradient direction is opposite to beta-band gradient direction) (Mahjoory et al., 2020). As shown in Figure 5, the current hypothesis does not agree with these evidence; i.e., the current mechanism principally produce cortical waves independent to their frequency band (when the conduction speed is around 7 m/s (which is still within a physiologically reasonable value)). A short, interesting discussion on cortical layer-specific structural connectivity is described (line 480-), however, it does not appear convincingly; i.e., the layer-specific oscillation in the experiments does not seem to be organized with gradual frequency-dependent profiles (i.e., Halgren, et al.(2019) showed dominantly alpha oscillation the layers. Bastos, et al. (2018) showed that alpha/beta band is dominant is deep layers and gamma-band is dominant in superficial layer). Please solve the disagreements of the experimental evidence with the current hypothesis and results.

Response 6

In response to the reviewer's comment, we have investigated further measures of functional connectivity (FC) that showed frequency-specificity. In our original manuscript, we used the phase locking value (PLV) as a measure of FC, which is known to be sensitive to volume conduction. We decided to use the PLV despite known limitations because it is commonly used and zero-lag interactions have been shown to be meaningful (Petkoski et al., 2018). We now also use the phase lag index (PLI) to fit our models to the empirical resting-state MEG FC. The PLI is only sensitive to non-zero-lagged interactions and thus possibly more suitable for FCs reflecting some aspect of traveling waves. We found that the simulated and empirical PLI-FCs strongly correlated only in the alpha band (see Figure below). While the origin of alpha oscillations is still highly debated (deep vs. superficial vs. network interactions), it is possible that alpha interacts through the full SC network unspecific to cortical layers and thus we observe strong alpha-specific PLI-FC correlation.

We added these findings to our section on FC fit on lines 518 - 524

*“Signal volume conduction can result in spurious zero-lag phase relations with high PLV. We additionally analyzed FC based on the phase lag index (PLI),⁵² which ignores spurious but also true zero-lag interactions. Resting-state alpha PLI-FC decreased notably compared to the PLV (**Figure 8g and b**). We found that simulated and empirical PLI-FCs strongly correlated in the alpha band ($r = 0.46$), while beta and gamma bands had weaker correlations (beta: 0.36; gamma: 0.14; **Figure 8f**). This suggests that non-zero-lag phase interactions could be frequency-specific; we investigated this further in the next section.”*

And changed our section summary on lines 528 - 533 accordingly

“We found that our cortical network model achieved high correlation with empirically derived PLV-FCs if directed traveling waves emerged and smooth EF gradients were produced across the alpha, beta, and gamma bands. PLI-FC fit decreased from alpha to beta to gamma band. Our findings suggest that zero-lag FC across frequency bands could be coordinated by cortical traveling waves following SC instrength gradients, while non-zero-lag FC is more specific to the alpha band.”

We introduced the PLI in our methods on lines 1128 - 1136

“We further used the phase lag index (PLI),⁵² which is insensitive to zero-phase lag interactions and thus also to possible volume conduction. PLI quantifies the asymmetry of the phase difference distribution between signals. It is 1 if the two regions have consistent non-zero phase differences and 0 if they have zero or random phase differences. The PLI between regions k and l is defined as

$$PLI_{kl} = \left| \frac{1}{N} \sum_{n=0}^N \text{sign}[\text{Im}(e^{j(\theta_k(n) - \theta_l(n))})] \right|$$

Where $\text{sign}[\cdot]$ is the sign-function and $\text{Im}(\cdot)$ is the imaginary part.”

We used the number of streamlines estimated with diffusion weighted imaging (DWI) between cortical regions as a proxy for connection strengths. The spatial resolution of DWI is crude and cannot resolve cortical layer-specific connectivity. Thus, DWI

likely captures the total connectivity across all cortical layers and detailed layer-specific connectivity will hopefully become available with advanced neuroimaging technologies. Nevertheless, it is conceivable that cortical layers of distributed regions are connected through layer-specific networks with instrength patterns distinct from the one found in our study. Hence, layer-specific instrength gradients could explain the distinct effective frequency gradients found by Mahjoory et al. (2020). Halgren and colleagues (2019) showed that alpha activity might originate from superficial layers of the cortex. Superficial layers of distributed regions generating alpha oscillations could be connected through a network with an instrength gradient increasing from posterior to anterior areas. This would result in alpha waves traveling from posterior to anterior areas and an effective frequency gradient as observed by Mahjoory in the alpha band. In parallel, deep cortical layers of distributed regions generating beta oscillations could be connected through a network with an instrength gradient increasing from anterior to posterior areas, resulting in an effective frequency gradient consistent with Mahjoory et al's observations in the beta band. Of course, such a system becomes more complex if cortical layers were interconnected between distinct brain regions as is commonly assumed (Mejias 2016; Bastos 2018, etc.). However, it is still debated whether there is strong overlap between frequency-specific networks or if they operate largely independent of each other (Deco et al. 2017).

In response to the reviewer's comment, we tested if the full SC network could be decomposed into subnetworks that explain the findings by Mahjoory et al. Indeed, we identified such subnetworks within the total SC of the HCP subjects using nonnegative matrix factorization (NMF). We used those subnetworks to build alpha and beta cortical network models and found that they fit Mahjoory and colleagues' observations. We included a new section to our manuscript detailing our findings on lines 550 - 642

“Connectome subnetworks explain effective frequency gradients and traveling wave direction in alpha and beta bands

Mahjoory and colleagues³¹ found large-scale EF gradients in resting-state MEG; alpha EF gradients increased along the anterior-posterior axis while beta EF gradients decreased. Does our cortical network model with instrength gradients generate EF patterns consistent with those observations?

We computed resting-state MEG EF gradients in the alpha and beta bands (see Methods). We found an alpha EF gradient increasing from occipital to prefrontal areas, while a beta EF gradient increased from prefrontal to occipital regions (Figure 9a). Our findings corroborate the EF gradients that Mahjoory and colleagues observed.³¹ Next, we explored the fit between empirical and simulated EF for the same coupling strengths and conduction speeds as before. We used the concordance correlation coefficient (CCC) to evaluate empirical and simulated EF fit (see Methods). CCC comprehensively measured if the empirical and simulated EFs were correlated and of similar magnitude. If two measurements are fully correlated and have the same magnitude the CCC is 1, if they are fully anti-correlated and have the same magnitude it is -1, and if they are not correlated and of different magnitude it is 0. We found parameter regions where empirical and simulated EFs fit best (Figure 9b).

Alpha EF fit peaked with a CCC of 0.2 at a conduction speed of 9 m/s and medium coupling scaling. This model's alpha EF gradient strongly correlated with the instrength gradient ($r = -0.73$; Figure 7a) but frontal EF was lower than occipital EF (Supplementary Fig. 8a).

Instrength-directed alpha traveling waves emerged across all timepoints and simulations (Figure 6 and Supplementary Fig. 8a).

Beta EF fit peaked with a CCC of 0.38 at 2 m/s and medium coupling scaling (Figure 9b) and the emerging EF gradient increased from posterior to anterior sites (instrength – EF correlation = -0.39; Supplementary Fig. 8a). We detected beta traveling waves in 38% of timepoints but only 6% were instrength-directed (Figure 6). The wave flow potential suggested frontal and temporal sources along with sinks in parietal areas and the temporal poles (Supplementary Fig. 8a) but beta traveling waves were not noticeable in corresponding movies.

Next, we studied the combined PLI-FC and EF fits and found similar best fit parameters to the ones obtained by EF fit only (alpha fit = 0.66 at 9 m/s and slightly lower coupling scaling; beta fit = 0.57 at 2 m/s and same coupling scaling; Figure 9c) resulting in similar EF patterns and wave potentials for both models. In sum, while the cortical network model partially explains the empirical EF gradients, our results were ambiguous for emerging beta traveling waves.

We next explored if SC subnetworks could better explain experimentally observed opposing alpha and beta EF gradients and traveling waves.^{31,56} Such putative subnetworks could be cortical layer- or frequency-specific.^{57,58}

To identify SC subnetworks, we used nonnegative matrix factorization (NMF) an unsupervised algorithm that decomposes nonnegative high-dimensional data into nonnegative low-dimensional components and associated loadings (see Methods).⁵⁹ Because of the nonnegativity constraint, we can interpret NMF components as additive parts of a whole, for example subnetworks of the full SC network.

We constructed a large matrix where each column was associated with the flattened SC of individual HCP subjects (see Methods). Given the subjects' SCs, NMF finds subnetworks that are common across SCs. Determining the optimal number of components — in our case subnetworks — remains challenging. If the number of components is too small, they might miss important detail and if it is too large, they capture noise. We determined the number of subnetworks using rank-2 to -10 NMF cross-validation that randomly zeroed and imputed datapoints for training and testing, respectively^{60,61} (Figure 9c). Five subnetworks minimized the error between held-out original values and imputed datapoints. These subnetworks' instrength gradients matched our expectations for alpha- and beta-specific SCs (Figure 9d).

To validate subnetwork stability, we compared their similarity across rank-2 to -5 NMFs (14 subnetworks in total) — the ranks with low reconstruction error (Figure 9c). Hierarchical clustering identified two groups of subnetworks whose instrength patterns decreased or increased along the anterior-posterior axis as would be expected for alpha- and beta-subnetworks (Figure 9e). In sum, NMFs found consistent alpha- and beta-subnetworks across ranks that explained up to 63% of SC variance.

We built two rank-5 NMF subnetwork models to test if they generated EF gradients and traveling wave directions consistent with empirical observations. We chose the putative alpha- and beta-subnetworks with the strongest positive and negative correlation between instrength and posterior-anterior position (Figure 9d). Combining the alpha- and beta-subnetworks explained 36% of SC variance; this network's instrength strongly correlated with the total SC's instrength ($r = 0.96$, $p < 0.01$). The alpha- and beta-subnetwork models improved the EF fit drastically compared to the original cortical network model (compare Figure 9b and f). The alpha band CCC was 0.69 at 10 m/s and medium coupling scaling, while the beta band CCC was 0.50 at 2 m/s and medium coupling scaling. Empirical and simulated

EFs in the alpha and beta bands at peak CCC were similar with opposing anterior-posterior gradients (compare **Figure 9a** and **g**). While the alpha wave potential suggested posterior to anterior propagation (**Figure 9i** top), beta wave potentials were more complex (**Figure 9i** bottom). Example movies of beta waves did not show noticeable waves. We replicated these findings using the rank-2 NMF subnetworks – with fewer parameters and similar reconstruction error (**Figure 9d** inset and Supplementary Fig. 8c-f).

Using the combined FC and EF fit, we found parameters for the rank-5 alpha- and beta-subnetworks that improved the fit by 62% and 47% compared to the original cortical model (**Figure 9j**). The parameters for the alpha network coincided with the ones found with the EF CCC and this model's empirical – simulated PLI-FC correlation peaked at 0.38 (**Figure 9g**). This network generated traveling waves 88.7% of time across 100 simulations and the alpha-subnetwork's instrength gradient directed all emerging waves (see Supplementary Fig. 4a and Movie 12). The subnetwork's instrength gradient correlated significantly with the average wave flow potential ($r = -0.75$, $p < 0.01$) and effective frequency pattern ($r = -0.77$, $p < 0.01$). The beta network model fit best at an increased conduction speed of 9 m/s with an almost unchanged EF CCC of 0.49, while the PLI-FC fit increased from 0.31 to 0.40. This beta-subnetwork model showed a less pronounced anterior-posterior EF decrease compared to the best EF CCC fit (**Figure 9k**). The wave potential suggested beta waves travel from frontal and temporal to posterior but also anterior sites (**Figure 9l**). Traveling waves emerged 87.8% of time and all of them were instrength-directed (see Supplementary Fig. 4a and Movie 13). The beta-subnetwork's instrength gradient correlated significantly with the average wave flow potential ($r = -0.78$, $p < 0.01$) and effective frequency pattern ($r = -0.84$, $p < 0.01$). We achieved comparable results using the 2-rank NMF subnetworks (Supplementary Fig. 8g-i).

We showed that the full SC network can be decomposed into additive subnetworks whose instrength gradients generated alpha- and beta-band EF gradients and traveling wave directions matching empirical data while maintaining high PLI-FC.”

Figure 9. Alpha- and beta-subnetworks match resting-state MEG effective frequency gradients and functional connectivity. **a** Average alpha and beta effective frequency gradients measured across HCP resting-state MEG of 80 subjects. **b** Parameter exploration of cortical network model using the full network (structural connectivity shown in Figure 4) and the concordance correlation coefficient (CCC) between empirical (see **a**) and simulated effective frequency. **c** Parameter exploration of the full network with the combined PLI-FC and effective frequency fit. **d** Cross-validation of root mean squared error between training and test datapoints estimated by nonnegative matrix factorizations (NMF) with different numbers of subnetworks (blue dots and bars represent the mean and standard deviation across 50 randomly initialized NMF runs). The inset shows a zoom-in of the root mean squared error of rank-2 to -5 NMFs. **e** Instrength patterns of the rank-5 NMF subnetworks projected onto the cortical surface with highlighted alpha- and beta-subnetworks used for simulations. **f** Pairwise correlation matrix between subnetwork components across rank-2 to -5 NMFs (total of 14 components; see inset in **d**) ordered in hierarchical clusters (dendrogram in black). The two high-level clusters show subnetwork instrength patterns decreasing and increasing along the anterior-posterior axis corresponding to putative alpha- and beta-subnetworks. **g** Empirical – simulated effective frequency CCC for parameter exploration of alpha- and beta-subnetworks (compare with **b**). **h** and **i** show the average effective frequency and flow potentials estimated from alpha- and beta-subnetwork simulations at the best effective frequency CCC fit (across 10 randomly initialized simulations). **j** Empirical – simulated PLI-FC + effective frequency CCC fit for parameter exploration of alpha- and beta-subnetworks (compare with **c**). **k** and **l** show the simulated average effective frequency at the best fit for the beta-subnetwork in **j**. The best fit for the alpha-subnetwork coincided with the one obtained for the effective frequency CCC fit in **g**.

And in our methods on lines 1161 - 1202

„Decomposing the structural connectivity into subnetworks

Tractography is indifferent to subnetworks underlying the total SC. Nonnegative matrix factorization⁵⁹ (NMF) – an unsupervised decomposition algorithm – allowed us to identify common subnetworks across the HCP subjects total SCs. NMF decomposes a nonnegative matrix V into two low-rank nonnegative matrices W and H

$$V \approx WH$$

V is a $n \times m$ matrix, where n is the number of features and m is the number of samples. W and H are $n \times r$ and $r \times m$ matrices, where r is the number of components. The components of W are feature patterns common across the samples, while the coefficients in H express how much each feature pattern contributes to each sample.

We built a data matrix V where each sample was the upper triangular matrix of an individual subject's SC flattened into a column vector. Then, we removed all connections across subjects that did not survive consistency-thresholding for the average SC (see Estimation of structural connectivity), which resulted in a 75000×776 nonnegative data matrix V . Notably, using the full upper triangular matrices of all subjects' SCs (V with dimensions 499500×776) resulted in similar subnetworks (see Supplementary Fig. 8b). We used the reduced data matrix in our simulations and analyses to speed up computations. We determined the number of components using a cross-validation method that first randomly sets 5% of the datapoints to zero, then computes the NMF, which is eventually used to impute the missing data.⁶⁰ The mean squared error between the imputed datapoints and the original held-out values was computed for NMFs with 2 to 10 components. We repeated this procedure 100 times because the NMF low-rank matrices require random initialization resulting in different outcomes across runs. We found that 5 components minimized the average mean squared error across all 100 runs. Finally, we used the 5-rank NMF with the lowest mean squared error across 100 random initializations. We calculated the average subnetwork SCs by reconstructing V from each component, followed by averaging across subjects. We used the RcppML package for the NMF and cross-validation.⁶¹

We identified putative alpha- and beta-subnetworks by finding the subnetworks whose instrengths correlated most strongly with the anterior-posterior position. The subnetwork with instrength decreasing along the anterior-posterior axis was chosen to be the putative alpha-subnetwork, and the one with increasing instrength the putative beta-subnetwork. These networks were used in the parameter exploration presented in the main text.

The cross-validation revealed that NMFs with fewer than 5 subnetworks had similar reconstruction errors compared to the best model. Thus, we wondered if these NMFs also identified similar subnetworks. To test this, we first computed the pairwise Pearson correlation between all feature patterns of the rank-2 to 5 NMFs (total of 14 subnetworks). Hierarchical clustering of the resulting correlation matrix using the Ward algorithm, revealed two high-level clusters. The instrength patterns of subnetworks belonging to one of the two clusters showed that the NMFs consistently found alpha- and beta-subnetworks.“

We showed that the total SC can be decomposed into subnetworks that explain the alpha and beta EF gradients observed by Mahjoory and colleagues (2020). Notably, we used an unsupervised decomposition that was not biased towards our expectations. The putative alpha and beta subnetworks were also stable across several low-rank decompositions. Our results provide novel insights on the SC and suggest that frequency-/layer-specific subnetworks might underlie the total connection strengths measured with DWI.

We expanded our discussion on frequency-specific networks in our revised manuscript on lines 763 - 782

„Furthermore, Mahjoory and colleagues³¹ found that EF gradients differed noticeably between theta, alpha and beta bands. We corroborated opposing alpha and beta EF gradients during resting-state MEG of HCP subjects. In contrast, our cortical network models

generated similar EF gradients across frequency bands. These differences might be explained by cortical layer-specific frequency and connectivity profiles.^{31,66,76,77} While the origin of alpha oscillations is still debated (infragranular⁷⁷ vs. supragranular⁹ vs. network interactions^{71,78}), a putative alpha-subnetwork with an instrength gradient decreasing from anterior to posterior sites and a parallel beta-subnetwork with instrength gradient increasing from anterior to posterior regions could explain EF gradients observed by Mahjoory and colleagues.³¹ We showed that an unsupervised algorithm finds such putative alpha- and beta-subnetworks within the full SC network. Simulating cortical network models with these subnetworks improved the empirical and simulated EF fit markedly. We assumed that putative alpha- and beta-subnetworks act independently in our cortical network model but whether frequency-specific subnetworks interact or operate largely independently remains debated.^{58,79} In a network with cross-frequency interactions, alpha oscillations could dynamically hijack the full network explaining the distinct traveling wave directions and EF gradients observed in ECoG vs M/EEG. Our findings suggest that frequency-specific structural subnetworks could exist, but the spatial resolution of MRI currently limits more direct evidence. Advances in neuroimaging are needed to conclusively identify cortical-layer and frequency-specific subnetworks.“

Furthermore, we expanded our discussion on FC with our PLI-FC results on lines 783 - 795

„Previous studies proposed that traveling waves could coordinate FC networks.^{10,50,75} We fit our cortical network model to resting-state MEG PLV-FC and found that the best fitting models produced instrength-directed waves corroborating that they could coordinate FC. Additionally, these cortical network models generated smooth EF gradients resembling observations from ECoG recordings during a memory task.³ We did not model volume conduction in our cortical networks and thus, high PLV between empirical and simulated source activity could reflect true zero-lag interactions, which are physiologically meaningful.^{64,80} To fully outrule volume conduction effects, we investigated non-zero-lag interactions with PLI-FC, which was largest in the alpha band and less pronounced in beta and gamma bands suggesting that instrength-directed alpha traveling waves could coordinate non-zero-lag FC. Simulated activity from putative alpha- and beta-subnetworks maintained a similar PLI-FC fit while also generating EF gradients consistent with resting-state MEG.³¹“

We thank the reviewer for this thought-provoking question, which has led to novel insights about effective frequency and the structural connectivity.

Minor points

(1) The current manuscript does not clearly describe why the understanding of the neural mechanism generating traveling waves is important. Please add some descriptions on previous arguments or its necessity on functional point of view.

Response 7

We thank the reviewer for pointing this out and added a paragraph on the clinical relevance of understanding the mechanisms that determine traveling wave direction to our introduction on lines 50 - 52

„Clinical studies have found that schizophrenia,¹¹ ADHD,¹² and memory deficits¹³ are related to altered cortical wave directions. Understanding mechanisms of wave direction could yield insights on healthy and pathological cognition.“

(2) The section on the 2-D network model appears trivial to me from the given Kuramoto model where the phase of oscillators receiving stronger connection weights results in earlier phase. Please revise the manuscript to show the novel points of these results or to move them to Supplemental information.

Response 8

To our knowledge, the relationship between instrength-gradients and wave propagation direction is novel. While previous studies (Moon et al., 2015, 2017; Petkoski et al., 2018) have investigated the relationship between node degree/strength and phase lead/lag, they did not explicitly investigate the emergence of node degree/strength/instrength-directed traveling waves. Notably, phase lag relations only translate into traveling waves if they are systematic across space; We test spatial phase relations by explicitly analyzing spatial phase gradients to detect traveling waves. Nevertheless, these earlier studies' analytical framework could provide a theoretical underpinning of the instrength-gradient mechanism proposed in our study. Thus, we mention their findings prominently at the beginning of our revised discussion on lines 671 - 675

„Analytical expressions derived in earlier studies suggest that higher instrength nodes phase-lag, while lower instrength nodes phase-lead in coupled oscillator models.^{62–64} When combined with spatial instrength gradients, this analytical framework could be a theoretical foundation for the instrength gradient mechanism.“

Additionally, we now also present how instrength-gradients shape effective frequency in this 2D model (see Figure 2e) and further explore interactions between instrength- and IF-gradients in **Response 3** to your comment 1a (see Figure 3). Thus, we think these explorations should remain part of the main text.

(3) What is the boundary constraints in the 2D network model?

Response 9

The boundaries of our 2D network model were unconstrained. We have modified the description of our network to

“We studied if traveling waves emerge and follow an instrength gradient in a 30 x 30 network model of Kuramoto oscillators assuming a side length of 140 mm and unconstrained boundaries.“

(4) Please describe the absolute values (and their distribution) of instrength in the connectome-based model and discuss their associates with experimental evidence (on cortical thickness, neuron density, etc.). This would be a support of the instrength gradient hypothesis.

Response 10

We have now added a figure showing the distribution of absolute instrength values (number of streamlines; Figure 4d) and describe their distribution in our revised manuscript on lines 280 - 283

„Next, we calculated each brain region's instrength by summing the incoming SC connection strengths and found that instrength is significantly right-skewed indicating that few regions have very large instrength (skewness = 1.33, $z = 13.48$, $p < 0.01$) while most have weaker connections (Figure 4d).“

We also expanded our discussion on other factors (neuron density, synaptic spine count, etc.) that could affect connection strength between cortical regions on lines 796 - 805

„We used the tractography-derived number of streamlines as a proxy for coupling strength between brain regions. Cortical traveling wave direction and EF patterns are likely subject to additional factors such as cortical gradients of neuron density, synaptic spine count, receptor distributions, myelin content, cortical thickness, and excitation-inhibition ratio.^{24,81–87} While some cortical gradients may vary on faster timescales large-scale SC gradients remain relatively stable over longer time periods such as many days, weeks or years.^{48,49} Thus, SC gradients could contribute to traveling wave direction across brain states, e.g. similar traveling wave directions have been observed in the alpha band during rest and memory-tasks.^{3,9} Other studies have shown that traveling wave direction changes rapidly in response to tasks.^{70,71,88}“

We thank the reviewer for their insightful comments and suggestions.

Reviewer #2

This study proposes a complementary mechanism for the generation of neural traveling waves using gradients of connection strength between neural networks, which is a different framework compared to frequency gradients suggested by other work. The researchers tested their hypothesis using a 2D oscillator model and human connectome data, which they described as indicating that the brain's connectivity shows patterns of gradients consistent with their model. Although there are some interesting ideas in this paper, I do not find this contribution to be compelling because the paper overstates the novelty of their model, inadequately describes the existing literature, and does not fully consider alternative hypotheses.

Response 11

We thank the reviewer for their insightful suggestions that have helped us to improve our manuscript.

Main comments:

The paper claims that in their simulations traveling waves only emerges in models where there is a high “instrength” gradient—I think this is quite misleading! Although the text Figure 1B claims there are no any stable traveling waves in the uniform instrength simulation, however the spatio-temporal patterns generated in this situation indeed look very much like actual electrophysiology recordings that is reported by earlier studies! There seem to be spatiotemporal wave propagation patterns that resemble spatially complex patterns of traveling waves. I think the authors really mean to state that these patterns are not linear plane waves... But nonetheless the demonstration of some type of traveling wave here seems to be important because it contradicts some of the author's claims. Relatedly, the paper should provide a clear quantitative definition for measuring traveling waves and identifying their robustness.

Response 12

We appreciate the reviewer's concerns and want to emphasize that we did **not** claim that traveling waves emerge in Kuramoto networks with non-uniform instrength gradients only. Instead, we proposed that instrength gradients systematically affect the propagation **direction** of traveling waves. In response, we have fully restructured our introduction to hopefully convey this better. We now also included a figure that explains the various mechanisms of wave emergence and wave direction:

Figure 3. Mechanisms of wave emergence and direction in weakly-coupled oscillator networks. *a* Mechanisms of wave emergence are based on neighborhood connectivity. Illustrated is a cortical network where circles and lines represent oscillators and their connections (top). The zoomed in graph (bottom) shows a chain of oscillators and how their connection strengths decrease with distance (blue line; this could also be connection probability), while their conduction delay increases (beige line). These mechanisms create neighborhood connectivity by emphasizing local synchronization between oscillators. *b* Mechanisms of wave direction include instrength gradients (top) and intrinsic frequency (bottom). Intrinsic frequency is the frequency at which the nodes oscillate isolated from the network (connections are removed). Here, the intrinsic frequency is equal across oscillators in the instrength gradient mechanism indicated by the same oscillator color and example activity (black sine curves). In contrast, the intrinsic frequency gradient mechanism is exemplified by a gradual increase of intrinsic frequency along the anterior-posterior axis illustrated by a gradual change of oscillator color. *c* Instrength is the sum of incoming connection strengths. This is illustrated for two oscillators by the addition of connections with different strength (thick and thin black lines are high and low strength connections, respectively). The instrength increases along the anterior-posterior axis for the instrength gradient mechanism, while it is equal for the intrinsic frequency gradient mechanism. *d* Effective frequency is the frequency assumed by oscillators connected within a network (black lines present). Here, effective frequency is again illustrated by the oscillator color and activity examples. The instrength gradient mechanism generates a smooth effective frequency gradient decreasing along the anterior-posterior axis (opposite of instrength gradient), while the intrinsic frequency gradient mechanism shows clusters with gradually increasing effective frequency along the anterior-posterior axis (same as intrinsic frequency gradient). *e* Traveling waves emerge in both networks from the mechanisms described in *a* and are directed by the instrength or intrinsic frequency gradient mechanisms. Traveling waves propagate from low to high instrength oscillators and from fast to slow intrinsic frequency oscillators as illustrated by the thick black arrows and the color gradients. Both mechanisms of wave direction can interact as shown in Figure 3.

The left side shows mechanisms that enable emergence of traveling waves, which are mechanisms that establish neighborhood connectivity. These mechanisms have been deeply studied in various papers (Ermentrout and Kopell 1984; Ko et al., 2004; Crook et al., 1997; Ko et al., 2007) and include distance-dependent connection strengths, probability, or delays. The mechanisms of wave direction include the intrinsic frequency gradient and instrength gradient. While the intrinsic frequency gradient mechanism has gotten some attention (to our knowledge, exclusively studied in 1D chains of weakly-coupled oscillators), the instrength gradient mechanism proposed in our manuscript is novel. Additionally, we show that the instrength gradient mechanism generates effective frequency gradients similar to empirical observations that were previously unexplained (Mahjoory et al., 2020 and Zhang et al., 2018).

We have referenced our new figure at several points in our manuscript and added to our revised introduction on lines 75 - 77

„While previous computational studies investigated the emergence of traveling waves (Figure 1a),^{27,28} we focus on their propagation direction (Figure 1b).“

We think these changes and our new figure in particular improved the clarity of our manuscript and we thank the reviewer for drawing our attention to this issue. We hope that these changes will allow the reader to appreciate our findings in the 2D

network model: The reviewer correctly observes complex wave patterns in the uniform instrength model in our **old** Figure 1b and Movie 2. We also described these waves in the caption of Figure 1b on lines 147-149 of our **old** manuscript

“The time series at the bottom shows traveling and rotating waves emerging in the control network model with a uniform instrength distribution (see Movie 2).”

We added a statement to this description to emphasize that these complex waves are highly variable in their direction across simulations on lines 196 - 197

*“The time series at the bottom shows traveling and rotating waves emerging in the control network model with a uniform instrength distribution (see Movie 2). **These complex waves are highly variable across simulations and do not show a systematic direction.**”*

Additionally, we appreciated the emergence of complex waves in the uniform instrength gradient model in our **old** main text on lines 126-135

*“Next, we built a control model with uniform instrength distribution to test if the observed propagation direction indeed resulted from the gradient. **We found that the control model produces waves 27% (11 – 48%) of the time across all simulations. Figure 1B (bottom; Movie 2) shows multiple waves emerging in the control model. ...**”*

However, the reviewer’s criticism has drawn our attention to a limitation of our analysis, namely, that we do not detect rotating waves. We mention this limitation on lines 160 - 164 of the revised manuscript

*„The control model produced waves **39%** of the time across all simulations (Figure S1). **Complex spatiotemporal patterns including traveling and rotating waves emerged in the control model (Figure 2b bottom; Movie 2). We did not quantify rotating waves in our analyses but visual inspection suggested they could make up a large proportion of time (for example Movie 2).**“*

We also added to our discussion that rotating waves appeared whenever instrength-gradients were absent or weak on lines 719 - 725

„Movies of our instrength-normalized cortical network model showed prominent rotating waves too. This was also the case in the 2D control network with instrength-normalized connectivity and when interacting IF and instrength gradients were balanced. We hypothesize that mechanisms underlying rotating waves take over whenever IF or instrength gradients are either weak or balanced. Our experiment on interacting IF and instrength gradients suggests that the brain could dynamically switch between different wave types by modulating IF gradients.“

Furthermore, the reviewer asked for a definition of traveling waves. We now provide a definition of cortical waves on the first line of our introduction on lines 40 - 42

„Cortical waves are signals of neuronal origin, measured e.g. with M/EEG, VSD, LFP, fMRI, that propagate systematically across space and time (e.g. plane waves, expanding waves, spiral waves, or impulse waves).“

And clearly define our operational definition of traveling waves further on lines 52 - 54

„In this work, we propose a mechanism that directs traveling waves - operationally defined as oscillations that show repeated (periodic) spatial propagation of their phase from sources to sinks.“

We thank the reviewer for their critical comments, they have significantly strengthened the clarity of our manuscript.

The paper claims that gradients of connectivity give rise to traveling waves. But the paper does not even show statistically that these gradients exist in individual subjects' data. To address this concern, the authors should compare pairs of regions and demonstrate the significance of gradient differences, particularly for longer distances between the regions. It is crucial to show the existence of this gradient at the single-subject level as well, considering the presence of traveling waves in individual subjects.

Response 13

We do **not** claim that instrength gradients give rise to traveling waves. However, we do claim that they affect their propagation direction. The reviewer raises the crucial point that we did not statistically assess instrength gradients with our current analyses. The reviewer suggests comparing pairwise differences between brain regions and we implemented this suggestion with a semivariogram, a method from geospatial statistics, that is frequently used to assess spatial dependence in data. Below, we plotted the semivariogram for the group connectome instrength:

Here, the blue line is the semivariogram of the group instrength. One can clearly see that the semivariance (Matheron method: average squared differences of instrength) increases with geodesic distance suggesting the presence of gradients. We

assessed the significance of the semivariogram using a bonferroni-corrected permutation test (n-permutations = 1000): The mean \pm SD of the permutations is shown in beige and distances at which the squared instrength differences vary significantly are indicated with asterisks. We further fit a spherical model to the instrength semivariogram to identify the so-called range, which should correspond to the spatial extent of gradients in the data. The estimated range was 110 mm (model fit was $r^2 = 0.998$), which corresponds well with the visually approximated gradient range of the instrength distribution (Figure 2C in old manuscript).

While this method is suitable to statistically assess spatial instrength gradients, we decided to use a computationally more efficient method that also allows us to identify gradients at multiple scales: Modal spectrum analysis, which can be understood as an extension of the Fourier analysis to our parcellated mesh. Briefly, we do an eigendecomposition of the Laplace-Beltrami operator, which gives us the geometric eigenmodes and their corresponding eigenvalues. Each mode is associated to a characteristic spatial frequency approximated by the square-root of the eigenvalues. Next, we compute the modal power spectrum by projecting the instrength onto the eigenmodes followed by squaring. The modal power spectrum of the group instrength is dominated by a spatial frequency of 5 m^{-1} , corresponding to a wavelength of ~ 200 mm. Thus, gradient troughs and peaks are separated by ~ 100 mm, which neatly coincides with the gradient range estimated visually and by the semivariogram. We assessed significance of power by comparing the original spectrum with random permutations (n-permutations = 10000 and significance level = 0.01; mean \pm -SD shown in beige below; dots indicate significant modes).

We applied the same method to identify gradients in the subject-level instrength distributions. A corresponding mean \pm -SD spectrum shows that the 5 m^{-1} mode dominates the subject-level spectra:

We further quantified which modes are significant on the subject-level and found that mostly low-frequency modes contributed significant power to the instrength patterns, e.g. the 5th mode (wavelength of ~200 mm) was significant in all 776 subjects:

We added these analyses to our section on instrength gradients in the human connectome on lines 284 - 294

„Visually, instrength resembled a spatial gradient increasing from temporal and parietal to frontal and occipital areas (Figure 4e). To quantify cortical instrength patterns, we used spectrospatial mode analysis – extending classical Fourier analysis to surface meshes (see Methods).⁴⁷ Figure 4f shows six modes ordered from higher to lower spatial wavelengths or lower to higher spatial frequency. Projecting the instrength onto each mode quantifies their contribution to spatial instrength patterns. We found that low-frequency modes significantly contributed to the group-averaged instrength pattern (Figure 4g and f). Mode 5 dominated the instrength pattern with a spatial wavelength of 202 mm matching the visually observed gradients (Figure 4e). We confirmed that low-frequency modes also dominated individual subjects’ instrength patterns (Figure 4h and i). Remarkably, mode 5 contributed significantly to all subjects’ instrength patterns (Figure 4i).“

Figure 4. The human connectome hosts instrength gradients. **a** Average structural connectivity based on 1000 cortical regions of the Schaefer parcellation (\log_{10} weights threshold at 90th percentile). The connection weights were estimated from 776 subjects of the Human Connectome Project. **b** The log-transformed connection weights are negatively correlated with the fiber lengths. **c** The euclidean distances between connected regions are positively correlated with fiber lengths. **d** The instrength distribution across all regions of the parcellation is right-skewed. **e** Instrength derived from the average structural connectivity projected onto the cortical regions of the Schaefer atlas. An instrength gradient increasing from temporal and parietal areas to frontal and occipital areas is visible. See also Supplementary Figure 3 for instrength gradients found in other cohorts and parcellations. **f** Modes that significantly explained the group-averaged instrength pattern in **e** and **g**. These modes were computed by decomposing the Laplace-Beltrami operator of the parcellated mesh and used for modal spectrospatial analysis (see Methods). Note that the spatial wavelength increases with the modes. **g** Modal power spectrum of the group-averaged instrength pattern (blue). Statistical significance was assessed with permutation tests ($n = 10,000$; beige line represents mean modal power of all permutations and shaded area the respective standard deviation; see Methods). **h** Average modal power of subject-level power spectra (thick blue line is the mean and the shaded area represents the standard deviation). **i** Percentage of subjects for which significant modes were identified. The inset shows the first 20 modes in more detail.

Furthermore, we conducted the same analysis on the control cohorts and parcellations. We were able to replicate our results in two independent cohorts (eNKI and Griffa et al.) and three different parcellations (Random, Lausanne, and Schaefer400 parcellations): All controls showed that low frequency modes dominated the power spectra. Thus, instrength gradients with a range ~ 100 -150 mm were present across cohorts and parcellations (see Supplementary Figure 3). We mention these results on lines 294 - 297

„We validated instrength gradients in distinct cohorts and parcellations. The three validation sets showed instrength gradients (Figure S3a,e,i) and low-frequency modes significantly dominated their modal spectra on the group- and subject-level (Figures S3).“

We also added a corresponding methods section on lines 898 - 914

“Identification of instrength gradients in the human connectome

We calculated the instrength for each brain region by summing the incoming connection weights of the SC matrix. We identified instrength gradients across the cortex with modal spectrospatial analysis, which extends Fourier analysis to meshes with arbitrary topology.⁴⁷ First, we decomposed the Laplace-Beltrami operator to get a hundred eigenmodes and eigenvalues of a mesh that characterizes the topology between brain parcels (see Constructing a mesh for discrete operators). Next, we calculated the power spectrum by projecting the spatial instrength distribution onto the eigenmodes and normalizing by the total power. Spatial frequencies corresponding to each eigenmode were approximated by $\sqrt{\lambda/2\pi}$, where λ are the eigenvalues.⁴⁷ We detected instrength gradients with permutation tests by randomly shuffling the instrength distribution 10,000 times and computing the corresponding spectra. For each mode, the fraction of spectral powers that exceeded the original power corresponds to the p-value. We identified significant modes by comparing the Bonferroni-corrected p-values to a significance level of $\alpha = 0.01$. We applied this analysis to group- and subject-level instrength distributions of the left hemisphere of the Schaefer parcellation and all control connectomes (eNKI cohort had average connectome only; see Figure S3).“

And further discussed our observations in the discussion on lines 676 - 692

*„After demonstrating that instrength gradients **direct** traveling waves in a 2D **network** model, we explored **if** gradients **exist** in the human connectome and found that instrength*

increased from temporal and parietal towards frontal and occipital regions. In contrast to previous studies,⁶⁵⁻⁶⁷ we quantified instrength patterns statistically with spectrospatial mode analysis and found similar gradients across different cohorts and parcellations. We only studied SCs with similar-sized parcels, high spatial resolution (> 400 regions), and across many subjects (> 80 subjects) to ensure high quality estimates. While our instrength patterns were relatively consistent compared to other studies,⁶⁵⁻⁶⁷ differing processing pipelines could explain variations between SCs. For example, normalizing the streamline count between brain regions using either brain region size or fiber lengths can impact instrength patterns.⁶⁵ Additional processing such as thresholding individual or group averaged SCs can further affect instrength patterns.⁴⁴ Gajwani and colleagues⁶⁷ extensively studied how 1760 distinct processing pipelines for diffusion MRI affected instrength patterns. They found that different tractography algorithms and parcellations had the largest influence on instrength topographies. Developing methods and processing pipelines that reliably estimate connection strengths will help understanding how instrength gradients direct traveling waves and shape EF gradients.”

The reviewer’s suggestions have tremendously improved the strength of our work.

I think the paper overstates their claims of novelty regarding their modeling. It is known in physics that traveling waves are a general form of solution for Kuramoto models. Previous studies have shown that traveling waves can occur even in the absence of instrength gradients, which makes me concerned about the authors’ claim that traveling waves specifically emerge in Kuramoto models with non-uniform instrength spatial distribution. For example, see Iatsenko et al., 2013. At the very least, the authors should emphasize why the emergence of traveling waves from a 2D Kuramoto model is surprising or distinct from previous findings.

Response 14

We appreciate the reviewer’s concerns and want to emphasize that we do **not** claim that traveling waves emerge in Kuramoto networks with non-uniform instrength gradients only. Instead, we propose that instrength gradients affect the propagation **direction** of traveling waves. To make this point clear, we made changes to our manuscript explained in our **Response 12**.

I was confused by the author’s claims on line 67 that there were no observations of frequency gradients in earlier papers. The authors themselves cited at least two papers mentioning such ideas (Zhang et al and Mahjoury et al). Doesn’t this undermine the entire motivation for this project?

Response 15

We agree with the reviewer and think the confusion arises from our insufficient definitions of **intrinsic** and **effective** frequency. On lines 66-70 (old manuscript), we say

*“While there is experimental evidence for **intrinsic frequency** gradients in the nervous system of non-human animals, evidence for such gradients across the human cortex is lacking. In contrast, large-scale spatial patterns of **effective frequency** are a robust finding across species. In particular, recent studies have found **effective frequency** gradients across specific human brain structures as well as the entire cortex.”*

In our old manuscript we introduced these terms in a confusing nested structure on lines 71-75 as

*“Studies of chains of weakly-coupled oscillators demonstrated that traveling waves propagate along **intrinsic frequency** gradients, **the frequency assumed by oscillators isolated from the network. Once coupled in a network**, oscillators form one or several clusters with similar **effective frequency** and emerging traveling waves propagate from oscillators with high to those with low **intrinsic frequency**.”*

We fully restructured our introduction to make the difference between intrinsic and effective frequency (as well as other key concepts) clearer. The paragraph on the intrinsic frequency mechanism on lines 55 - 66 now says

*„Early theoretical work has shown that distance-dependent **connectivity or time delays (Figure 1a)** give rise to traveling waves in weakly-coupled oscillator networks, a frequently used system to study synchronization phenomena.^{14–16} Further simulation studies demonstrated that traveling waves **can be directed by intrinsic frequency (IF) gradients, where IF is the frequency of an oscillatory unit disconnected from a network (Figure 1b).**¹⁵ Once coupled in a network, traveling waves propagate from high to low **IF oscillators (Figure 1e)**. The **IF gradient mechanism has been proposed to explain the propagation direction of cortical traveling waves in experimental recordings^{3,17} but we lack evidence for IF gradients across the human cortex due to methodological challenges. In non-human animals, IF gradients have been measured invasively by slicing neural tissue into disconnected self-oscillatory units.**^{18,19”}*

We defined the effective frequency in a new paragraph on lines 78 - 90

*“Zhang and colleagues have found that cortical traveling wave direction correlates with **effective frequency (EF) gradients.**³ We define EF as the oscillation frequency that emerges when a unit (e.g. cortical region, neuronal population, or weakly-coupled oscillator) is connected to a network (**Figure 1d**); this contrasts the self-generated IF of a disconnected unit introduced earlier. EF is the oscillation frequency that we typically estimate in MEG, EEG, or ECoG from the connected cortical network in humans. Zhang and colleagues found that alpha and theta traveling waves measured by ECoG propagated from high to low EF regions but whether this association is causal or correlative remains unknown. Other studies have found large-scale EF gradients across the human cortex but did not investigate traveling waves.^{29–33} Previous theoretical studies have shown that increasing instrength decreases an oscillator’s EF in a weakly-coupled oscillator network.^{34,35} Thus, we hypothesized that instrength gradients could systematically suppress EFs thereby explaining experimentally observed large-scale EF gradients.”*

Furthermore, these concepts are illustrated and explained in our new introduction Figure 1 that we already presented in **Response 12**. Additionally, we introduced acronyms for intrinsic frequency (IF) and effective frequency (EF), which hopefully allows clear distinction between the two concepts. We thank the reviewer for pointing out these seemingly contradictory statements, which made us realize that our definitions did not have the saliency they deserve. We think that the above changes drastically improved the clarity of our manuscript.

The paper is hard to understand because many key terms are not explained or not used in standard ways, beginning in the abstract. First, the term "structural connectivity gradient" used in line 15 is unclear and requires clarification—a precise definition should be provided. Additionally, the abstract contains redundancy and lacks a clear statement regarding the problem addressed or the novelty of the study.

Response 16

We understand the reviewer's concern about the clarity of our abstract and hope that our revised version addresses this sufficiently:

„Traveling waves and neural oscillation frequency gradients are pervasive in the human cortex. While the direction of traveling waves has been linked to brain function and dysfunction, the factors that determine this direction remain elusive. We hypothesized that structural connectivity instrength gradients — defined as the gradually varying sum of incoming connection strengths across the cortex — could shape both traveling wave direction and frequency gradients. We confirm the presence of instrength gradients in the human connectome across diverse cohorts and parcellations. Using a cortical network model, we demonstrate how these instrength gradients direct traveling waves and shape frequency gradients. Our model fits resting-state MEG functional connectivity best in a regime where instrength-directed traveling waves and frequency gradients emerge. We further show how structural subnetworks of the human connectome generate opposing wave directions and frequency gradients observed in the alpha and beta bands. Our findings suggest that structural connectivity instrength gradients affect both traveling wave direction and frequency gradients.“

Moreover, the introduction figure presented in **Response 12** visualizes many key concepts of our manuscript and we hope that this makes our manuscript an easier read.

Furthermore, the use of the term "in-strength" is problematic. Is this presumed to refer to the weighted in-degree, considering the synaptic strengths, rather than just the counts? However, this assumption should be explicitly explained and defined to avoid ambiguity. Overall, the abstract needs improvement in terms of clarity, conciseness, and providing a more compelling overview of the research.

Response 17

The reviewer raises a crucial point and we now clearly state which measure we used to identify instrength gradients in the human connectome in our revised introduction on lines 67 - 71

*“While we speculate that cortical IF gradients exist in humans, we propose an additional mechanism that could affect the direction of traveling waves and is accessible through non-invasive tractography, namely structural connectivity (SC) instrength gradients — the sum of incoming connection strengths (with number of streamlines as proxy for connection strength; also known as weighted in-degree; **Figure 1c**).”*

And in our results on lines 271 - 272

„We used the number of streamlines as a proxy for connection strength between cortical areas.“

Furthermore, we addressed this limitation in our revised discussion on lines 796 - 800

“We used the tractography-derived number of streamlines as a proxy for coupling strength between brain regions. Cortical traveling wave direction and EF patterns are likely subject to additional factors such as cortical gradients of neuron density, synaptic spine count, receptor distributions, myelin content, cortical thickness, and excitation-inhibition ratio.^{24,80–86”}

The overlap of the paper's findings in relation to the study by Breakspear's group in Nature Communications is questionable. The discussion paragraph (line 452) fails to elucidate how the current results contribute to a better understanding or provide additional insights beyond that work. The paper needs to explain how this paper goes beyond this paper and others)—such clarification, the significance of the research remains uncertain.

Response 18

Roberts et al. (2019) focused on metastability of waves in a computational network model. In contrast, we focus on mechanisms that determine traveling wave direction and shape effective frequency patterns. While Roberts et al. present how sinks and sources of traveling waves were spatially distributed in their network (see their Figure 7), they do not present a mechanism that determines this spatial arrangement. They correlated node strength with the fraction of “visits” to sinks or sources but only found very weak correlations (sources: $r = 0.098$, $p = 0.026$; sinks: $r = -0.025$, $p = 0.57$). Next, they split all nodes by strength in hubs, feeders, and non-hubs and find that sink visits are significantly aligned with feeder nodes (mid strength) while source visits are significantly aligned with hub nodes (high strength). They themselves note that there is substantial overlap between these three categories; we agree, the effect is unclear. We believe they did not find strong correlations with node strength because they instrength-normalized their

connectome. Our work shows that instrength gradients exist in several cohorts and parcellations (see Figures 4 and Supplementary Figure 3).

Besides the above differences, Roberts et al. do not investigate effective frequency at all. Our work presents the first mechanism that explains cortical effective frequency gradients observed in empirical findings (Mahjoory et al., 2020; Zhang et al., 2018) along with traveling wave direction, thereby advancing the growing cortical wave and gradient literature.

There are few other studies that investigated mechanisms of traveling wave directions. To our knowledge, all of these studies investigated the IF gradient mechanism in a simple 1D chain of weakly-coupled oscillators (Diamant et al., 1979; Sarna et al., 1971; Robertson-Dunn and Linkens, 1974; Ermentrout and Kopell, 1984; Zhang et al., 2019). We are not aware of any evidence that IF gradients exist in the human cortex (but we suspect they do). In contrast, our study shows that cortical instrength-gradients exist in the human connectome estimated from independent cohorts and parcellations. Our study is the first to investigate if instrength-gradients direct traveling waves in a computational model using an empirically derived connectome. Furthermore, we show that our findings are consistent with resting-state MEG FC and EF.

We added a paragraph to our discussion that elaborates on the differences between Roberts et al. and our study on lines 714 - 719

„While Roberts and colleagues²⁷ focused on metastability of cortical waves, they also analyzed if sink and source locations were associated with instrength in a cortical model with empirical SC. They found that instrength correlated weakly with source occurrences ($r = 0.09$) but not with sinks. Their results might be due to eliminating instrength patterns by instrength-normalizing their SC.“

We hope that we could convince the reviewer that our study goes beyond the highly important and original work by Roberts et al.

Minor comments:

[116] This parts of the text discussing 100-100% ranges are confusing.

Response 19

We indicated the median and interquartile range (IQR) for the proportion of timepoints across all simulations that exhibit traveling waves. These proportions are not normally distributed and thus we found the median and IQR more informative than e.g. the mean +- SD. We see the reviewer's point that the way we reported those measures is confusing. For the revised manuscript, we have created Supplementary Figures 1a and 4a that show the proportions of waves and directed-waves in violinplots for all models. Within the main text, we now only report the median proportions across timepoints, simulations, and hemispheres for each model. We think this change has improved the readability of this section and made it more concise.

Is there never any noise in these models?

The initial phases of all models were uniformly random but we did not add any extra noise. In response to the reviewer's comment, we now included a control model with additive gaussian noise and another control model with random gaussian IF dispersion. We added the statistical findings and visualizations for these models to the Supplementary Figure 4e-j and mention our results on lines 399 – 401

„Further, our results were robust in cortical network models with added noise (Supplementary Figure 4 e-g; Movie 10) and random IF dispersion (Supplementary Figure 4 h-j; Movie 11).“

These 100-100% ranges are not described in a conventional way and seem to be biologically implausible.

We now describe these ranges in violinplots (see Supplementary Figures 1a and 4a). Our 2D network model that the reviewer referenced is a proof-of-principle that traveling waves follow an artificially created instrength-gradient. We did not tweak the parameters to achieve biological realism. Generally, we think that traveling waves are likely prominent during oscillatory activity, however, detecting traveling waves is notoriously difficult as they are obscured by noise and methodological limitations. Nevertheless, Townsend et al. (2015) found that roughly 75% of delta activity was organized as cortical waves in anesthetized marmoset; Muller et al. (2016) observed that 50.8% of 41,860 spindle oscillation cycles formed rotating waves in ECoG recordings of humans; Zhang et al. (2018) reported that alpha and theta traveling waves were present in 61% (median) of single trials. Thus, despite challenges in detecting traveling waves, studies found they are prevalent in cortical dynamics.

[191]: What does the word “should” mean here? There should be a citation to support this point, and in any case this type of claim is not appropriate for a caption. There are lots of situations where traveling waves might not emerge even with distant dependent connectivity and I think getting the details right is important here.

Response 20

We agree with the reviewer and have removed this claim from the figure caption.

[Figure 2]: The average instrength plot is potentially interesting but it requires a statistical test to show if the observed gradient pattern is significant, especially at the single subject level.

Response 21

We have now implemented group- and subject-level statistics to identify instrength gradients. Please refer to our **Response 13**.

Reviewer #3

The authors demonstrate that traveling waves (TW) propagate along a gradient of the local sums of incoming connection strengths (i.e., instrength gradients) in a simulated weakly coupled oscillator network. They further showed that connectivity profiles in the human connectome are distance dependent and display instrength gradients that would allow for the emergence of directed traveling waves. When fitting cortical network models to empirically derived functional connectivity profiles, the authors found strong correlations when guided TWs emerged in the model. These models also produced smooth effective frequency gradients, leading the authors to suggest that functional connectivity could be generated via TWs following instrength gradients, which simultaneously generate resting-state frequency gradients.

The paper is thorough, clearly written and potentially of interest for a broad audience. The accompanying code seems complete and runs out of the box and all data necessary to reproduce the results is publicly available. I have some concerns and remarks, listed below in no particular order, that could, however, strengthen and clarify the manuscript:

Response 22

We thank the reviewer for this accurate and favorable assessment of our study and address each concern below.

1. The concepts of “instrength gradients” is very prominent in the manuscript, but it is only briefly defined without further explanation. The paper would be easier to follow after a careful introduction to instrength gradients early on in the text.

Response 23

We thank the reviewer for pointing this out and agree that we should have introduced cortical gradients and instrength gradients in particular more carefully. We now restructured our introduction with a new paragraph introducing the concept of cortical gradients:

*„While we speculate that cortical IF gradients exist in humans, we propose an additional mechanism that could affect the direction of traveling waves and is accessible through non-invasive tractography, namely structural connectivity (SC) instrength gradients – the sum of incoming connection strengths (with number of streamlines as proxy for connection strength; also known as weighted in-degree; **Figure 1c**). Here, SC instrength gradually changes across cortical space - similar to other cortical gradients such as functional connectivity²⁰, gene expression,²¹ receptor distributions,²² myelin content,²³ cortical thickness,²⁴ or synaptic spine density.^{25,26} We postulate that this instrength gradient directs traveling waves from low to high instrength cortical regions. While previous computational studies investigated the emergence of traveling waves (**Figure 1a**),^{27,28} we focus on their propagation direction (**Figure 1b**).“*

Furthermore, we now included an introduction figure that presents the key concepts of our manuscript including instrength and effective frequency gradients:

Figure 5. Mechanisms of wave emergence and direction in weakly-coupled oscillator networks. *a* Mechanisms of wave emergence are based on neighborhood connectivity. Illustrated is a cortical network where circles and lines represent oscillators and their connections (top). The zoomed in graph (bottom) shows a chain of oscillators and how their connection strengths decrease with distance (blue line; this could also be connection probability), while their conduction delay increases (beige line). These mechanisms create neighborhood connectivity by emphasizing local synchronization between oscillators. *b* Mechanisms of wave direction include instrength gradients (top) and intrinsic frequency (bottom). Intrinsic frequency is the frequency at which the nodes oscillate isolated from the network (connections are removed). Here, the intrinsic frequency is equal across oscillators in the instrength gradient mechanism indicated by the same oscillator color and example activity (black sine curves). In contrast, the intrinsic frequency gradient mechanism is exemplified by a gradual increase of intrinsic frequency along the anterior-posterior axis illustrated by a gradual change of oscillator color. *c* Instrength is the sum of incoming connection strengths. This is illustrated for two oscillators by the addition of connections with different strength (thick and thin black lines are high and low strength connections, respectively). The instrength increases along the anterior-posterior axis for the instrength gradient mechanism, while it is equal for the intrinsic frequency gradient mechanism. *d* Effective frequency is the frequency assumed by oscillators connected within a network (black lines present). Here, effective frequency is again illustrated by the oscillator color and activity examples. The instrength gradient mechanism generates a smooth effective frequency gradient decreasing along the anterior-posterior axis (opposite of instrength gradient), while the intrinsic frequency gradient mechanism shows clusters with gradually increasing effective frequency along the anterior-posterior axis (same as intrinsic frequency gradient). *e* Traveling waves emerge in both networks from the mechanisms described in *a* and are directed by the instrength or intrinsic frequency gradient mechanisms. Traveling waves propagate from low to high instrength oscillators and from fast to slow intrinsic frequency oscillators as illustrated by the thick black arrows and the color gradients. Both mechanisms of wave direction can interact as shown in Figure 3.

We think the reviewer's suggestion has greatly improved the clarity of our manuscript.

2. The authors report that the human connectome hosts instrength gradients and that traveling waves propagate along those gradients. Given that gradients found in the connectome are structural in nature and (relatively) stable across time, can the authors relate their findings to those of task based experimental work that found traveling wave direction to depend on task demands (e.g., Alamia, Mohan, etc.)?

Response 24

The reviewer raises a crucial distinction that we clarified in our revised manuscript on lines 796 - 815

"We used the tractography-derived number of streamlines as a proxy for coupling strength between brain regions. Cortical traveling wave direction and EF patterns are likely subject to additional factors such as cortical gradients of neuron density, synaptic spine count, receptor

distributions, myelin content, cortical thickness, and excitation-inhibition ratio.^{24,81–87} While some cortical gradients may vary on faster timescales large-scale SC gradients remain relatively stable over longer time periods such as many days, weeks or years.^{48,49} Thus, SC gradients could contribute to traveling wave direction across brain states, e.g. similar traveling wave directions have been observed in the alpha band during rest and memory-tasks.^{3,9} Other studies have shown that traveling wave direction changes rapidly in response to tasks.^{70,71,88} We hypothesize that external or self-generated stimuli dynamically affect traveling waves by modulating IF.⁸⁹ For example, a stimulus arriving in the visual cortex could accelerate local oscillations⁹⁰ that interact with large-scale stable coupling and IF gradients to achieve cortex-wide processing through traveling waves. Alternatively, stimuli could induce large-scale IF gradients to direct cortical traveling waves, this could be achieved through thalamocortical loops preparing the cortex for incoming stimuli. We explored this latter mechanism in our 2D network model and found that stable instrength and dynamic IF gradients could cooperatively direct traveling waves. Alternatively, inter-regional coupling strength could be dynamically controlled – for instance by modulating the long-range excitation-inhibition ratio.⁸⁶ To explain behavior, deeply understanding stable and dynamic coupling and IF gradients will be necessary.”

As mentioned in this paragraph, we now also investigate the interaction between IF and instrength gradients in our 2D network model. This experiment elucidates how stable or dynamic (e.g. task-dependent) large-scale IF gradients interact with a fixed instrength gradient. These findings are presented on lines 207 - 231

“Previous studies proposed that IF gradients direct traveling waves in the human cortex.^{3,18} We hypothesize that instrength gradients contribute to traveling wave direction and interact with IF gradients. To investigate this, we generated superimposed instrength and IF gradients with the exact same shape from a gradient template (**Figure 3a**). Notably, the two mechanisms generate opposing wave directions from low to high instrength and high to low IF,¹⁵ respectively. We scaled the IF gradient from zero (IF is 10 Hz for all oscillators) to range from 8.5 to 11.5 Hz (gradient scaling = 1.5) with a superimposed fixed instrength gradient. The instrength gradient fully directed traveling waves without the IF gradient as assessed by gradient template and flow potential correlation (**Figure 3b** top row). This is also reflected in the average flow potential and illustrated for an example wave timeseries (**Figure 3c** and **e** top). The instrength gradient’s influence weakened with increasing IF gradient scaling until both balanced each other out (gradient scaling @ 0.75; IF gradient ranging from 9.25 to 10.75 Hz; **Figure 3b**). At this point, simulated activity varied: we observed spiral waves, plane waves, traveling waves, and full synchrony (**Figure 3e** middle). The average flow potential suggested that waves source from the periphery and sink into the network center (**Figure 3c**) but individual simulations’ flow potentials varied reflecting the diversity of waves observed. A further increasing IF gradient switched the wave direction from instrength- to IF-directed (**Figure 3b**) as shown by the average flow potential (**Figure 3c**) and an example timeseries (**Figure 3e**).

Earlier, we showed that instrength gradients shape EF patterns (**Figure 2e**). We next investigated if instrength and IF gradients could cooperatively shape EF patterns. We found that the correlation between gradient template and EF switched from strongly negative to positive as we increased the IF gradient scaling (**Figure 3b** and **d**). This suggests that instrength and IF gradients together shaped EF gradients in the 2D network model.”

Figure 6. Instrength and intrinsic frequency gradients interact to determine wave direction and shape effective frequency. **a** Gradient template used to create scaled intrinsic frequency gradients (0 to 1.5) and the instrength gradient. The intrinsic frequency gradients entered the 2D network model through the intrinsic frequency term ω_i , while the instrength gradient was used to scale the connectivity matrix before it entered through the local coupling term a_{ij} . **b** The top figure shows how the gradient template and flow potential correlated depending on the intrinsic frequency gradient scaling. The bottom figure shows how the gradient template correlated with the effective frequency pattern. Both graphs are based on 100 randomly initialized simulations of 10 s duration (1 s transient removed) per gradient scaling. The thick blue lines show the mean correlation and the light blue shaded areas show the standard deviation across simulations. **c** Average flow potentials for three distinct gradient scalings describe the average propagation of traveling waves from higher to lower potentials. **d** Effective frequency maps show how the gradient scaling affects effective frequency patterns. **e** Example timeseries of activity (cosine of instantaneous phases) emerging at three distinct gradient scalings are shown. The top row shows a traveling wave following the instrength gradient. The middle row shows a spiral wave that emerged when instrength and intrinsic frequency gradients were balanced. The bottom row shows a traveling wave that follows the intrinsic frequency gradient.

and in the methods on lines 976 - 988

*“We investigated if instrength and IF gradients interact to direct traveling waves. To do so, we built on the 2D instrength gradient model introduced earlier. We added an IF gradient generated from the same gradient template $g(x_i)$ that we used to construct the instrength gradient (see **Figure 3a**). We chose to scale the IF gradient from 0 to 1.5 (IF ranges from 8.5 to 11.5 Hz at maximum scaling) in steps of 0.05 (IF range increment of 0.1 Hz) superimposed on a fixed instrength gradient and simulated 100 randomly initialized models per scaling factor. We investigated gradient template – flow potential Spearman correlation for all scaling factors. In this model, a negative correlation indicates instrength-directed waves, while a positive correlation indicates IF-directed waves. We also investigated the gradient template – EF Spearman correlation, where a negative correlation indicates that EF patterns are shaped by the instrength gradient, while a positive correlation means that they are shaped by the IF gradient.”*

3. Line 137: a p-value of 0.315 is reported as significant. Is that a typo?

Response 25

We thank the reviewer for pointing this out. We presented this in a misleading way and corrected this statement in our revised manuscript on lines 169 - 171

„The average flow potentials and thus traveling wave directions of the gradient and uniform models were not significantly correlated ($r = 0.55$, $p = 0.316$).“

The numbers are slightly different because we re-ran our stochastic simulations, analyses and improved the statistics (we now use shift-spin permutation tests that preserve the spatial autocorrelation and other characteristics of the 2D network model activity; see “Spin permutation tests” for details).

4. In the section on control models (line 222ff), the authors test a model that couples connection strength to Euclidean distance while destroying original instrength gradients and show that this model produces waves that do not follow the original instrength gradients. With regards to other control models they also talk of “distance dependent conduction delays.” First: Is it correct to assume that distance here also refers to Euclidian distance?

Response 26

Conduction delays in the cortical network model and controls were calculated by dividing the fiber lengths by a given conduction speed. The control cortical model with synthesized connections based on the euclidean distance and the 2D network models used euclidean distance-dependent time delays. Notably, fiber length and euclidean distance were strongly correlated ($r = 0.8$, $p < 0.01$) and thus fiber length-dependent delays correlate strongly with euclidean distance-dependent delays. We thank the reviewer for pointing this out and realized that it is worth making these relationships more explicit. In response, we explained this in the revised manuscript on lines 326 - 330

„... conduction delays calculated by dividing the tractography-derived average fiber lengths with a biologically realistic conduction speed of 3 m/s (see parameter exploration for a broader range of conduction speeds). Resulting conduction delays depended on distance because fiber length depends on euclidean-distance (Figure 4c).“

and added Figure 4c that shows the fiber length - euclidean distance relationship:

We further specified whether fiber length or euclidean distance was used to estimate time delays at multiple places in the revised manuscript.

Furthermore, the reviewer’s comment made us realize that we should address how the fiber length – connection strength relationship results in euclidean distance-dependent connection strengths, which are crucial for traveling waves to emerge. We now explain this relationship in our manuscript on lines 277 - 278

“Fiber lengths increased with increasing euclidean distance while connection strengths decreased (Figure 4c and S5b).”

We also adapted the Methods section in the revised manuscript on lines 995 - 998

„We calculated the time delays between cortical regions by dividing the empirically estimated fiber lengths by the conduction speed v . This also results in distance-dependent conduction delays because fiber length and euclidean distance are strongly correlated ($r = 0.8$, $p < 0.01$; see Figure 4b,c and S5).“

We now also verify that the euclidean distance – connection strength is destroyed in the control model with randomly shuffled connection strengths on lines 1005 - 1011 of the revised manuscript

„This model preserved the network topology and conduction delays while destroying the fiber length – connection strength relationship as well as the instrength gradient. Notably, random shuffling of the connection strengths also destroys the euclidean distance – connection strength relationship. We verified this by fitting the euclidean distance – shuffled connection strength relationship with a linear, exponential, and power law model (coefficients of determination: coefficients of determination: $r_{linear}^2 < 0.001$, $r_{exp}^2 < 0.001$, $r_{power}^2 < 0.001$; see next section for further details).“

We thank the reviewer for their comment and think that the resulting changes improved our manuscript.

And second: Could the authors comment on how their results relate to the findings of a recent publication by Pang et al (Nature, 2023), in which the authors find that

geometric eigenmodes derived from the cortical surface explain a large fraction of the variance in functional connectivity?

Response 27

Pang et al. (2023) elegantly showed how the cortical geometry constrains fMRI activity. We now discuss their findings on lines 704 - 713

“Pang and colleagues⁶⁹ showed that a neural field model encoding the cortical geometry generated wave-like activity and explained fMRI functional connectivity. They further found that geometric eigenmodes explained fMRI variability better compared to connectome eigenmodes. This difference nearly vanished when they used a stochastic exponential distance rule to synthesize a connectome. Intriguingly, an instrength gradient increasing from frontal and temporal to parietal areas emerged in our control model with deterministic exponential distance rule. We found that this instrength gradient reliably directed traveling waves as predicted by our hypothesis. Future research could address how the cortical geometry constrains instrength patterns, traveling wave direction, and EF gradients.”

5. Lines 66-70 read as somewhat contradictory. The authors first state that evidence for intrinsic frequency gradients in humans is lacking, to then contrast with findings in animals. My hangup is with the “in particular,” followed by a bunch of evidence in human cortex, that comes after. Could this section be rephrased for clarity?

Response 28

We agree with the reviewer and think the confusion arises from our insufficient definitions of **intrinsic** and **effective** frequency. On lines 66-70 (old manuscript), we said

*“While there is experimental evidence for **intrinsic frequency** gradients in the nervous system of non-human animals, evidence for such gradients across the human cortex is lacking. In contrast, large-scale spatial patterns of **effective frequency** are a robust finding across species. In particular, recent studies have found **effective frequency** gradients across specific human brain structures as well as the entire cortex.”*

In our old manuscript we introduced these terms in a confusing nested structure on lines 71-75 as

*“Studies of chains of weakly-coupled oscillators demonstrated that traveling waves propagate along **intrinsic frequency** gradients, **the frequency assumed by oscillators isolated from the network. Once coupled in a network**, oscillators form one or several clusters with similar **effective frequency** and emerging traveling waves propagate from oscillators with high to those with low **intrinsic frequency**.”*

We fully restructured our introduction to make the difference between intrinsic and effective frequency (as well as other key concepts) clearer. We changed the paragraph on the intrinsic frequency mechanism on lines 55 - 66

„Early theoretical work has shown that distance-dependent *connectivity or time delays (Figure 1a)* give rise to traveling waves in weakly-coupled oscillator networks, a frequently used system to study synchronization phenomena.^{14–16} *Further simulation studies demonstrated that traveling waves can be directed by intrinsic frequency (IF) gradients, where IF is the frequency of an oscillatory unit disconnected from a network (Figure 1b).*¹⁵ *Once coupled in a network, traveling waves propagate from high to low IF oscillators (Figure 1e). The IF gradient mechanism has been proposed to explain the propagation direction of cortical traveling waves in experimental recordings^{3,17} but we lack evidence for IF gradients across the human cortex due to methodological challenges. In non-human animals, IF gradients have been measured invasively by slicing neural tissue into disconnected self-oscillatory units.*^{18,19}”

We defined the effective frequency in a new paragraph on lines 78 - 90

“Zhang and colleagues have found that cortical traveling wave direction correlates with effective frequency (EF) gradients.³ We define EF as the oscillation frequency that emerges when a unit (e.g. cortical region, neuronal population, or weakly-coupled oscillator) is connected to a network (**Figure 1d**); this contrasts the self-generated IF of a disconnected unit introduced earlier. EF is the oscillation frequency that we typically estimate in MEG, EEG, or ECoG from the connected cortical network in humans. Zhang and colleagues found that alpha and theta traveling waves measured by ECoG propagated from high to low EF regions but whether this association is causal or correlative remains unknown. Other studies have found large-scale EF gradients across the human cortex but did not investigate traveling waves.^{29–33} Previous theoretical studies have shown that increasing instrength decreases an oscillator’s EF in a weakly-coupled oscillator network.^{34,35} Thus, we hypothesized that instrength gradients could systematically suppress EFs thereby explaining experimentally observed large-scale EF gradients.”

Additionally, we introduced acronyms for intrinsic frequency (IF) and effective frequency (EF), which hopefully allows clear distinction between the concepts while reading. Also, our new introduction figure illustrates both concepts visually (see Figure 1). We thank the reviewer for pointing out these seemingly contradictory statements, which made us realize that we did not define those concepts clearly. We think that the above changes improved the clarity of our manuscript.

6. Line 340 “effective frequency:” Could the authors please provide a brief definition there?

Response 29

We realized that the crucial difference between effective and intrinsic frequency was ill-defined in our old manuscript. We made a couple of changes as outlined in our **Response 28** to make this clearer and added an introduction figure that clarifies key concepts.

7. Line 399ff: “This suggests that FC could be coordinated by cortical traveling

waves following structural connectivity gradients that simultaneously generate resting-state frequency gradients.” Could the authors please comment on how they get to the conclusion that instrength gradients independently lead to traveling waves and frequency gradients, and not e.g., that instrength gradients lead to frequency gradients, which in turn lead to traveling waves, especially considering that they state in lines 516ff: “...We hypothesize that external or self-generated stimuli affect traveling waves by changing oscillation frequency patterns.”

Response 30

We did not want to suggest a specific causal chain and simply proposed that instrength gradients affect traveling wave direction and effective frequency patterns. We reasoned that effective frequency gradients do not cause but are strongly correlated with wave direction because we observed parameter combinations where the effective frequency is almost homogeneous throughout the cortical network model. To simplify the problem, we investigated the 2D network model and attempted to find parameters that resulted in completely homogeneous effective frequency. The effective frequency is less likely to drive wave direction if such parameters exist. Interestingly, we found a parameter combination where effective frequency was not only homogeneous but orthogonal to or opposing the instrength gradient and traveling wave direction. Thus, we believe that instrength gradients direct traveling waves and shape effective frequency patterns under many parameter combinations. We now mention this experiment in our new section on instrength vs. IF gradients on lines 232 - 242

„Do instrength gradients directly guide traveling waves or do they shape EF gradients which in turn direct traveling waves? We reasoned that EF cannot be a mediating mechanism of wave direction if parameters exist where EF patterns do not match traveling wave direction. We found a set of parameters that produced reliable instrength-directed traveling waves but varying EF patterns (Supplementary Fig. 2) including some that were orthogonal to or opposing the flow potential (Supplementary Fig. 2d and e). We found that all but one flow potential across 100 randomly initialized simulations correlated significantly with the instrength gradient, while only one simulation had a significant flow potential - EF correlation (Supplementary Fig. 2a). Thus, instrength gradients guide traveling waves directly and not through mediating EF gradients. However, further systematic investigations of the precise relation between instrength gradients and EF are needed.“

Supplementary Figure 2. 2D network model with inconsistent effective frequency maps. **a** Swarmplots of Instrength - flow potential and effective frequency - flow potential correlations across 100 randomly initialized simulations of 10 s duration (1 s transients removed). **b** The average flow potential indicates that traveling waves propagated along the instrength gradient. **c** The average effective frequency shows clusters of varying effective frequency due to distinct effective frequency patterns in individual simulations (see **e**). **d** Example wave potentials of eight individual simulations; all of them are consistent with instrength-directed traveling waves. **e** Example effective frequency patterns of the same eight simulations shown in **d**. Effective frequency patterns vary between simulations inconsistent with the wave potential. **f** Example timeseries of simulated traveling waves propagating along the instrength-gradient despite an orthogonal effective frequency pattern (top row) or opposing effective frequency gradient (bottom row).

... especially considering that they state in lines 516ff: "...We hypothesize that external or self-generated stimuli affect traveling waves by changing oscillation frequency patterns."

We have now expanded this sentence into a paragraph on lines 796 - 815

"We used the tractography-derived number of streamlines as a proxy for coupling strength between brain regions. Cortical traveling wave direction and EF patterns are likely subject to additional factors such as cortical gradients of neuron density, synaptic spine count, receptor distributions, myelin content, cortical thickness, and excitation-inhibition ratio.^{24,81-87} While some cortical gradients may vary on faster timescales large-scale SC gradients remain relatively stable over longer time periods such as many days, weeks or years.^{48,49} Thus, SC gradients could contribute to traveling wave direction across brain states, e.g. similar traveling wave directions have been observed in the alpha band during rest and memory-tasks.^{3,9} Other studies have shown that traveling wave direction changes rapidly in response to tasks.^{70,71,88} We hypothesize that external or self-generated stimuli dynamically affect traveling waves by modulating IF.⁸⁹ For example, a stimulus arriving in the visual cortex could accelerate local oscillations⁹⁰ that interact with large-scale stable coupling and IF gradients to achieve cortex-wide processing through traveling waves. Alternatively, stimuli could induce large-scale IF gradients to direct cortical traveling waves, this could be achieved through thalamocortical loops preparing the cortex for incoming stimuli. We explored this latter mechanism in our 2D network model and found that stable instrength and dynamic IF gradients could cooperatively direct traveling waves. Alternatively, inter-regional coupling strength could be dynamically controlled – for instance by modulating the long-range

excitation-inhibition ratio.⁸⁶ To explain behavior, deeply understanding stable and dynamic coupling and IF gradients will be necessary.”

We thank the reviewer for this thought-provoking comment that led to new insights concerning the relationship between instrength gradients, wave direction and effective frequency.

Reviewer #1 (Remarks to the Author):

I congratulate the author for their elaborate revision of the manuscript. I think the authors did a great revision and the manuscript was substantially improved by the additional analyses.

Reviewer #2 (Remarks to the Author):

The authors have done a really strong and comprehensive job addressing my previous concerns. Whereas I had been concerned about the impact of the work in some cases and the degree to which the results backed up the paper's claims, I think all of these issues have been addressed in the revision. In particular, the revised paper is improved with the integration of the new introductory figure, which frames the work nicely, as well as the better statistical justification of the results. The clarity of the writing in the revised paper is also much improved, and the paper is now also much clearer regarding which aspects of the findings are novel compared to earlier work. Overall, the paper will make a nice contribution and is ready for publication, in my view.

Reviewer #3 (Remarks to the Author):

The authors addressed all comments and have successfully clarified the terms "instrength gradients," "intrinsic frequency" and "effective frequency" and how they relate to each other. The manuscript also benefits from the new figure 1 outlining different mechanisms for the emergence of traveling waves in weakly coupled oscillator networks.

One little thing remaining is in the new results section:

lines 230f: "... At this point, simulated activity varied: we observed spiral waves, plane waves, traveling waves, and full synchrony (Figure 3e middle)..."

The bit in BOLD is misleading as it suggests that plane waves and spiral waves are not traveling waves.

Point-by-point response

We thank the reviewers for their favorable assessment after the revision of our manuscript. We genuinely appreciate your time and effort and the contributions you have made to our work!

Reviewer #1 (Remarks to the Author):

I congratulate the author for their elaborate revision of the manuscript. I think the authors did a great revision and the manuscript was substantially improved by the additional analyses.

We once again thank the reviewer for their insightful suggestions, and we are happy that they approve our revised manuscript.

Reviewer #2 (Remarks to the Author):

The authors have done a really strong and comprehensive job addressing my previous concerns. Whereas I had been concerned about the impact of the work in some cases and the degree to which the results backed up the paper's claims, I think all of these issues have been addressed in the revision. In particular, the revised paper is improved with the integration of the new introductory figure, which frames the work nicely, as well as the better statistical justification of the results. The clarity of the writing in the revised paper is also much improved, and the paper is now also much clearer regarding which aspects of the findings are novel compared to earlier work. Overall, the paper will make a nice contribution and is ready for publication, in my view.

We are glad that we could convince the reviewer of our contribution's impact. The reviewer's critical comments have strengthened our revised manuscript notably.

Reviewer #3 (Remarks to the Author):

The authors addressed all comments and have successfully clarified the terms "instrength gradients," "intrinsic frequency" and "effective frequency" and how they relate to each other. The manuscript also benefits from the new figure 1 outlining different mechanisms for the emergence of traveling waves in weakly coupled oscillator networks.

We thank the reviewer for their comments and favorable assessment of our manuscript. We are pleased that our revision clarified the reviewer's concerns.

One little thing remaining is in the new results section:
lines 230f: "... At this point, simulated activity varied: we observed spiral waves, plane waves, traveling waves, and full synchrony (Figure 3e middle)...."

The bit in BOLD is misleading as it suggests that plane waves and spiral waves are not traveling waves.

We agree with the reviewer and have changed the wording of the above sentence to

*„At this point, simulated activity varied: we observed spiral waves, plane waves, **source-sink waves**, and full synchrony...”*

We hope this change clarifies that we do not want to suggest that spiral and plane waves are not traveling waves. We thank the reviewer for this comment!